# Diffusion Models With Learned Adaptive Noise

**Subham Sekhar Sahoo**
Cornell Tech, NYC, USA.
ssahoo@cs.cornell.edu

**Aaron Gokaslan**
Cornell Tech, NYC, USA.
akg87@cs.cornell.edu

**Chris De Sa**
Cornell University, Ithaca, USA.
cdesa@cs.cornell.edu

**Volodymyr Kuleshov**
Cornell Tech, NYC, USA.
kuleshov@cornell.edu

## Abstract

Diffusion models have gained traction as powerful algorithms for synthesizing high-quality images. Central to these algorithms is the diffusion process, a set of equations which maps data to noise in a way that can significantly affect performance. In this paper, we explore whether the diffusion process can be learned from data. Our work is grounded in Bayesian inference and seeks to improve log-likelihood estimation by casting the learned diffusion process as an approximate variational posterior that yields a tighter lower bound (ELBO) on the likelihood. A widely held assumption is that the ELBO is invariant to the noise process: our work dispels this assumption and proposes multivariate learned adaptive noise (MᴜLAN), a learned diffusion process that applies noise at different rates across an image. Specifically, our method relies on a multivariate noise schedule that is a function of the data to ensure that the ELBO is no longer invariant to the choice of the noise schedule as in previous works. Empirically, MᴜLAN sets a new state-of-the-art in density estimation on CIFAR-10 and ImageNet and reduces the number of training steps by 50%. We provide the code[1], along with a blog post and video tutorial on the project page: https://s-sahoo.com/MuLAN

## 1 Introduction

Diffusion models, inspired by the physics of heat diffusion, have gained traction as powerful tools for generative modeling, capable of synthesizing realistic, high-quality images [51, 16, 43, 14]. Central to these algorithms is the diffusion process, a gradual mapping of clean images into white noise. The reverse of this mapping defines the data-generating process we seek to learn—hence, its choice can significantly impact performance [22]. The conventional approach involves adopting a diffusion process derived from the laws of thermodynamics, which, albeit simple and principled, may be suboptimal due to its lack of adaptability to the dataset.

In this study, we investigate whether the notion of diffusion can be instead *learned from data*. Our motivating goal is to perform accurate log-likelihood estimation and probabilistic modelling, and we take an approach grounded in Bayesian inference [23]. We view the diffusion process as an approximate variational posterior: learning this process induces a tighter lower bound (ELBO) on the marginal likelihood of the data. Although previous work argued that the ELBO objective of a diffusion model is invariant to the choice of diffusion process [20, 22], we show that this claim is only true for the simplest types of univariate Gaussian noise: we identify a broader class of noising processes whose optimization yields significant performance gains.

---

[1] https://github.com/s-sahoo/MuLAN

38th Conference on Neural Information Processing Systems (NeurIPS 2024).

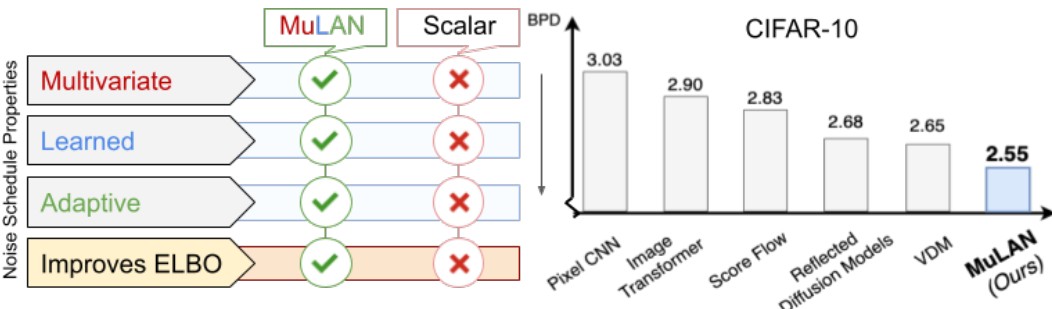

Figure 1: *(Left)* Comparison of noise schedule properties: Multivariate Learned Adaptive Noise schedule (MuLAN) (ours) versus a typical scalar noise schedule. Unlike scalar noise schedules, MuLAN's multivariate and input-adaptive properties improve likelihood. *(Right)* Likelihood in bits-per-dimension (BPD) on CIFAR-10 without data augmentation.

Specifically, we propose a new diffusion process, multivariate learned adaptive noise (MuLAN), which augments classical diffusion models [51, 20] with three innovations: a per-pixel polynomial noise schedule, an adaptive input-conditional noising process, and auxiliary latent variables. In practice, this method learns the schedule by which Gaussian noise is applied to different parts of an image, and allows tuning this noise schedule to the each image instance.

Our learned diffusion process yields improved log-likelihood estimates on two standard image datasets, CIFAR10 and ImageNet. Remarkably, we achieve state-of-the-art performance with less than half of the training time of previous methods. Our method also does not require any modifications to the underlying UNet architecture, which makes it compatible with most existing diffusion algorithms.

**Contributions**   In summary, our paper makes the following contributions:

1. We demonstrate that the ELBO of a diffusion model is not invariant to the choice of noise process for many types of noise, thus dispelling a common assumption in the field.

2. We introduce MuLAN, a learned noise process that adaptively adds multivariate Gaussian noise at different rates across an image in a way that is conditioned on arbitrary context (including the image itself).

3. We empirically demonstrate that learning the diffusion process speeds up training and matches the previous state-of-the-art models using **2x less compute**, and also achieves a new **state-of-the-art** in density estimation on CIFAR-10 and ImageNet

## 2   Background

A diffusion process $q$ transforms an input datapoint denoted by $\mathbf{x}_0$ and sampled from a distribution $q(\mathbf{x}_0)$ into a sequence of noisy latent variables $\mathbf{x}_t$ for $t \in [0, 1]$ by progressively adding Gaussian noise of increasing magnitude [51, 16, 53]. The marginal distribution of each latent is defined by $q(\mathbf{x}_t|\mathbf{x}_0) = \mathcal{N}(\mathbf{x}_t; \alpha_t \mathbf{x}_0, \sigma_t \boldsymbol{I})$ where the diffusion parameters $\alpha_t, \sigma_t \in \mathbb{R}^+$ implicitly define a noise schedule as a function of $t$, such that $\nu(t) = \alpha_t^2/\sigma_t^2$ is a monotonically decreasing function in $t$. Given any discretization of time into $T$ timesteps of width $1/T$, we define $t(i) = i/T$ and $s(i) = (i-1)/T$ and we use $\mathbf{x}_{0:1}$ to denote the subset of variables associated with these timesteps; the forward process $q$ can be shown to factorize into a Markov chain $q(\mathbf{x}_{0:1}) = q(\mathbf{x}_0) \prod_{i=1}^{T} q(\mathbf{x}_{t(i)}|\mathbf{x}_{s(i)})$.

The diffusion model $p_\theta$ is defined by a neural network (with parameters $\theta$) used to denoise the forward process $q$. Given a discretization of time into $T$ steps, $p$ factorizes as $p_\theta(\mathbf{x}_{0:1}) = p_\theta(\mathbf{x}_1) \prod_{i=1}^{T} p_\theta(\mathbf{x}_{s(i)}|\mathbf{x}_{t(i)})$. We treat the $\mathbf{x}_t$ for $t > 0$ as latent variables and fit $p_\theta$ by maximizing the evidence lower bound (ELBO) on the marginal log-likelihood given by:

$$\log p_\theta(\mathbf{x}_0) = \text{ELBO}(p_\theta, q) + \text{D}_{\text{KL}}[q(\mathbf{x}_{t(1):t(T)}|\mathbf{x}_0)\|p_\theta(\mathbf{x}_{t(1):t(T)}|\mathbf{x}_0)] \geq \text{ELBO}(p_\theta, q) \quad (1)$$

In most works, the noise schedule, as defined by $\nu(t)$, is either fixed or treated as a hyper-parameter [16, 3, 18]. Chen [3], Hoogeboom et al. [18] show that the noise schedule can have

a significant impact on sample quality. Kingma et al. [20] consider learning $\nu(t)$, but argue that the KL divergence terms in the ELBO are invariant to the choice of function $\nu$, except for the initial values $\nu(0), \nu(1)$, and they set these values to hand-specified constants in their experiments. They only consider learning $\nu$ for the purpose of minimizing the variance of the gradient of the ELBO. In this work, we show that the ELBO is not invariant to more complex forward processes.

## 3 Diffusion Models With Multivariate Learned Adaptive Noise

Here, we introduce a new diffusion process, multivariate learned adaptive noise (MuLAN), which introduces three innovations: a per-pixel polynomial noise schedule, a conditional noising process, and auxiliary-variable reverse diffusion. We describe these below.

### 3.1 Why Learned Diffusion?

Our goal is to perform accurate density estimation and probabilistic modelling, and we take an approach grounded in Bayesian inference [23]. Notice that the gap between the evidence lower bound ELBO$(p, q)$ and the marginal log-likelihood (MLL) in Eq. 1 is precisely the KL divergence $D_{KL}[q(\mathbf{x}_{t(1):t(T)}|\mathbf{x}_0)\|p_\theta(\mathbf{x}_{t(1):t(T)}|\mathbf{x}_0)]$ between the diffusion process $q$ over the latents $\mathbf{x}_t$ and the true posterior of the diffusion model. The diffusion process plays the role of a variational posterior $q$ in ELBO$(p, q)$; optimizing $q$ thus tightens the gap (MLL − ELBO).

This observation suggests that the ELBO can be made tighter by choosing a diffusion processes $q$ that is closer to the true posterior $p_\theta(\mathbf{x}_{t(1):t(T)}|\mathbf{x}_0)$. In fact, the key idea of variational inference is to optimize $\max_{q\in\mathcal{Q}}$ ELBO$(p, q)$ over a family of approximate posteriors $\mathcal{Q}$ to induce a tighter ELBO [23]. Most diffusion algorithms, however optimize $\max_{p\in\mathcal{P}}$ ELBO$(p, q)$ within some family $\mathcal{P}$ with a fixed $q$. Our work seeks to jointly optimize $\max_{p\in\mathcal{P}, q\in\mathcal{Q}}$ ELBO$(p, q)$; we will show in our experiments that this improves the likelihood estimation.

The task of log-likelihood estimation is directly motivated by applied problems such as data compression [31]. In that domain, arithmetic coding techniques can take a generative model and produce a compression algorithm that provably achieves a compression rate (in bits per dimension) that equals the model's log-likelihood [4]. Other applications of log-likelihood estimation include adversarial example detection [52], semi-supervised learning [5], and others.

Note that our primary focus is density estimation and probabilistic modeling rather than sample quality. The visual appeal of generated images (as measured by e.g., FID) correlates imperfectly with log-likelihood. We focus here on pushing the state-of-the-art in log-likelihood estimation, and while we report FID for completeness, we defer sample quality optimization to future work.

### 3.2 A Forward Diffusion Process With Multivariate Adaptive Noise

Next, our plan is to define a family of approximate posteriors $\mathcal{Q}$, as well as a family suitably matching reverse processes $\mathcal{P}$, such that the optimization problem $\max_{p\in\mathcal{P}, q\in\mathcal{Q}}$ ELBO$(p, q)$ is tractable and does not suffer from the aforementioned invariance to the choice of $q$. This subsection focuses on defining $\mathcal{Q}$; the next sections will show how to parameterize and train a reverse model $p \in \mathcal{P}$.

**Notation.** Given two vectors $\mathbf{a}$ and $\mathbf{b}$, we use the notation $\mathbf{ab}$ to represent the Hadamard product (element-wise multiplication). Additionally, we denote element-wise division of $\mathbf{a}$ by $\mathbf{b}$ as $\mathbf{a} / \mathbf{b}$. We denote the mapping diag(.) that takes a vector as input and produces a diagonal matrix as output.

#### 3.2.1 Multivariate Gaussian Noise Schedule

Intuitively, a multivariate noise schedule injects noise at different rates for each pixel of an input image. This enables adapting the diffusion process to spatial variations within the image. We will also see that this change is sufficient to make the ELBO no longer invariant in $q$.

Formally, we define a forward diffusion process with a multivariate noise schedule $q$ via the marginal for each latent noise variable $\mathbf{x}_t$ for $t \in [0, 1]$, where the marginal is given by:

$$q(\mathbf{x}_t|\mathbf{x}_0) = \mathcal{N}(\mathbf{x}_t; \boldsymbol{\alpha}_t\mathbf{x}_0, \text{diag}(\boldsymbol{\sigma}_t^2)), \tag{2}$$

where $\mathbf{x}_t, \mathbf{x}_0 \in \mathbb{R}^d$, $\boldsymbol{\alpha}_t, \boldsymbol{\sigma}_t \in \mathbb{R}^d_+$ and $d$ is the dimensionality of the input data. The $\boldsymbol{\alpha}_t, \boldsymbol{\sigma}_t$ denote varying amounts of signal associated with each component (i.e., each pixel) of $\mathbf{x}_0$ as a function of

time $t(i)$. We define the multivariate signal-to-noise ratio as $\boldsymbol{\nu}(t) = \boldsymbol{\alpha}_t^2/\boldsymbol{\sigma}_t^2$ and choose $\boldsymbol{\alpha}_t, \boldsymbol{\sigma}_t$ so that $\boldsymbol{\nu}(t)$ decreases monotonically in $t$ along all dimensions and is differentiable in $t \in [0, 1]$. Let $\boldsymbol{\alpha}_{t|s} = \boldsymbol{\alpha}_t/\boldsymbol{\alpha}_s$ and $\boldsymbol{\sigma}_{t|s}^2 = \boldsymbol{\sigma}_t^2 - \boldsymbol{\alpha}_{t|s}^2/\boldsymbol{\sigma}_s^2$ with all operations applied elementwise.

These marginals induce transition kernels between steps $s < t$ given by (Suppl. 19):

$$q(\mathbf{x}_s|\mathbf{x}_t, \mathbf{x}_0) = \mathcal{N}\left(\mathbf{x}_s; \boldsymbol{\mu}_q = \frac{\boldsymbol{\alpha}_{t|s}\boldsymbol{\sigma}_s^2}{\boldsymbol{\sigma}_t^2}\mathbf{x}_t + \frac{\boldsymbol{\sigma}_{t|s}^2\boldsymbol{\alpha}_s}{\boldsymbol{\sigma}_t^2}\mathbf{x}_0, \ \boldsymbol{\Sigma}_q = \text{diag}\left(\frac{\boldsymbol{\sigma}_s^2\boldsymbol{\sigma}_{t|s}^2}{\boldsymbol{\sigma}_t^2}\right)\right). \tag{3}$$

In Sec. 3.5, we argue that this class of diffusion process $\mathcal{Q}$ induces an ELBO that is not invariant to $q \in \mathcal{Q}$. The ELBO consists of a line integral along the diffusion trajectory specified by $\boldsymbol{\nu}(t)$. A line integrand is almost always path-dependent, unless its integral corresponds to a conservative force field, which is rarely the case for a diffusion process [55]. See Sec. 3.5 for details.

### 3.2.2 Adaptive Noise Schedule Conditioned On Context

Next, we extend the diffusion process to support context-adaptive noise. This enables injecting noise in a way that is dependent on the features of an image. Formally, suppose we have access to a context variable $\mathbf{c} \in \mathbb{R}^m$ which encapsulates high-level information regarding $\mathbf{x}_0$. Examples of $\mathbf{c}$ could be a class label, a vector of attributes (e.g., features characterizing a human face), or even the input $\mathbf{x}_0$ itself. We define the marginal of the latent $\mathbf{x}_t$ in the forward process as $q(\mathbf{x}_t|\mathbf{x}_0, \mathbf{c}) = \mathcal{N}(\mathbf{x}_t; \boldsymbol{\alpha}_t(\mathbf{c})\mathbf{x}_0, \boldsymbol{\sigma}_t^2(\mathbf{c}))$; the reverse process can be similarly derived (Suppl. 19) as:

$$q(\mathbf{x}_s|\mathbf{x}_t, \mathbf{x}_0, \mathbf{c}) = \mathcal{N}\left(\boldsymbol{\mu}_q = \frac{\boldsymbol{\alpha}_{t|s}(\mathbf{c})\boldsymbol{\sigma}_s^2(\mathbf{c})}{\boldsymbol{\sigma}_t^2(\mathbf{c})}\mathbf{x}_t + \frac{\boldsymbol{\sigma}_{t|s}^2(\mathbf{c})\boldsymbol{\alpha}_s(\mathbf{c})}{\boldsymbol{\sigma}_t^2(\mathbf{c})}\mathbf{x}_0, \ \boldsymbol{\Sigma}_q = \text{diag}\left(\frac{\boldsymbol{\sigma}_s^2(\mathbf{c})\boldsymbol{\sigma}_{t|s}^2(\mathbf{c})}{\boldsymbol{\sigma}_t^2(\mathbf{c})}\right)\right),$$
$$\tag{4}$$

where the diffusion parameters $\boldsymbol{\alpha}_t, \boldsymbol{\sigma}_t$ are now conditioned on $\mathbf{c}$ via a neural network.

Specifically, we parameterize the diffusion parameters $\boldsymbol{\alpha}_t(\mathbf{c}), \boldsymbol{\sigma}_t(\mathbf{c}), \boldsymbol{\nu}(t, \mathbf{c})$ as $\boldsymbol{\alpha}_t^2(\mathbf{c}) = \text{sigmoid}(-\boldsymbol{\gamma}_\phi(\mathbf{c}, t))$, $\boldsymbol{\sigma}_t^2(\mathbf{c}) = \text{sigmoid}(\boldsymbol{\gamma}_\phi(\mathbf{c}, t))$, and $\boldsymbol{\nu}(\mathbf{c}, t) = \exp(-\boldsymbol{\gamma}_\phi(\mathbf{c}, t))$. Here, $\boldsymbol{\gamma}_\phi(\mathbf{c}, t) : \mathbb{R}^m \times [0, 1] \to [\gamma_{\min}, \gamma_{\max}]^d$ is a neural network with the property that $\boldsymbol{\gamma}_\phi(\mathbf{c}, t)$ is monotonic in $t$. Following Kingma et al. [20], Zheng et al. [65], we set $\gamma_{\min} = -13.30, \gamma_{\max} = 5.0$.

We explore various parameterizations for $\boldsymbol{\gamma}_\phi(\mathbf{c}, t)$. These schedules are designed in a manner that guarantees $\boldsymbol{\gamma}_\phi(\mathbf{c}, 0) = \gamma_{\min}\mathbf{1_d}$ and $\boldsymbol{\gamma}_\phi(\mathbf{c}, 1) = \gamma_{\max}\mathbf{1_d}$, Below, we list these parameterizations. The polynomial parameterization is novel to our work and yields significant performance gains.

**Monotonic Neural Network [20].** We use the monotonic neural network $\boldsymbol{\gamma}_{\text{vdm}}(t)$, proposed in VDM to express $\boldsymbol{\gamma}$ as a function of $t$ such that $\boldsymbol{\gamma}_{\text{vdm}}(t) : [0, 1] \to [\gamma_{\min}, \gamma_{\max}]^d$. Then we use FiLM conditioning [38] in the intermediate layers of this network via a neural network that maps $\mathbf{z}$. The activations of the FiLM layer are constrained to be positive.

**Polynomial.** (Ours) We express $\boldsymbol{\gamma}_\phi(\mathbf{c}, t)$ as a monotonic degree 5 polynomial in $t$. Details about the exact functional form of this polynomial and its implementation can be found in Suppl. E.2.

### 3.3 Auxiliary-Variable Reverse Diffusion Processes

In principle, we can fit a normal diffusion model in conjunction with our proposed forward diffusion process. However, variational inference suggests that the variational and the true posterior ought to have the same dependency structure: that is the only way for the KL divergence between these two distributions to be zero. Thus, we introduce a class of approximate reverse processes $\mathcal{P}$ that match the structure of $\mathcal{Q}$ and that are naturally suitable for joint optimization $\max_{p \in \mathcal{P}, q \in \mathcal{Q}} \text{ELBO}(p, q)$.

Formally, we define a diffusion model where the reverse diffusion process is conditioned on the context $\mathbf{c}$. Specifically, given any discretization of $t \in [0, 1]$ into $T$ time steps as in Sec. 2, we introduce a context-conditional diffusion model $p_\theta(\mathbf{x}_{0:1}|\mathbf{c})$ that factorizes as the Markov chain

$$p_\theta(\mathbf{x}_{0:1}|\mathbf{c}) = p_\theta(\mathbf{x}_1|\mathbf{c}) \prod_{i=1}^{T} p_\theta(\mathbf{x}_{s(i)}|\mathbf{x}_{t(i)}, \mathbf{c}). \tag{5}$$

Given that the true reverse process is a Gaussian as specified in Eq. 4, the ideal $p_\theta$ matches this parameterization (the proof mirrors that of regular diffusion models; Suppl. D), which yields

$$p_\theta(\mathbf{x}_s|\mathbf{c}, \mathbf{x}_t) = \mathcal{N}\left(\boldsymbol{\mu}_p = \frac{\boldsymbol{\alpha}_{t|s}(\mathbf{c})\boldsymbol{\sigma}_s^2(\mathbf{c})}{\boldsymbol{\sigma}_t^2(\mathbf{c})}\mathbf{x}_t + \frac{\boldsymbol{\sigma}_{t|s}^2(\mathbf{c})\boldsymbol{\alpha}_s(\mathbf{c})}{\boldsymbol{\sigma}_t^2(\mathbf{c})}\mathbf{x}_\theta(\mathbf{x}_t, t), \boldsymbol{\Sigma}_p = \text{diag}\left(\boldsymbol{\sigma}_s^2(\mathbf{c})\boldsymbol{\sigma}_{t|s}^2(\mathbf{c})/\boldsymbol{\sigma}_t^2(\mathbf{c})\right)\right),$$
(6)

where $\mathbf{x}_\theta(\mathbf{x}_t, t)$, is a neural network that approximates $\mathbf{x}_0$. Instead of parameterizing $\mathbf{x}_\theta(\mathbf{x}_t, t)$ directly using a neural network, we consider two other parameterizations. One is the noise parameterization [16] where $\boldsymbol{\epsilon}_\theta(\mathbf{x}_t, \mathbf{c}, t)$ is the denoising model which is parameterized as $\boldsymbol{\epsilon}_\theta(\mathbf{x}_t, t) = (\mathbf{x}_t - \boldsymbol{\alpha}_t(\mathbf{c})\mathbf{x}_\theta(\mathbf{x}_t, t, \mathbf{c}))/\boldsymbol{\sigma}_t(\mathbf{c})$; see Suppl. E.1.1 and the other is v-parameterization [45] where $\mathbf{v}_\theta(\mathbf{x}_t, \mathbf{c}, t)$ is a neural network that models $\mathbf{v}_\theta(\mathbf{x}_t, \mathbf{c}, t) = (\boldsymbol{\alpha}_t(\mathbf{c})\mathbf{x}_t - \mathbf{x}_\theta(\mathbf{x}_t, \mathbf{c}, t))/\boldsymbol{\sigma}_t(\mathbf{c})$; see Suppl. E.1.2.

### 3.3.1 Challenges in Conditioning on Context

Note that the model $p_\theta(\mathbf{x}_{0:1}|\mathbf{c})$ implicitly assumes the availability of $\mathbf{c}$ at generation time. Sometimes, this context may be available, such as when we condition on a label. We may then fit a conditional diffusion process with a standard diffusion objective $\mathbb{E}_{\mathbf{x}_0, c}[\text{ELBO}(\mathbf{x}_0, p_\theta(\mathbf{x}_{0:1}|\mathbf{c}), q_\phi(\mathbf{x}_{0:1}|\mathbf{c})]$, in which both the forward and the backward processes are conditioned on $\mathbf{c}$ (see Sec. 3.4).

When $\mathbf{c}$ is not known at generation time, we may fit a model $p_\theta$ that does not condition on $\mathbf{c}$. Unfortunately, this also forces us to define $p_\theta(\mathbf{x}_s|\mathbf{x}_t) = \mathcal{N}(\boldsymbol{\mu}_p(\mathbf{x}_t, t), \boldsymbol{\Sigma}_p(\mathbf{x}_t, t))$ where $\boldsymbol{\mu}_p(\mathbf{x}_t, t), \boldsymbol{\Sigma}_p(\mathbf{x}_t, t)$ is parameterized directly by a neural network. We can no longer use a noise parameterization $\boldsymbol{\epsilon}_\theta(\mathbf{x}_t, t) = (\mathbf{x}_t - \boldsymbol{\alpha}_t(\mathbf{c})\mathbf{x}_\theta(\mathbf{x}_t, t, \mathbf{c}))/\boldsymbol{\sigma}_t(\mathbf{c})$ because it requires us to compute $\boldsymbol{\alpha}_t(\mathbf{c})$ and $\boldsymbol{\sigma}_t(\mathbf{c})$, which we do not know. Since noise parameterization plays a key role in the sample quality of diffusion models [16], this approach limits performance.

### 3.3.2 Conditioning Noise on an Auxiliary Latent Variable

We propose an alternative strategy for learning conditional forward and reverse processes $p, q$ that feature the same structure and hence support efficient noise parameterization. Our approach is based on the introduction of auxiliary variables [60], which lift the distribution $p_\theta$ into an augmented latent space. Experiments (Suppl. D.3) and theory (Suppl. D) confirm that this approach performs better than parameterizing $\mathbf{c}$ using a neural network, $\mathbf{c}_\theta(\mathbf{x}_t, t)$.

Specifically, we introduce an auxiliary latent variable $\mathbf{z} \in \mathbb{R}^m$ and define a lifted $p_\theta(\mathbf{x}, \mathbf{z}) = p_\theta(\mathbf{x}|\mathbf{z})p_\theta(\mathbf{z})$, where $p_\theta(\mathbf{x}|\mathbf{z})$ is the conditional diffusion model from Eq. 5 (with context $\mathbf{c}$ set to $\mathbf{z}$) and $p_\theta(\mathbf{z})$ is a simple prior (e.g., unit Gaussian or fully factored Bernoulli). The latents $\mathbf{z}$ can be interpreted as a high-level semantic representation of $\mathbf{x}$ that conditions both the forward and the reverse processes. Unlike $\mathbf{x}_{0:1}$, the $\mathbf{z}$ are not constrained to have a particular dimension and can be a low-dimensional vector of latent factors of variation. They can be continuous or discrete. The learning objective for the lifted $p_\theta$ is given by:

$$\log p_\theta(\mathbf{x}_0) \geq \mathbb{E}_{q_\phi(\mathbf{z}|\mathbf{x}_0)}[\log p_\theta(\mathbf{x}_0|\mathbf{z})] - \text{D}_{\text{KL}}(q_\phi(\mathbf{z}|\mathbf{x}_0)\|p_\theta(\mathbf{z})) \tag{7}$$

$$\geq \mathbb{E}_{q_\phi(\mathbf{z}|\mathbf{x}_0)}\text{ELBO}(p_\theta(\mathbf{x}_{0:1}|\mathbf{z}), q_\phi(\mathbf{x}_{0:1}|\mathbf{z})) - \text{D}_{\text{KL}}(q_\phi(\mathbf{z}|\mathbf{x}_0)\|p_\theta(\mathbf{z})), \tag{8}$$

where $\text{ELBO}(p_\theta(\mathbf{x}_{0:1}|\mathbf{z}), q_\phi(\mathbf{x}_{0:1}|\mathbf{z}))$ denotes the variational lower bound (VLB) of a diffusion model (defined in Eq. 1) with a forward process $q_\phi(\mathbf{x}_{0:1}|\mathbf{z})$ (defined in Eq. 4 and Sec. 3.2.2) and and an approximate reverse process $p_\theta(\mathbf{x}_{0:1}|\mathbf{z})$ (defined in Eq. 5), both conditioned on $\mathbf{z}$. The distribution $q_\phi(\mathbf{z}|\mathbf{x}_0)$ is an approximate posterior for $\mathbf{z}$ parameterized by a neural network with parameters $\phi$.

Crucially, note that in the learning objective (Eq. 8), the context, which in this case is $\mathbf{z}$, is available at training time in both the forward and reverse processes. At generation time, we can still obtain a valid context vector by sampling an auxiliary latent from $p_\theta(\mathbf{z})$. Thus, this approach addresses the aforementioned challenges and enables us to use the noise parameterization in Eq. 6.

Although we apply Jensen's inequality twice to get (8), this also enables us to learn the noise process, which significantly offsets any potential increase in ELBO gap reduction and improves $\text{ELBO}(p_\theta(\mathbf{x}_{0:1}|\mathbf{z}), q_\phi(\mathbf{x}_{0:1}|\mathbf{z}))$ by optimizing over a more expressive class of posteriors. This claim is empirically validated in Table 2.

## 3.4 Variational Lower Bound

Next, we derive a precise formula for the learning objective (8) of the auxiliary-variable diffusion model. Using the objective of a diffusion model in (1) we can write (8) as the sum of four terms:

$$\log p_\theta(\mathbf{x}_0) \geq \mathbb{E}_{q_\phi}[\mathcal{L}_{\text{recons}} + \mathcal{L}_{\text{diffusion}} + \mathcal{L}_{\text{prior}} + \mathcal{L}_{\text{latent}}], \tag{9}$$

The reconstruction loss, $\mathcal{L}_{\text{recons}}$, can be (stochastically and differentiably) estimated using standard techniques; see [23], $\mathcal{L}_{\text{prior}} = -\text{D}_{\text{KL}}[q_\phi(\mathbf{x}_1|\mathbf{x}_0, \mathbf{z})\|p_\theta(\mathbf{x}_1)]$ is the diffusion prior term, $\mathcal{L}_{\text{latent}} = -\text{D}_{\text{KL}}[q_\phi(\mathbf{z}|\mathbf{x}_0)\|p_\theta(\mathbf{z})]$ is the latent prior term, and $\mathcal{L}_{\text{diffusion}}$ is the diffusion loss term, which we examine below. The complete derivation is given in Suppl. E.3.

### 3.4.1 Diffusion Loss

**Discrete-Time Diffusion.** We start by defining $p_\theta$ in discrete time, and as in Sec. 2, we let $T > 0$ be the number of total time steps and define $t(i) = i/T$ and $s(i) = (i-1)/T$ as indexing variables over the time steps. We also use $\mathbf{x}_{0:1}$ to denote the subset of variables associated with these timesteps. Starting with the expression in Eq. 1 and following the steps in Suppl. E, we can write $\mathcal{L}_{\text{diffusion}}$ as:

$$\mathcal{L}_{\text{diffusion}} = -\sum_{i=2}^{T} \text{D}_{\text{KL}}[q_\phi(\mathbf{x}_{s(i)}|\mathbf{x}_{t(i)}, \mathbf{x}_0, \mathbf{z})\|p_\theta(\mathbf{x}_{s(i)}|\mathbf{x}_{t(i)}, \mathbf{z})]$$

$$= \frac{1}{2}\sum_{i=2}^{T}[(\boldsymbol{\epsilon}_t - \boldsymbol{\epsilon}_\theta(\mathbf{x}_t, \mathbf{z}, t(i)))^\top \text{diag}\left(\boldsymbol{\gamma}(\mathbf{z}, s(i)) - \boldsymbol{\gamma}(\mathbf{z}, t(i))\right)(\boldsymbol{\epsilon}_t - \boldsymbol{\epsilon}_\theta(\mathbf{x}_t, \mathbf{z}, t(i)))] \tag{10}$$

**Continuous-Time Diffusion.** We can also consider the limit of the above objective as we take an infinitesimally small partition of $t \in [0, 1]$, which corresponds to the limit when $T \to \infty$. In Suppl. E we show that taking this limit of Eq. 10 yields the continuous-time diffusion loss:

$$\mathcal{L}_{\text{diffusion}} = -\frac{1}{2}\mathbb{E}_{t \sim [0,1]}[(\boldsymbol{\epsilon}_t - \boldsymbol{\epsilon}_\theta(\mathbf{x}_t, \mathbf{z}, t))^\top \text{diag}\left(\nabla_t \boldsymbol{\gamma}(\mathbf{z}, t)\right)(\boldsymbol{\epsilon}_t - \boldsymbol{\epsilon}_\theta(\mathbf{x}_t, \mathbf{z}, t))] \tag{11}$$

where $\nabla_t \boldsymbol{\gamma}(\mathbf{z}, t) \in \mathbb{R}^d$ denotes the Jacobian of $\boldsymbol{\gamma}(\mathbf{z}, t)$ with respect to the scalar $t$. We observe that the limit of $T \to \infty$ yields improved performance, matching the existing theoretical argument by Kingma et al. [20].

### 3.4.2 Auxiliary latent loss

We try two different kinds of priors for $p_\theta(\mathbf{z})$: discrete ($\mathbf{z} \in \{0, 1\}^m$) and continuous ($\mathbf{z} \in \mathbb{R}^m$).

**Continuous Auxiliary Latents.** In the case where $\mathbf{z}$ is continuous, we select $p_\theta(\mathbf{z})$ as $\mathcal{N}(\mathbf{0}, \mathbf{I}_m)$. This leads to the following KL loss term:
$\text{D}_{\text{KL}}(q_\phi(\mathbf{z}|\mathbf{x}_0)\|p_\theta(\mathbf{z})) = \frac{1}{2}(\boldsymbol{\mu}^\top(\mathbf{x}_0)\boldsymbol{\mu}(\mathbf{x}_0)) + \text{tr}(\boldsymbol{\Sigma}^2(\mathbf{x}_0) - \mathbf{I}_m) - \log|\boldsymbol{\Sigma}^2(\mathbf{x}_0)|).$

**Discrete Auxiliary Latents.** In the case where $\mathbf{z}$ is discrete, we select $p_\theta(\mathbf{z})$ as a uniform distribution. Let $\mathbf{z} \in \{0, 1\}^m$ be a $k$-hot vector sampled from a discrete Exponential Family distribution $p_\theta(\mathbf{z}; \theta)$ with logits $\theta$. Niepert et al. [34] show that $\mathbf{z} \sim p_\theta(\mathbf{z}; \theta)$ is equivalent to $\mathbf{z} = \arg\max_{y \in Y}\langle \theta + \epsilon_g, y\rangle$ where $\epsilon_g$ denotes the sum of gamma distribution Suppl. F, $Y$ denotes the set of all $k$-hot vectors of some fixed length $m$. For $k > 1$, To differentiate through the $\arg\max$ we use a relaxed estimator, Identity, as proposed by Sahoo et al. [44]. This leads to the following KL loss term: $\text{D}_{\text{KL}}(q_\phi(\mathbf{z}|\mathbf{x}_0)\|p_\theta(\mathbf{z})) = -\sum_{i=1}^{m} q_\phi(\mathbf{z}|\mathbf{x}_0)_i(\log q_\phi(\mathbf{z}|\mathbf{x}_0)_i + \log m).$

## 3.5 The Variational Lower Bound as a Line Integral Over The Noise Schedule

Having defined our loss, we now return to the question of whether it is invariant to the choice of diffusion process. Notice that we may rewrite Eq. 11 in the following vectorized form:

$$\mathcal{L}_{\text{diffusion}} = -\frac{1}{2}\int_0^1 (\mathbf{x}_0 - \mathbf{x}_\theta(\mathbf{x}_t, \mathbf{z}, t))^2 \cdot \nabla_t \boldsymbol{\nu}(\mathbf{z}, t)\text{d}t \tag{12}$$

where the square is applied elementwise. We seek to rewrite (12) as a line integral $\int_a^b \mathbf{f}(\mathbf{r}(t)) \cdot \frac{\mathrm{d}}{\mathrm{d}t}\mathbf{r}(t)\mathrm{d}t$ for some vector field $\mathbf{f}$ and trajectory $\mathbf{r}(t)$. Recall that $\boldsymbol{\nu}(\mathbf{z}, t)$ is monotonically decreasing in each coordinate as a function of $t$; hence, it is invertible on its image, and we can write $t = \boldsymbol{\nu}_{\mathbf{z}}^{-1}(\boldsymbol{\nu}(\mathbf{z}, t))$ for some $\boldsymbol{\nu}_z^{-1}$. Let $\bar{\mathbf{x}}_\theta(\mathbf{x}_{\boldsymbol{\nu}(\mathbf{z},t)}, \mathbf{z}, \boldsymbol{\nu}(\mathbf{z}, t)) \equiv \mathbf{x}_\theta(\mathbf{x}_{\boldsymbol{\nu}_z^{-1}(\boldsymbol{\nu}(\mathbf{z},t))}, \mathbf{z}, \boldsymbol{\nu}_{\mathbf{z}}^{-1}(\boldsymbol{\nu}(\mathbf{z}, t)))$ and note that for all $t$, we can write $\mathbf{x}_t$ as $\mathbf{x}_{\boldsymbol{\nu}(\mathbf{z},t)}$; see Eq. 30, and have $\bar{\mathbf{x}}_\theta(\mathbf{x}_{\boldsymbol{\nu}(\mathbf{z},t)}, \mathbf{z}, \boldsymbol{\nu}(\mathbf{z}, t)) \equiv \mathbf{x}_\theta(\mathbf{x}_t, \mathbf{z}, t)$. We can then write the integral in (12) as $\int_0^1 (\mathbf{x}_0 - \bar{\mathbf{x}}_\theta(\mathbf{x}_{\boldsymbol{\nu}(\mathbf{z},t)}, \mathbf{z}, \boldsymbol{\nu}(\mathbf{z}, t)))^2 \cdot \frac{\mathrm{d}}{\mathrm{d}t}\boldsymbol{\nu}(\mathbf{z}, t)\rangle \mathrm{d}t$, which is a line integral with $\mathbf{f}(\mathbf{r}(t)) \equiv (\mathbf{x}_0 - \bar{\mathbf{x}}_\theta(\mathbf{x}_{\boldsymbol{\nu}(\mathbf{z},t)}, \mathbf{z}, \boldsymbol{\nu}(\mathbf{z}, t)))^2$ and $\mathbf{r}(t) \equiv \boldsymbol{\nu}(\mathbf{z}, t)$.

**Intuitive explanation.** Imagine piloting a plane across a region with cyclones and strong winds, as shown in Fig. 5. Plotting a direct, straight-line course through these adverse weather conditions requires more fuel and effort due to increased resistance. By navigating around the cyclones and winds, however, the plane reaches its destination with less energy, even if the route is longer.

This intuition translates into mathematical and physical terms. The plane's trajectory is denoted by $\mathbf{r}(t) \in \mathbb{R}_+^n$, while the forces acting on it are represented by $\mathbf{f}(\mathbf{r}(t)) \in \mathbb{R}^n$. The work required to navigate is given by $\int_0^1 \mathbf{f}(\mathbf{r}(t)) \cdot \frac{d}{dt}\mathbf{r}(t), dt$. Here, the work depends on the trajectory because $\mathbf{f}(\mathbf{r}(t))$ is not a conservative field.

This concept also applies to the diffusion NELBO. From Eq. 12, it's clear that the trajectory $\mathbf{r}(t)$ is parameterized by the noise schedule $\boldsymbol{\nu}(\mathbf{z}, t)$, which is influenced by complex forces, $\mathbf{f}$ (analogous to weather patterns), represented by the dimension-wise reconstruction error of the denoising model, $(\mathbf{x}_0 - \mathbf{x}_\theta(\mathbf{x}_t, \mathbf{z}, t))^2$. Thus, the diffusion loss, $\mathcal{L}_{\text{diffusion}}$, can be interpreted as the work done along the trajectory $\boldsymbol{\nu}(\mathbf{z}, t)$ in the presence of these vector field forces $\mathbf{f}$. By learning the noise schedule, we can avoid "high-resistance" paths (those where the loss accumulates rapidly), thereby minimizing the overall "energy" expended, as measured by the NELBO. Since the diffusion process corresponds to non-conservative force fields, as noted in Spinney & Ford [55], different noise schedules should yield different NELBOs—a result supported by our empirical findings. In Suppl. E.5, we show that variational diffusion models are limited to linear trajectories $\boldsymbol{\nu}(t)$, rendering their objective invariant to the noise schedule. In contrast, our approach learns a multivariate $\boldsymbol{\nu}$, enabling paths that achieve a better ELBO.

## 4 Experiments

This section reports experiments on the CIFAR-10 [25] and ImageNet-32 [58] datasets. We don't employ data augmentation and we use the same architecture and settings as in the VDM model [20]. The encoder, $q_\phi(\mathbf{z}|\mathbf{x})$, is modeled using a sequence of 4 ResNet blocks which is much smaller than the denoising network that uses 32 such blocks (i.e., we increase parameter count by only about 10%); the noise schedule $\boldsymbol{\gamma}_\phi$ is modeled using a two-layer MLP. In all our experiments, we use discrete auxiliary latents with $m = 50$ and $k = 15$. A detailed description can be found in Suppl. G.

### 4.1 Training Speed

In these experiments, we replace VDM's noise process with MuLAN. On CIFAR-10, MuLAN **attains VDM's likelihood score of 2.65 in just 2M steps, compared to VDM's 10M steps** 1). When trained on 4 V100 GPUs, VDM achieves a training rate of 2.6 steps/second, while MuLAN trains slightly slower at 2.24 steps/second due to the inclusion of an additional encoder network. However, despite this slower training pace, VDM requires 30 days to reach a BPD of 2.65, whereas Mulan achieves the same BPD within a significantly shorter timeframe of 10 days. On ImageNet-32, VDM integrated with MuLAN reaches a likelihood of 3.71 in half the time, **achieving this score in 1M steps versus the 2M steps required by VDM**.

### 4.2 Likelihood Estimation

In Table 2, we also compare MuLAN with other recent methods on CIFAR-10 and ImageNet-32. MuLAN was trained using v-parameterization for 8M steps on CIFAR-10 and 2M steps on Imagenet-32. During inference, we extract the underlying probability flow ODE and use it to estimate the log-likelihood; see Suppl. I.2. Our algorithm **establishes a new state-of-the-art in density estimation** on both ImageNet-32 and CIFAR-10. In Table 8, we also compute variational lower

Table 1: Likelihood in bits per dimension (BPD) based on the Variational Lower Bound (VLB) estimate (Suppl. I.1), sample quality (FID scores) and number of function evaluations (NFE) on CIFAR-10, for vanilla VDM and VDM when endowed with MuLAN. FID and NFE were computed for 10k samples generated using an adaptive-step ODE solver. Both methods use noise parameterization (Suppl. E.1.1).

| Model | CIFAR-10 | | | | ImageNet | | | |
|---|---|---|---|---|---|---|---|---|
| | Steps | VLB ($\downarrow$) | FID ($\downarrow$) | NFE ($\downarrow$) | Steps | VLB ($\downarrow$) | FID ($\downarrow$) | NFE ($\downarrow$) |
| VDM [20] | 10M | 2.65 | 23.91 | 56 | 2M | 3.72 | 14.26 | **56** |
| + MuLAN | **2M** | 2.65 | 18.54 | 55 | **1M** | 3.72 | 15.00 | 62 |
| + MuLAN | 10M | **2.60** | **17.62** | **50** | 2M | **3.71** | **13.19** | 62 |

Table 2: Likelihood in bits per dimension (BPD) on the test set of CIFAR-10 and ImageNet. Results with "/" means they are not reported in the original papers. Model types are autoregressive (AR), normalizing flows (Flow), diffusion models (Diff). We only compare with results achieved without data augmentation.

| Model | Type | CIFAR-10 ($\downarrow$) | ImageNet ($\downarrow$) |
|---|---|---|---|
| PixelCNN [57] | AR | 3.03 | 3.83 |
| Image Transformer [35] | AR | 2.90 | 3.77 |
| DDPM [16] | Diff | $\leq 3.69$ | / |
| ScoreFlow [54] | Diff | 2.83 | 3.76 |
| VDM [20] | Diff | $\leq 2.65$ | $\leq 3.72$ |
| Flow Matching [28] | Flow | 2.99 | / |
| Reflected Diffusion Models [30] | Diff | 2.68 | 3.74 |
| MuLAN (**Ours**) | Diff | **2.55** $\pm 10^{-3}$ | **3.67** $\pm 10^{-3}$ |

bounds (VLBs) of $\leq 2.59$ and $\leq 3.71$ on CIFAR-10 and ImageNet, respectively. Each bound improves over published results (Table 2); our true NLLs (via flow ODEs) are even lower.

## 4.3 Alternative Learned Diffusion Methods

Concurrent work that seeks to improve log-likelihood estimation by learning the forward diffusion process includes Neural Diffusion Models (NDMs) [1] and DiffEnc [33]. In NDMs, the noise schedule is fixed, but the mean of each marginal $q(\mathbf{x}_t|x_0)$ is learned, while DiffEnc adds a correction term to $q$. Diffusion normalizing flows (DNFs) represent an earlier effort where $q$ is a normalizing flow trained by backpropagating through sampling. In Table 3, we compare against NDMs, DiffEnc, and DNFs on the CIFAR-10 dataset, using the authors' published results; note that their published ImageNet numbers are either not available or are reported on a different dataset version that is not comparable. Our approach to learned diffusion outperforms previous and concurrent work.

Table 3: Likelihood in bits per dimension (bpd) on CIFAR-10 for learned diffusion methods.

| Model | NLL ($\downarrow$) |
|---|---|
| DNF [64] | 3.04 |
| NDM [1] | $\leq 2.70$ |
| DiffEnc [33] | $\leq 2.62$ |
| MuLAN | **2.55** |

## 4.4 Ablation Analysis And Additional Experiments

Due to the expensive cost of training, we only performed ablation studies on CIFAR-10 with a reduced batch size of 64 and trained the model for 2.5M training steps. In Fig. 2a we ablate each component of MuLAN: when we remove the conditioning on an auxiliary latent space from MuLAN so that we have a multivariate noise schedule that is solely conditioned on time $t$, our performance becomes comparable to that of VDM, on which our model is based. Changing to a scalar noise schedule based on latent variable $\mathbf{z}$ initially underperforms compared to VDM. This drop aligns with our likelihood formula (Eq. 6) which includes $D_{KL}(q_\phi(\mathbf{z}|\mathbf{x}_0)|p_\theta(\mathbf{z}))$, an extra term not in VDM. The input-conditioned scalar schedule doesn't offer any advantage over the scalar schedule used in VDM. This is due to the reasons outlined in Sec. 3.5.

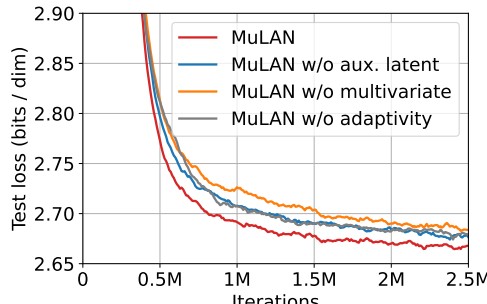 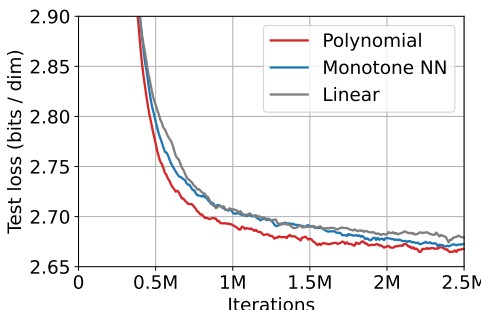

(a) In MuLAN w/o aux. latent, the noise isn't conditioned on a latent. MuLAN w/o multivariate uses a scalar noise schedule. MuLAN w/o adaptivity has a linear schedule and no auxiliary latents.

(b) MuLAN with different noise schedule parameterizations: polynomial, monotonic neural network, and linear. Our proposed polynomial parameterization performs the best.

Figure 2: Ablating components of MuLAN on CIFAR-10 over 2.5M steps with batch size of 64.

**Perceptual Quality**   While perceptual quality is not the focus of this work, we report FID numbers for the VDM model and MuLAN (Table 1). We use RK45 ODE solver to generate samples by solving the reverse time Flow ODE (Eq. 76). We observe that MuLAN does not degrade FIDs, while improving log-likelihood estimation. Note that MuLAN does not incorporate many tricks that improve FID such as exponential moving averages, truncations, specialized learning schedules, etc.; our FID numbers can be improved in future work using these techniques.

**Loss curves for different noise schedules.**   We investigate different parameterizations of the noise schedule in Fig. 2b. Among polynomial, linear, and monotonic neural network, we find that the polynomial parameterization yields the best performance. The polynomial noise schedule is a novel component introduced in our work. The reason why a polynomial function works better than a linear or a monotonic neural network as proposed by VDM is rooted in Occam's razor. In Suppl. E.2, we show that a degree 5 polynomial is the simplest polynomial that satisfies several desirable properties, including monotonicity and having a derivative that equals zero exactly twice. More expressive models (e.g., monotonic 3-layer MLPs) are more difficult to optimize.

**Examining the noise schedule.**   Since the noise schedule, $\boldsymbol{\gamma}_\phi(\mathbf{z}, t)$ is multivariate, we expect to learn different noise schedules for different input dimensions and different inputs $\mathbf{z} \sim p_\theta(\mathbf{z})$. In Fig. 3, we take our best trained model on CIFAR-10 and visualize the variance of the noise schedule at each point in time for different pixels, where the variance is taken on 128 samples $\mathbf{z} \sim p_\theta(\mathbf{z})$.

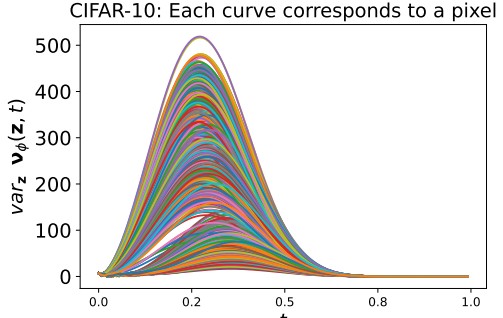

Figure 3: Noise schedule visualizations for MuLAN on CIFAR-10. In this figure, we plot the variance of $\boldsymbol{\nu}_\phi(\mathbf{z}, t)$ across different $\mathbf{z} \sim p_\theta(\mathbf{z})$ where each curve represents the SNR corresponding to an input dimension.

We note an increased variation in the early portions of the noise schedule. However, on an absolute scale, the variance of this noise is smaller than we expected. We also tried to visualize noise schedules across different dataset images and across different areas of the same image; refer to Fig. 13. We also generated synthetic datasets in which each datapoint contained only high frequencies or only low frequencies, and with random masking applied to parts of the data points; see Suppl. H. Surprisingly, none of these experiments revealed human-interpretable patterns in the learned schedule, although we did observe clear differences in likelihood estimation. We hypothesize that other architectures and other forms of conditioning may reveal interpretable patterns of variation; however, we leave this exploration to future work.

**Replacing the noise schedules in a trained denoising model.** We also confirm experimentally our claim that the learning objective is not invariant to the multivariate noise schedule. We replace the noise schedule in the trained denoising model with two alternatives: MULAN with scalar noise schedule, and a linear noise schedule: $\boldsymbol{\gamma}_\phi(\mathbf{z}, t) = \gamma_{\min} + t(\gamma_{\max} - \gamma_{\min})\mathbf{1_d}$; see Kingma et al. [20]. For both the noise schedules the likelihood reduces to the same value as that of the VDM: 2.65.

## 5 Related Work

Diffusion models have emerged in recent years as powerful tools for modeling complex distributions [51, 16], extending flow-based methods [53, 24, 48, 49] The noise schedule, which determines the amount and type of noise added at each step, plays a critical role in diffusion models. Chen [3] empirically demonstrate that different noise schedules can significantly impact the generated image quality using various handcrafted noise schedules. Kingma et al. [20] showed that the likelihood of a diffusion model remains invariant to the noise schedule with a scalar noise schedule. In this work we show that the ELBO is no longer invariant to multivariate noise schedules.

Recent works explored multivariate noise schedules (including blurring, masking, etc.) [17, 42, 36, 12], yet none have delved into learning the noise schedule conditioned on the input data itself. Likewise, conditional noise processes are typically not learned [26, 39, 62] and their conditioner (e.g., a prompt) is always available. Auxiliary variable models [63, 60] add semantic latents in $p$, but not in $q$, and they don't condition or learn $q$. In contrast, we learn multivariate noise conditioned on latent context.

Diffusion normalizing flows (DNFs) [64] learn a $q$ parameterized by a normalizing flow; however, such $q$ do not admit tractable marginals and require sampling full data-to-noise trajectories from $q$, which is expensive. Concurrent work on neural diffusion models (NDMs) and DiffEnc admits tractable marginals $q$ with learned means and univariate schedules; this yields more expressive $q$ than ours but requires computing losses in a modified space that precludes using a noise parameterization and certain sampling strategies. Empirically, MuLAN performs better with fewer parameters (Suppl. A).

Optimal transport techniques seek to learn a noise process that minimizes the transport cost from data to noise, which in practice produces smoother diffusion trajectories that facilitate sampling. Schrondinger bridges [47, 6, 59, 37] learn expressive $q$ do not admit analytical marginals, require computing full data-to-noise trajectories and involve iterative optimization (e.g., sinkhorn), which can be slow. Rectification [27] seeks diffusion paths that are close to linear; this improves sampling, while our method chooses paths that improve log-likelihood. See Suppl. A for more detailed comparisons.

## 6 Conclusion

We introduced MULAN, a context-adaptive noise process that applies Gaussian noise at varying rates across input data. Our theory challenges the prevailing notion that the likelihood of diffusion models is independent of the noise schedule: this independence only holds true for univariate schedules. Our evaluation of MULAN spans multiple image datasets, where it outperforms state-of-the-art generative diffusion models. We hope our work will motivate further research into the design of noise schedules, not only for improving likelihood estimation but also to improve image quality generation [35, 53]. A stronger fit to the data distribution also holds promise for improving downstream applications of generative modeling, e.g., compression or decision-making [32, 9, 8, 41].

## Acknowledgments and Disclosure of Funding

This work was partially funded by Volodymyr Kuleshov's the National Science Foundation under awards DGE-1922551, CAREER awards 2046760 and 2145577, and the National Institute of Health under award MIRA R35GM151243, and by Christopher De Sa's NSF RI-CAREER award 2046760.

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

# Contents

# Appendices

## Appendix A   Comparing to Previous Work

MᴜLAN is the first method to introduce a learned adaptive noise process. A widely held assumption is that the ELBO objective of a diffusion model is invariant to the noise process [20]. We dispel this assumption: we show that when input-conditioned noise is combined with (a) multivariate noise, (b) a novel polynomial parameterization, and (c) auxiliary variables, a learned noise process yields an improved variational posterior and a tighter ELBO. This approach sets a new state-of-the-art in density estimation. While (a), (c) were proposed in other contexts, we leverage them as subcomponents of a novel algorithm. We elaborate further on this below.

## A.1 Diffusion Models with Custom Noise

The noise schedule, which determines the amount and type of noise added at each step, plays a critical role in diffusion models. Chen [3] empirically demonstrate that different noise schedules can significantly impact the generated image quality using various handcrafted noise schedules. Kingma et al. [20] showed that the likelihood of a diffusion model remains invariant to the noise schedule with a scalar noise schedule. In this work we show that the ELBO is no longer invariant to multivariate noise schedules.

Recent works, including Hoogeboom & Salimans [17], Rissanen et al. [42], Pearl et al. [36], have explored per-pixel noise schedules (including blurring and other types of noising), yet none have delved into learning or conditioning the noise schedule on the input data itself. The shared components among these models are summarized and compared in Table 4.

## A.2 Advanced Diffusion Models

Yang et al. [62] proposes noise processes that are conditioned on an external context (e.g., text). We also propose context-conditioned noise processes; however, their setting is that of conditional generation, where the context is always available at training and inference time, and the context represents external data. Our paper instead looks at unconditional generation, and we condition the noising process on the image itself that we want to generate (via latent variable) and learn how to apply noise across an image as a function of the image.

Lee et al. [26], Popov et al. [39] proposed using a data-dependent prior: however, they do not learn q and their noise process is not adaptive to the input $\mathbf{x}_0$. Thus they propose a fairly different set of methods from what we introduce.

Yang & Mandt [63], Wang et al. [60] have explored diffusion models with an auxiliary latent space, where the denoising network is conditioned on a latent distribution. Our paper also incorporate auxiliary latents, but unlike previous works, we add them to both $p$ and $q$ and we also also focus on learning the process $q$ (as opposed to doing representation learning using the auxiliary learned space). Lastly, our algorithm relies on many other components including a custom noise schedule, multivariate noise, etc. The shared components among these models are summarized and compared in Table 4.

Table 4: MULAN is a noise schedule that can be integrated into any diffusion model such as VDM [20] or InfoDiffusion [60]. The shared components between MULAN and these models are summarized and compared in this table.

| Method | learned noise | multivariate noise | input conditioned noise | auxiliary latents | noise schedule |
|---|---|---|---|---|---|
| VDM [20] | **Yes** | No | No | No | Monotonic neural network |
| Blurring Diffusion Model [17] | No | **Yes** | No | No | Frequency scaling |
| Inverse Heat Dissipation [42] | No | **Yes** | No | No | Exponential |
| SVNR [36] | No | **Yes** | No | No | Linear |
| InfoDiffusion [60] | No | No | No | In denoising process | Cosine |
| Lossy Compression [63] | No | No | No | In denoising process | Linear / Cosine |
| MULAN (**Ours**) | **Yes** | **Yes** | **Yes** | In **noising** and **denoising** process | Polynomial |

## A.3   Learned Diffusion

Diffusion Normalizing Flow (DNF) uses the following forward process:

$$\mathrm{d}\mathbf{x}_t = \mathbf{f}_\theta(\mathbf{x}_t, t)\mathrm{d}t + g(t)\mathrm{d}\mathbf{w}, \tag{13}$$

where the drift term $\mathbf{f}_\theta : \mathbb{R}^d \times \mathbb{R} \to \mathbb{R}^d$ is parameterized by a neural network with parameters $\theta$ and the diffusion term $g(t) \in \mathbb{R}^+$ is a scalar constant and $\mathbf{w}$ is the standard Brownian motion. However, in MuLAN, the forward process is given by

$$\mathrm{d}\mathbf{x}_t = \mathbf{f}_\theta(\mathbf{z}, t) \odot \mathbf{x}_t \mathrm{d}t + \mathbf{g}_\theta(\mathbf{z}, t) \odot \mathrm{d}\mathbf{w}; \mathbf{z} \sim q_\phi(\mathbf{z}|\mathbf{x}_0), \tag{14}$$

where $\mathbf{z} \in \{0, 1\}^m$ is the auxiliary latent vector, $\mathbf{f}_\theta : \mathbb{R}^m \times \mathbb{R} \to \mathbb{R}^d$ and $\mathbf{g}_\theta : \mathbb{R}^m \times \mathbb{R} \to \mathbb{R}^d$ are parameterized by a neural network. Notice that the drift term in DNF, $\mathbf{f}_\theta(\mathbf{x}, t)$, is a non-linear function in $\mathbf{x}_0$, and the same holds for MuLAN since in the drift term, $\mathbf{f}_\theta(\mathbf{z}, t) \odot \mathbf{x}$, $\mathbf{z}$ and $\mathbf{x}$ depend on $\mathbf{x}_0$. Additionally, the diffusion coefficient, $\mathbf{g}_\theta(\mathbf{z}, t)$, is multivariate and conditioned on $\mathbf{x}_0$ via $\mathbf{z}$. The two parameterizations are different: on one hand, DNF admits more general classes of neural networks because it does not require marginals to be tractable. On the other hand MuLAN admits a more flexible noise model $\mathbf{g}_\theta(\mathbf{z}, t)$ and admits more efficient training (see the summarized Table 5 below).

MuLAN has the advantage that it is simulation free; which means that given a data $\mathbf{x}_0$, the noisy sample $\mathbf{x}_t$ can be computed in closed form; however, in Diffusion Normalizing Flow, to compute $\mathbf{x}_t$, one needs to simulate the forward SDE which is resource intensive and limits its scalability to larger denoising models. While MuLAN optimizes the ELBO, DNF optimizes an approximation for the ELBO. In particular, the DNF training objective does not involve a term that accounts for the entropy of the encoder. Thus, the objective is closer to that of a normal auto-encoder in that regard.

Table 5: The key differences between MULAN and DNF is listed below.

| Aspect | Property | Diffusion Normalizing Flow | MULAN |
|---|---|---|---|
| Drift Term | Multivariate | Yes | Yes |
| | Adaptive | Yes | Yes |
| | Learnable | Yes | Yes |
| Diffusion Term | Multivariate | No | **Yes** |
| | Adaptive | No | **Yes** |
| | Learnable | No | **Yes** |
| Simulation Free Training | | No | **Yes** |
| Exact ELBO Optimization | | No | **Yes** |
| NLL ($\downarrow$) | CIFAR-10 | 3.04 | **2.55** |

Other concurrent work seeks to improve log-likelihood estimation by learning the forward diffusion process in a simulation-free setting. In neural diffusion models (NDMs), the noise schedule is fixed, but the mean of each marginal $q(\mathbf{x}_t|\mathbf{x}_0)$ is learned. This requires denoising $\mathbf{x}$ in a transformed space, which prevents using noise parameterization, a design choice that is important for performance. Their denoising family also induces a parameterization that limits the kinds of samplers that can be sued. Lastly, NDMs use a model that is 2x larger than a regular diffusion model, while ours only adds 10% more parameters.

The DiffEnc framework adds an extra learned correction term to $q$ to adjust the mean of each marginal $q(\mathbf{x}_t|\mathbf{x}_0)$. This noise choice also requires using certain parameterizations for $\mathbf{x}$ that are not compatible with noise parameterization; while their approach supports v-parameterization, it also requires training a mean parameterization network. Similarly to NDMs, the noise schedule remains fixed, while the mean of each marginal is adjusted by the network. Our approach towards learning the noise schedule yields better empirical performance and is, in our opinion, simpler; it can also be combined with this prior work on learning the marginals' means.

## A.4 Optimal Transport

In techniques based on optimal transport, the goal is to learn a noise process that minimizes the transport cost from data to noise, which in practice produces smoother diffusion trajectories that facilitate sampling.

Minimizing Trajectory Curvature (MTC) of ODE-based generative models Lee et al. [27], the primary goal is to design the forward diffusion process that is optimal for fast sampling; however, MuLAN strives to learn a forward process that optimizes for log-likelihood. In the former, the marginals $\mathbf{x}_t$ in the forward process are given as

$$\mathbf{x}_t = (1 - t)\mathbf{x}_0 + t\mathbf{z}; \mathbf{z} \sim q_\phi(\mathbf{z}|\mathbf{x}_0) \tag{15}$$

where $\mathbf{x}_t, \mathbf{x}_0, F \in \mathbb{R}^d$. However for MuLAN the marginals are $\mathbf{x}_t = \boldsymbol{\alpha}_\phi(\mathbf{z}, t) \odot \mathbf{x}_0 + \sqrt{1 - \boldsymbol{\alpha}_\phi^2(\mathbf{z}, t)} \odot \epsilon$ ; $\epsilon \sim \mathcal{N}(0, \mathbf{I}_d)$ ; $\mathbf{z} \sim q_\phi(\mathbf{z}|\mathbf{x}_0)$ where $\boldsymbol{\alpha}_\phi(\mathbf{z}, t) : \mathbb{R}^d \times \mathbb{R} \to \mathbb{R}^d_{\geq 0}$ , $\mathbf{z} \in \{0, 1\}^m$ , $\epsilon \in \mathbb{R}^d$ Notice that in the MTC formula, the coefficient of $\mathbf{x}_0$ , the time integral of the drift term, is a scalar and linear function of, and is independent of the input $\mathbf{x}_0$. In MuLAN, that term is a multivariate non-linear function in $t$, and conditioned on $\mathbf{x}_0$ via the auxiliary latent variable $\mathbf{z}$. This implies that the forward diffusion process in MuLAN is more expressive than MTC. The simplistic forward process in MTC enables faster sampling whereas the richer / more expressive forward process in MuLAN leads to improved likelihood estimates. Table 6 summarizes the key differences.

Table 6: Comparison between Minimizing Trajectory Curvature and MuLAN methods.

| Aspect | Property | Minimizing Trajectory Curvature | MuLAN |
|---|---|---|---|
| Goal | | Design faster sampler | Improve log-likelihood |
| Drift Term | Learnable | No | **Yes** |
| | Linearity | Linear in time $t$, linear in $\mathbf{x}_0$ | **Non-linear** in time $t$, **Non-linear** in $\mathbf{z}$ (and hence $\mathbf{x}_0$) |
| | Dimensionality | Scalar | **Multivariate** |
| | Adaptive | No | **Yes** |
| Diffusion Term | Linearity | Linear in time $t$ | **Non-linear** in time $t$ |
| | Dimensionality | Multivariate | Multivariate |
| | Learnable | Yes | Yes |
| | Adaptive | Yes | Yes |

An alternative approach to learning a forward process that performs optimal transport is via the theory of Schrodinger bridges [47, 6, 59, 37] . Similarly to the DNF framework, these methods do not admit analytical marginals and therefore involve computing full trajectories from noisy and clean data. Additionally, they are typically trained using an iterative procedure that generalizes the sinkhorn algorithm and involves iteratively training $q$ and $p$. As such, these types of methods are typically more expensive to train and competitive results on standard benchmarks (e.g., CIFAR10, ImageNet) are not yet available to our knowledge.

# Appendix B   Standard Diffusion Models

We have a Gaussian diffusion process that begins with the data $\mathbf{x}_0$, and defines a sequence of increasingly noisy versions of $\mathbf{x}_0$ which we call the latent variables $\mathbf{x}_t$, where $t$ runs from $t = 0$ (least noisy) to $t = 1$ (most noisy). Given, $T$, we discretize time uniformly into $T$ timesteps each with a width $1/T$. We define $t(i) = i/T$ and $s(i) = (i - 1)/T$.

## B.1   Forward Process

$$q(\mathbf{x}_t|\mathbf{x}_s) = \mathcal{N}(\alpha_{t|s}\mathbf{x}_s, \sigma_{t|s}^2\mathbf{I}_n) \tag{16}$$

where

$$\alpha_{t|s} = \frac{\alpha_t}{\alpha_s} \tag{17}$$

$$\sigma_{t|s}^2 = \sigma_t^2 - \frac{\alpha_{t|s}^2}{\sigma_s^2} \tag{18}$$

## B.2   Reverse Process

Kingma et al. [20] show that the distribution $q(\mathbf{x}_s|\mathbf{x}_t, \mathbf{x}_0)$ is also gaussian,

$$q(\mathbf{x}_s|\mathbf{x}_t, \mathbf{x}_0) = \mathcal{N}\left(\boldsymbol{\mu}_q = \frac{\alpha_{t|s}\sigma_s^2}{\sigma_t^2}\mathbf{x}_t + \frac{\sigma_{t|s}^2\alpha_s}{\sigma_t^2}\mathbf{x}_0, \ \boldsymbol{\Sigma}_q = \frac{\sigma_s^2\sigma_{t|s}^2}{\sigma_t^2}\mathbf{I}_n\right) \tag{19}$$

Since during the reverse process, we don't have access to $\mathbf{x}_0$, we approximate it using a neural network $\mathbf{x}_\theta(\mathbf{x}_t, t)$ with parameters $\theta$. Thus,

$$p_\theta(\mathbf{x}_s|\mathbf{x}_t) = \mathcal{N}\left(\boldsymbol{\mu}_p = \frac{\alpha_{t|s}\sigma_s^2}{\sigma_t^2}\mathbf{x}_t + \frac{\sigma_{t|s}^2\alpha_s}{\sigma_t^2}\mathbf{x}_\theta(\mathbf{x}_t, t), \ \boldsymbol{\Sigma}_p = \frac{\sigma_s^2\sigma_{t|s}^2}{\sigma_t^2}\mathbf{I}_n\right) \tag{20}$$

## B.3 Variational Lower Bound

This corruption process $q$ is the following markov-chain as $q(\mathbf{x}_{0:1}) = q(\mathbf{x}_0) \left( \prod_{i=1}^{T} q(\mathbf{x}_{t(i)}|\mathbf{x}_{s(i)}) \right)$. In the Reverse Process, or the Denoising Process, $p_\theta$, a neural network (with parameters $\theta$) is used to denoise the noising process $q$. The Reverse Process factorizes as: $p_\theta(\mathbf{x}_{0:1}) = p_\theta(\mathbf{x}_1) \prod_{i=1}^{T} p_\theta(\mathbf{x}_{s(i)}|\mathbf{x}_{t(i)})$. Let $\mathbf{x}_\theta(\mathbf{x}_t, t)$ be the reconstructed input by a neural network from $\mathbf{x}_t$. Similar to Sohl-Dickstein et al. [51], Kingma et al. [20] we decompose the negative lower bound (VLB) as:

$$
-\log p_\theta(\mathbf{x}_0) \leq \underbrace{\mathbb{E}_{q_\phi} \left[ -\log \frac{p_\theta(\mathbf{x}_{t(0):t(T)})}{q(\mathbf{x}_{t(1):t(T)}|\mathbf{x}_0)} \right]}_{\text{ELBO}(p_\theta(\mathbf{x}_{0:1}), q(\mathbf{x}_{0:1})) \text{defined in Eq. } 1}
$$

$$
= \mathbb{E}_{\mathbf{x}_{t(1)} \sim q(\mathbf{x}_{t(1)}|\mathbf{x}_0)} [-\log p_\theta(\mathbf{x}_0|\mathbf{x}_{t(1)})]
$$

$$
+ \sum_{i=2}^{T} \mathbb{E}_{\mathbf{x}_{t(i)} \sim q(\mathbf{x}_{t(i)}|\mathbf{x}_0)} D_{\text{KL}}[p_\theta(\mathbf{x}_{s(i)}|\mathbf{x}_{t(i)}) \| q(\mathbf{x}_{s(i)}|\mathbf{x}_{t(i)}, \mathbf{x}_0)]
$$

$$
+ D_{\text{KL}}[p_\theta(\mathbf{x}_1) \| q_\phi(\mathbf{x}_1|\mathbf{x}_0)]
$$

$$
= \underbrace{\mathbb{E}_{\mathbf{x}_{t(1)} \sim q(\mathbf{x}_{t(1)}|\mathbf{x}_0)} [-\log p_\theta(\mathbf{x}_0|\mathbf{x}_{t(1)})]}_{\mathcal{L}_{\text{recons}}}
$$

$$
+ \underbrace{\frac{T}{2} \mathbb{E}_{\epsilon \sim \mathcal{N}(0, \mathbf{I}_n), i \sim U\{2,T\}} D_{\text{KL}}[p_\theta(\mathbf{x}_{s(i)}|\mathbf{x}_{t(i)}) \| q(\mathbf{x}_{s(i)}|\mathbf{x}_{t(i)}, \mathbf{x}_0)]}_{\mathcal{L}_{\text{diffusion}}}
$$

$$
+ \underbrace{D_{\text{KL}}[p_\theta(\mathbf{x}_1) \| q(\mathbf{x}_1|\mathbf{x}_0)]}_{\mathcal{L}_{\text{prior}}} \tag{21}
$$

The prior loss, $\mathcal{L}_{\text{prior}}$, and reconstruction loss, $\mathcal{L}_{\text{recons}}$, can be (stochastically and differentiably) estimated using standard techniques; see Kingma & Welling [23]. The diffusion loss, $\mathcal{L}_{\text{diffusion}}$, varies with the formulation of the noise schedule. We provide an exact formulation for it in the subsequent sections.

## B.4 Diffusion Loss

For brevity, we use the notation $s$ for $s(i)$ and $t$ for $t(i)$. From Eq. 31 and Eq. 32 we get the following expression for $q(\mathbf{x}_s|\mathbf{x}_t, \mathbf{x}_0)$:

$$
D_{\text{KL}}(q(\mathbf{x}_s|\mathbf{x}_t, \mathbf{x}_0) \| p_\theta(\mathbf{x}_s|\mathbf{x}_t))
$$

$$
= \frac{1}{2} \left( (\boldsymbol{\mu}_q - \boldsymbol{\mu}_p)^\top \boldsymbol{\Sigma}_\theta^{-1} (\boldsymbol{\mu}_q - \boldsymbol{\mu}_p) + \text{tr} \left( \boldsymbol{\Sigma}_q \boldsymbol{\Sigma}_p^{-1} - \mathbf{I}_n \right) - \log \frac{|\boldsymbol{\Sigma}_q|}{|\boldsymbol{\Sigma}_p|} \right)
$$

$$
= \frac{1}{2} (\boldsymbol{\mu}_q - \boldsymbol{\mu}_p)^\top \Sigma_\theta^{-1} (\boldsymbol{\mu}_q - \boldsymbol{\mu}_p)
$$

Substituting $\boldsymbol{\mu}_q, \boldsymbol{\Sigma}_q, \boldsymbol{\mu}_p, \boldsymbol{\Sigma}_p$ from equation 20 and equation 19; for the exact derivation see Kingma et al. [20]

$$
= \frac{1}{2} \left( \nu(s) - \nu(t) \right) \| (\mathbf{x}_0 - \mathbf{x}_\theta(\mathbf{x}_t, t)) \|_2^2 \tag{22}
$$

Thus $\mathcal{L}_{\text{diffusion}}$ is given by

$$
\mathcal{L}_{\text{diffusion}}
$$

$$
= \lim_{T \to \infty} \frac{T}{2} \mathbb{E}_{\epsilon \sim \mathcal{N}(0, \mathbf{I}_n), i \sim U\{2,T\}} D_{\text{KL}}[p_\theta(\mathbf{x}_{s(i)}|\mathbf{x}_{t(i)}) \| q_\phi(\mathbf{x}_{s(i)}|\mathbf{x}_{t(i)}, \mathbf{x}_0)]
$$

$$
= \lim_{T \to \infty} \frac{1}{2} \sum_{i=2}^{T} \mathbb{E}_{\epsilon \sim \mathcal{N}(0, \mathbf{I}_n)} \left( \nu(s) - \nu(t) \right) \| \mathbf{x}_0 - \mathbf{x}_\theta(\mathbf{x}_t, t) \|_2^2
$$

$$
= \frac{1}{2} \mathbb{E}_{\epsilon \sim \mathcal{N}(0, \mathbf{I}_n)} \left[ \lim_{T \to \infty} \sum_{i=2}^{T} \left( \nu(s) - \nu(t) \right) \| \mathbf{x}_0 - \mathbf{x}_\theta(\mathbf{x}_t, t) \|_2^2 \right]
$$

$$= \frac{1}{2} \mathbb{E}_{\epsilon \sim \mathcal{N}(0,\mathbf{I}_n)} \left[ \lim_{T \to \infty} \sum_{i=2}^{T} T \left( \nu(s) - \nu(t) \right) \| \mathbf{x}_0 - \mathbf{x}_\theta(\mathbf{x}_t, t) \|_2^2 \frac{1}{T} \right]$$

Substituting $\lim_{T \to \infty} T(\nu(s) - \nu(t)) = \frac{\mathrm{d}}{\mathrm{d}t} \nu(t) \equiv \nu'(t)$; see Kingma et al. [20]

$$= \frac{1}{2} \mathbb{E}_{\epsilon \sim \mathcal{N}(0,\mathbf{I}_n)} \left[ \int_0^1 \nu'(t) \| \mathbf{x}_0 - \mathbf{x}_\theta(\mathbf{x}_t, t) \|_2^2 \right] \mathrm{d}t \tag{23}$$

In practice instead of computing the integral is computed by MC sampling.

$$= -\frac{1}{2} \mathbb{E}_{\epsilon \sim \mathcal{N}(0,\mathbf{I}_n), t \sim U[0,1]} \left[ \nu'(t) \| \mathbf{x}_0 - \mathbf{x}_\theta(\mathbf{x}_t, t) \|_2^2 \right] \tag{24}$$

## Appendix C   Multivariate noise schedule

For a multivariate noise schedule we have $\boldsymbol{\alpha}_t, \boldsymbol{\sigma}_t \in \mathbb{R}_+^d$ where $t \in [0, 1]$. $\boldsymbol{\alpha}_t, \boldsymbol{\sigma}_t$ are vectors. The timesteps $s, t$ satisfy $0 \le s < t \le 1$. Furthermore, we use the following notations where arithmetic division represents element wise division between 2 vectors:

$$\boldsymbol{\alpha}_{t|s} = \frac{\boldsymbol{\alpha}_t}{\boldsymbol{\alpha}_s} \tag{25}$$

$$\boldsymbol{\sigma}_{t|s}^2 = \boldsymbol{\sigma}_t^2 - \frac{\boldsymbol{\alpha}_{t|s}^2}{\boldsymbol{\sigma}_s^2} \tag{26}$$

### C.1   Forward Process

$$q(\mathbf{x}_t | \mathbf{x}_s) = \mathcal{N} \left( \boldsymbol{\alpha}_{t|s} \mathbf{x}_s, \boldsymbol{\sigma}_{t|s}^2 \right) \tag{27}$$

**Change of variables.**   We can write $\mathbf{x}_t$ explicitly in terms of the signal-to-noise ratio, $\boldsymbol{\nu}(t)$, and input $\mathbf{x}_0$ in the following manner:

$$\boldsymbol{\nu}_t = \frac{\boldsymbol{\alpha}_t^2}{\boldsymbol{\sigma}_t^2}$$

We know $\alpha_t^2 = 1 - \sigma_t^2$ for Variance Preserving process; see Sec. 2.

$$\implies \frac{1 - \boldsymbol{\sigma}_t^2}{\boldsymbol{\sigma}_t^2} = \boldsymbol{\nu}_t$$

$$\implies \boldsymbol{\sigma}_t^2 = \frac{1}{1 + \boldsymbol{\nu}_t} \quad \text{and} \quad \boldsymbol{\alpha}_t^2 = \frac{\boldsymbol{\nu}_t}{1 + \boldsymbol{\nu}_t} \tag{28}$$

$$\nu_t = \frac{\alpha_t^2}{\sigma_t^2}$$

We know $\alpha_t^2 = 1 - \sigma_t^2$ for Variance Preserving process; see Sec. 2.

$$\implies \frac{1 - \sigma_t^2}{\sigma_t^2} = \nu_t$$

$$\implies \sigma_t^2 = \frac{1}{1 + \nu_t} \quad \text{and} \quad \alpha_t^2 = \frac{\nu_t}{1 + \nu_t} \tag{29}$$

Thus, we write $\mathbf{x}_t$ in terms of the signal-to-noise ratio in the following manner:

$$\mathbf{x}_{\boldsymbol{\nu}(t)} = \boldsymbol{\alpha}_t \mathbf{x}_0 + \boldsymbol{\sigma}_t \boldsymbol{\epsilon}_t; \ \boldsymbol{\epsilon}_t \sim \mathcal{N}(0, \mathbf{I}_n)$$

$$= \frac{\sqrt{\boldsymbol{\nu}(t)}}{\sqrt{1 + \boldsymbol{\nu}(t)}} \mathbf{x}_0 + \frac{1}{\sqrt{1 + \boldsymbol{\nu}(t)}} \boldsymbol{\epsilon}_t \qquad \text{Using Eq. 28} \tag{30}$$

## C.2 Reverse Process

The distribution of $\mathbf{x}_t$ given $\mathbf{x}_s$ is given by:

$$q(\mathbf{x}_s|\mathbf{x}_t, \mathbf{x}_0) = \mathcal{N}\left(\boldsymbol{\mu}_q = \frac{\boldsymbol{\alpha}_{t|s}\boldsymbol{\sigma}_s^2}{\boldsymbol{\sigma}_t^2}\mathbf{x}_t + \frac{\boldsymbol{\sigma}_{t|s}^2\boldsymbol{\alpha}_s}{\boldsymbol{\sigma}_t^2}\mathbf{x}_0, \; \boldsymbol{\Sigma}_q = \mathrm{diag}\left(\frac{\boldsymbol{\sigma}_s^2\boldsymbol{\sigma}_{t|s}^2}{\boldsymbol{\sigma}_t^2}\right)\right) \tag{31}$$

Let $\mathbf{x}_\theta(\mathbf{x}_t, t)$ be the neural network approximation for $\mathbf{x}_0$. Then we get the following reverse process:

$$p_\theta(\mathbf{x}_s|\mathbf{x}_t) \sim \mathcal{N}\left(\boldsymbol{\mu}_p = \frac{\boldsymbol{\alpha}_{t|s}\boldsymbol{\sigma}_s^2}{\boldsymbol{\sigma}_t^2}\mathbf{x}_t + \frac{\boldsymbol{\sigma}_{t|s}^2\boldsymbol{\alpha}_s}{\boldsymbol{\sigma}_t^2}\mathbf{x}_\theta(\mathbf{x}_t, t), \; \boldsymbol{\Sigma}_p = \mathrm{diag}\left(\frac{\boldsymbol{\sigma}_s^2\boldsymbol{\sigma}_{t|s}^2}{\boldsymbol{\sigma}_t^2}\right)\right) \tag{32}$$

## C.3 Diffusion Loss

For brevity we use the notation $s$ for $s(i)$ and $t$ for $t(i)$. From Eq. 31 and Eq. 32 we get the following expression for $q(\mathbf{x}_s|\mathbf{x}_t, \mathbf{x}_0)$:

$$D_{\mathrm{KL}}(q(\mathbf{x}_s|\mathbf{x}_t, \mathbf{x}_0)\|p_\theta(\mathbf{x}_s|\mathbf{x}_t))$$

$$= \frac{1}{2}\left((\boldsymbol{\mu}_q - \boldsymbol{\mu}_p)^\top\boldsymbol{\Sigma}_\theta^{-1}(\boldsymbol{\mu}_q - \boldsymbol{\mu}_p) + \mathrm{tr}\left(\boldsymbol{\Sigma}_q\boldsymbol{\Sigma}_p^{-1} - \mathbf{I}_n\right) - \log\frac{|\boldsymbol{\Sigma}_q|}{|\boldsymbol{\Sigma}_p|}\right)$$

$$= \frac{1}{2}(\boldsymbol{\mu}_q - \boldsymbol{\mu}_p)^\top\boldsymbol{\Sigma}_\theta^{-1}(\boldsymbol{\mu}_q - \boldsymbol{\mu}_p)$$

Substituting $\boldsymbol{\mu}_q, \boldsymbol{\mu}_p, \boldsymbol{\Sigma}_p$ from equation 32 and equation 31.

$$= \frac{1}{2}\left(\frac{\boldsymbol{\sigma}_{t|s}^2\boldsymbol{\alpha}_s}{\boldsymbol{\sigma}_t^2}\mathbf{x}_0 - \frac{\boldsymbol{\sigma}_{t|s}^2\boldsymbol{\alpha}_s}{\boldsymbol{\sigma}_t^2}\mathbf{x}_\theta(\mathbf{x}_t, t)\right)^\top \mathrm{diag}\left(\frac{\boldsymbol{\sigma}_s^2\boldsymbol{\sigma}_{t|s}^2}{\boldsymbol{\sigma}_t^2}\right)^{-1}\left(\frac{\boldsymbol{\sigma}_{t|s}^2\boldsymbol{\alpha}_s}{\boldsymbol{\sigma}_t^2}\mathbf{x}_0 - \frac{\boldsymbol{\sigma}_{t|s}^2\boldsymbol{\alpha}_s}{\boldsymbol{\sigma}_t^2}\mathbf{x}_\theta(\mathbf{x}_t, t)\right)$$

$$= \frac{1}{2}(\mathbf{x}_0 - \mathbf{x}_\theta(\mathbf{x}_t, t))^\top \mathrm{diag}\left(\frac{\boldsymbol{\sigma}_{t|s}^2\boldsymbol{\alpha}_s}{\boldsymbol{\sigma}_t^2}\right)^\top \mathrm{diag}\left(\frac{\boldsymbol{\sigma}_s^2\boldsymbol{\sigma}_{t|s}^2}{\boldsymbol{\sigma}_t^2}\right)^{-1}\mathrm{diag}\left(\frac{\boldsymbol{\sigma}_{t|s}^2\boldsymbol{\alpha}_s}{\boldsymbol{\sigma}_t^2}\right)(\mathbf{x}_0 - \mathbf{x}_\theta(\mathbf{x}_t, t))$$

$$= \frac{1}{2}(\mathbf{x}_0 - \mathbf{x}_\theta(\mathbf{x}_t, t))^\top \mathrm{diag}\left(\frac{\boldsymbol{\sigma}_{t|s}^2\boldsymbol{\alpha}_s}{\boldsymbol{\sigma}_t^2} \odot \frac{\boldsymbol{\sigma}_t^2}{\boldsymbol{\sigma}_s^2\boldsymbol{\sigma}_{t|s}^2} \odot \frac{\boldsymbol{\sigma}_{t|s}^2\boldsymbol{\alpha}_s}{\boldsymbol{\sigma}_t^2}\right)(\mathbf{x}_0 - \mathbf{x}_\theta(\mathbf{x}_t, t))$$

$$= \frac{1}{2}(\mathbf{x}_0 - \mathbf{x}_\theta(\mathbf{x}_t, t))^\top \mathrm{diag}\left(\frac{\boldsymbol{\sigma}_{t|s}^2\boldsymbol{\alpha}_s^2}{\boldsymbol{\sigma}_t^2\boldsymbol{\sigma}_s^2}\right)(\mathbf{x}_0 - \mathbf{x}_\theta(\mathbf{x}_t, t))$$

Simplifying the expression using Eq. 25 and Eq. 26 we get,

$$= \frac{1}{2}(\mathbf{x}_0 - \mathbf{x}_\theta(\mathbf{x}_t, t))^\top \mathrm{diag}\left(\frac{\boldsymbol{\alpha}_s^2}{\boldsymbol{\sigma}_s^2} - \frac{\boldsymbol{\alpha}_t^2}{\boldsymbol{\sigma}_t^2}\right)(\mathbf{x}_0 - \mathbf{x}_\theta(\mathbf{x}_t, t))$$

Using the relation $\boldsymbol{\nu}(t) = \boldsymbol{\alpha}_t^2/\boldsymbol{\sigma}_t^2$ we get,

$$= \frac{1}{2}(\mathbf{x}_0 - \mathbf{x}_\theta(\mathbf{x}_t, t))^\top \mathrm{diag}\left(\boldsymbol{\nu}(s) - \boldsymbol{\nu}(t)\right)(\mathbf{x}_0 - \mathbf{x}_\theta(\mathbf{x}_t, t)) \tag{33}$$

Like Kingma et al. [20] we train the model in the continuous domain with $T \to \infty$.

$$\mathcal{L}_{\mathrm{diffusion}}$$

$$= \lim_{T\to\infty}\frac{1}{2}\sum_{i=2}^T \mathbb{E}_{\epsilon\sim\mathcal{N}(0,\mathbf{I}_n)}D_{\mathrm{KL}}(q(\mathbf{x}_{s(i)}|\mathbf{x}_{t(i)}, \mathbf{x}_0)\|p_\theta(\mathbf{x}_{s(i)}|\mathbf{x}_{t(i)}))$$

$$= \lim_{T\to\infty}\frac{1}{2}\sum_{i=2}^T \mathbb{E}_{\epsilon\sim\mathcal{N}(0,\mathbf{I}_n)}(\mathbf{x}_0 - \mathbf{x}_\theta(\mathbf{x}_{t(i)}, t(i)))^\top \mathrm{diag}\left(\boldsymbol{\nu}_{s(i)} - \boldsymbol{\nu}_{t(i)}\right)(\mathbf{x}_0 - \mathbf{x}_\theta(\mathbf{x}_{t(i)}, t))$$

$$= \frac{1}{2}\mathbb{E}_{\epsilon\sim\mathcal{N}(0,\mathbf{I}_n)}\left[\lim_{T\to\infty}\sum_{i=2}^T (\mathbf{x}_0 - \mathbf{x}_\theta(\mathbf{x}_{t(i)}, t(i)))^\top \mathrm{diag}\left(\boldsymbol{\nu}_{s(i)} - \boldsymbol{\nu}_{t(i)}\right)(\mathbf{x}_0 - \mathbf{x}_\theta(\mathbf{x}_{t(i)}, t))\right]$$

$$= \frac{1}{2}\mathbb{E}_{\epsilon\sim\mathcal{N}(0,\mathbf{I}_n)}\left[\lim_{T\to\infty}\sum_{i=2}^T T(\mathbf{x}_0 - \mathbf{x}_\theta(\mathbf{x}_{t(i)}, t(i)))^\top \mathrm{diag}\left(\boldsymbol{\nu}_{s(i)} - \boldsymbol{\nu}_{t(i)}\right)(\mathbf{x}_0 - \mathbf{x}_\theta(\mathbf{x}_{t(i)}, t))\frac{1}{T}\right]$$

Let $\lim_{T\to\infty} T(\boldsymbol{\nu}_{s(i)} - \boldsymbol{\nu}_{t(i)}) = \frac{\mathrm{d}}{\mathrm{d}t}\boldsymbol{\nu}(t)$ denote the scalar derivative of the vector $\boldsymbol{\nu}(t)$ w.r.t $t$

$$= \frac{1}{2}\mathbb{E}_{\epsilon\sim\mathcal{N}(0,\mathbf{I}_n)}\left[\int_0^1 (\mathbf{x}_0 - \mathbf{x}_\theta(\mathbf{x}_t, t))^\top \mathrm{diag}\left(\frac{\mathrm{d}}{\mathrm{d}t}\boldsymbol{\nu}(t)\right)(\mathbf{x}_0 - \mathbf{x}_\theta(\mathbf{x}_t, t))\mathrm{d}t\right] \tag{34}$$

In practice instead of computing the integral is computed by MC sampling.

$$= -\frac{1}{2}\mathbb{E}_{\epsilon\sim\mathcal{N}(0,\mathbf{I}_n), t\sim U[0,1]}\left[(\mathbf{x}_0 - \mathbf{x}_\theta(\mathbf{x}_t, t))^\top \mathrm{diag}\left(\frac{\mathrm{d}}{\mathrm{d}t}\boldsymbol{\nu}(t)\right)(\mathbf{x}_0 - \mathbf{x}_\theta(\mathbf{x}_t, t))\right] \tag{35}$$

## C.4 Vectorized Representation of the diffusion loss

Let $\boldsymbol{\nu}(t)$ be the vectorized representation of the diagonal entries of the matrix $\boldsymbol{\nu}(t)$. We can rewrite the integral in 34 in the following vectorized form where $\odot$ denotes element wise multiplication and $\langle,\rangle$ denotes dot product between 2 vectors.

$$\mathcal{L}_{\text{diffusion}}$$

$$= -\frac{1}{2}\int_0^1 (\mathbf{x}_0 - \mathbf{x}_\theta(\mathbf{x}_t, t))^\top \mathrm{diag}\left(\frac{\mathrm{d}}{\mathrm{d}t}\boldsymbol{\nu}(t)\right)(\mathbf{x}_0 - \mathbf{x}_\theta(\mathbf{x}_t, t))\mathrm{d}t$$

$$= -\frac{1}{2}\int_0^1 \left\langle (\mathbf{x}_0 - \mathbf{x}_\theta(\mathbf{x}_t, t)) \odot (\mathbf{x}_0 - \mathbf{x}_\theta(\mathbf{x}_t, t)), \frac{\mathrm{d}}{\mathrm{d}t}\boldsymbol{\nu}(t)\right\rangle \mathrm{d}t$$

Using change of variables as mentioned in Sec. 3.2 we have

$$= -\frac{1}{2}\int_0^1 \left\langle (\mathbf{x}_0 - \tilde{\mathbf{x}}_\theta(\mathbf{x}_{\boldsymbol{\nu}(t)}, \boldsymbol{\nu}(t))) \odot (\mathbf{x}_0 - \tilde{\mathbf{x}}_\theta(\mathbf{x}_{\boldsymbol{\nu}(t)}, \boldsymbol{\nu}(t))), \frac{\mathrm{d}}{\mathrm{d}t}\boldsymbol{\nu}(t)\right\rangle \mathrm{d}t$$

Let $\mathbf{f}_\theta(\mathbf{x}_0, \boldsymbol{\nu}(t)) = (\mathbf{x}_0 - \tilde{\mathbf{x}}_\theta(\mathbf{x}_{\boldsymbol{\nu}(t)}, \boldsymbol{\nu}(t))) \odot (\mathbf{x}_0 - \tilde{\mathbf{x}}_\theta(\mathbf{x}_{\boldsymbol{\nu}(t)}, \boldsymbol{\nu}(t)))$

$$= \int_0^1 \left\langle \mathbf{f}_\theta(\mathbf{x}_0, \boldsymbol{\nu}(t)), \frac{\mathrm{d}}{\mathrm{d}t}\boldsymbol{\nu}(t)\right\rangle \mathrm{d}t \tag{36}$$

Thus $\mathcal{L}_{\text{diffusion}}$ can be interpreted as the amount of work done along the trajectory $\boldsymbol{\nu}(0) \to \boldsymbol{\nu}(1)$ in the presence of a vector field $\mathbf{f}_\theta(\mathbf{x}_0, \boldsymbol{\nu}(\mathbf{z}, t))$. From the perspective of thermodynamics, this is precisely equal to the amount of heat lost into the environment during the process of transition between 2 equilibria via the noise schedule specified by $\boldsymbol{\nu}(t)$.

## C.5 Log likelihood and Noise Schedules: A Thermodynamics perspective

A diffusion model characterizes a quasi-static process that occurs between two equilibrium distributions: $q(\mathbf{x}_0) \to q(\mathbf{x}_1)$, via a stochastic trajectory [51]. According to Spinney & Ford [55], it is demonstrated that the diffusion schedule or the noising process plays a pivotal role in determining the "measure of irreversibility" for this stochastic trajectory which is expressed as $\log \frac{P_F(\mathbf{x}_{0:1})}{P_B(\mathbf{x}_{1:0})}$. $P_F(\mathbf{x}_{0:1})$ represents the probability of observing the forward path $\mathbf{x}_{0:1}$ and $P_B(\mathbf{x}_{1:0})$ represents the probability of observing the reverse path $\mathbf{x}_{1:0}$. It's worth noting that $\log \frac{P_F(\mathbf{x}_{0:1})}{P_B(\mathbf{x}_{1:0})}$ corresponds precisely to the ELBO Eq. 1 that we optimize when training a diffusion model. Consequently, thermodynamics asserts that the noise schedule indeed has an impact on the log-likelihood of the diffusion model which contradicts Kingma et al. [20].

# Appendix D  Multivariate noise schedule conditioned on context

Let's say we have a context variable $\mathbf{c} \in \mathbb{R}^m$ that captures high level information about $\mathbf{x}_0$. $\boldsymbol{\alpha}_t(\mathbf{c}), \boldsymbol{\sigma}_t(\mathbf{c}) \in \mathbb{R}_+^d$ are vectors. The timesteps $s, t$ satisfy $0 \leq s < t \leq 1$. Furthermore, we use the following notations:

$$\boldsymbol{\alpha}_{t|s}(\mathbf{c}) = \frac{\boldsymbol{\alpha}_t(\mathbf{c})}{\boldsymbol{\alpha}_s(\mathbf{c})} \tag{37}$$

$$\boldsymbol{\sigma}_{t|s}^2(\mathbf{c}) = \boldsymbol{\sigma}_t^2(\mathbf{c}) - \frac{\boldsymbol{\alpha}_{t|s}^2(\mathbf{c})}{\boldsymbol{\sigma}_s^2(\mathbf{c})} \tag{38}$$

The forward process for such a method is given as:

$$q_\phi(\mathbf{x}_t|\mathbf{x}_s, \mathbf{c}) = \mathcal{N}\left(\boldsymbol{\alpha}_{t|s}(\mathbf{c})\mathbf{x}_s, \boldsymbol{\sigma}_{t|s}^2(\mathbf{c})\right) \tag{39}$$

The distribution of $\mathbf{x}_t$ given $\mathbf{x}_s$ is given by (the derivation is similar to Hoogeboom & Salimans [17]):

$$\begin{aligned} &q_\phi(\mathbf{x}_s|\mathbf{x}_t, \mathbf{x}_0, \mathbf{c}) \\ &= \mathcal{N}\left(\boldsymbol{\mu}_q = \frac{\boldsymbol{\alpha}_{t|s}(\mathbf{c})\boldsymbol{\sigma}_s^2(\mathbf{c})}{\boldsymbol{\sigma}_t^2(\mathbf{c})}\mathbf{x}_t + \frac{\boldsymbol{\sigma}_{t|s}^2(\mathbf{c})\boldsymbol{\alpha}_s(\mathbf{c})}{\boldsymbol{\sigma}_t^2(\mathbf{c})}\mathbf{x}_0, \ \boldsymbol{\Sigma}_q = \mathrm{diag}\left(\frac{\boldsymbol{\sigma}_s^2(\mathbf{c})\boldsymbol{\sigma}_{t|s}^2(\mathbf{c})}{\boldsymbol{\sigma}_t^2(\mathbf{c})}\right)\right) \end{aligned} \tag{40}$$

### D.1 context is available during the inference time.

Even though $\mathbf{c}$ represents the input $\mathbf{x}_0$, it could be available during during inference. For example $\mathbf{c}$ could be class labels [10] or prexisting embeddings from an auto-encoder [40].

#### D.1.1 Reverse Process: Approximate

Let $\mathbf{x}_\theta(\mathbf{x}_t, \mathbf{c}, t)$ be an approximation for $\mathbf{x}_0$. Then we get the following reverse process (for brevity we write $\mathbf{x}_\theta(\mathbf{x}_t, \mathbf{c}, t)$ as $\mathbf{x}_\theta$):

$$p_\theta(\mathbf{x}_s|\mathbf{x}_t, \mathbf{c}) = \mathcal{N}\left(\boldsymbol{\mu}_p = \frac{\boldsymbol{\alpha}_{t|s}(\mathbf{c})\boldsymbol{\sigma}_s^2(\mathbf{c})}{\boldsymbol{\sigma}_t^2(\mathbf{c})}\mathbf{x}_t + \frac{\boldsymbol{\sigma}_{t|s}^2(\mathbf{c})\boldsymbol{\alpha}_s(\mathbf{c})}{\boldsymbol{\sigma}_t^2(\mathbf{c})}\mathbf{x}_\theta, \ \boldsymbol{\Sigma}_p = \mathrm{diag}\left(\frac{\boldsymbol{\sigma}_s^2(\mathbf{c})\boldsymbol{\sigma}_{t|s}^2(\mathbf{c})}{\boldsymbol{\sigma}_t^2(\mathbf{c})}\right)\right) \tag{41}$$

#### D.1.2 Diffusion Loss

Similar to the derivation of multi-variate $\mathcal{L}_{\text{diffusion}}$ in Eq. 33 we can derive $\mathcal{L}_{\text{diffusion}}$ for this case too:

$$\mathcal{L}_{\text{diffusion}} = -\frac{1}{2}\mathbb{E}_{\epsilon\sim\mathcal{N}(0,\mathbf{I}_n), t\sim U[0,1]}\left[(\mathbf{x}_0 - \mathbf{x}_\theta(\mathbf{x}_t, \mathbf{c}, t))^\top \mathrm{diag}\left(\frac{\mathrm{d}}{\mathrm{d}t}\boldsymbol{\nu}(t)\right)(\mathbf{x}_0 - \mathbf{x}_\theta(\mathbf{x}_t, \mathbf{c}, t))\right] \tag{42}$$

#### D.1.3 Limitations of this method

This approach is very limited where the diffusion process is only conditioned on class labels. Using pre-existing embeddings like Diff-AE [40] is also not possible in general and is only limited to tasks such as attribute manipulation in datasets.

### D.2 context isn't available during the inference time.

If the context, $\mathbf{c}$ is an explicit function of the input $\mathbf{x}_0$ things become challenging because $\mathbf{x}_0$ isn't available during the inference stage. For this reason, Eq. 40 can't be used to parameterize $\boldsymbol{\mu}_p, \boldsymbol{\Sigma}_p$ in $p_\theta(\mathbf{x}_s|\mathbf{x}_t)$. Let $p_\theta(\mathbf{x}_s|\mathbf{x}_t) = \mathcal{N}(\boldsymbol{\mu}_p(\mathbf{x}_t, t), \boldsymbol{\Sigma}_p(\mathbf{x}_t, t))$ where $\boldsymbol{\mu}_p, \boldsymbol{\Sigma}_p$ are parameterized directly by a neural network. Using Eq. 4 we get the following diffusion loss:

$$\begin{aligned} \mathcal{L}_{\text{diffusion}} &= T\ \mathbb{E}_{i\sim U[0,T]}D_{\text{KL}}\left(q(\mathbf{x}_{s(i)}|\mathbf{x}_{t(i)}, \mathbf{x}_0)\|p_\theta(\mathbf{x}_{s(i)}|\mathbf{x}_{t(i)})\right) \\ &= \mathbb{E}_{q_\phi}\left(\underbrace{\frac{T}{2}(\boldsymbol{\mu}_q - \boldsymbol{\mu}_p)^\top\boldsymbol{\Sigma}_\theta^{-1}(\boldsymbol{\mu}_q - \boldsymbol{\mu}_p)}_{\text{term 1}} + \underbrace{\frac{T}{2}\left(\mathrm{tr}\left(\boldsymbol{\Sigma}_q\boldsymbol{\Sigma}_p^{-1} - \mathbf{I}_n\right) - \log\frac{|\boldsymbol{\Sigma}_q|}{|\boldsymbol{\Sigma}_p|}\right)}_{\text{term 2}}\right) \end{aligned} \tag{43}$$

#### D.2.1 Reverse Process: Approximate

Due to the challenges associated with parameterizing $\boldsymbol{\mu}_p, \boldsymbol{\Sigma}_p$ directly using a neural network we parameterize $\mathbf{c}$ using a neural network that approximates $\mathbf{c}$ in the reverse process. Let $\mathbf{x}_\theta(\mathbf{x}_t, t)$ be an approximation for $\mathbf{x}_0$. Then we get the following reverse Rrocess (for brevity we write $\mathbf{x}_\theta(\mathbf{x}_t, t)$ as $\mathbf{x}_\theta$, and $\mathbf{c}_\theta$ denotes an approximation to $\mathbf{c}$ in the reverse process.):

$$p_\theta(\mathbf{x}_s|\mathbf{x}_t)$$

$$= \mathcal{N}\left(\boldsymbol{\mu}_p = \frac{\boldsymbol{\alpha}_{t|s}(\mathbf{c}_\theta)\boldsymbol{\sigma}_s^2(\mathbf{c}_\theta)}{\boldsymbol{\sigma}_t^2(\mathbf{c}_\theta)}\mathbf{x}_t + \frac{\boldsymbol{\sigma}_{t|s}^2(\mathbf{c}_\theta)\boldsymbol{\alpha}_s(\mathbf{c}_\theta)}{\boldsymbol{\sigma}_t^2(\mathbf{c}_\theta)}\mathbf{x}_\theta, \ \boldsymbol{\Sigma}_p = \mathrm{diag}\left(\frac{\boldsymbol{\sigma}_s^2(\mathbf{c}_\theta)\boldsymbol{\sigma}_{t|s}^2(\mathbf{c}_\theta)}{\boldsymbol{\sigma}_t^2(\mathbf{c}_\theta)}\right)\right) \quad (44)$$

Consider the limiting case where $T \to \infty$. Let's analyze the 2 terms in Eq. 43 separately.

Using Eq. 4 and Eq. 6, **term 1** in Eq. 43 simplifies in the following manner:

$$\lim_{T\to\infty} \frac{T}{2}(\boldsymbol{\mu}_q - \boldsymbol{\mu}_p)^\top \boldsymbol{\Sigma}_\theta^{-1}(\boldsymbol{\mu}_q - \boldsymbol{\mu}_p)$$

$$\lim_{T\to\infty} \frac{T}{2}\sum_{i=1}^d \frac{((\boldsymbol{\mu}_q)_i - (\boldsymbol{\mu}_p)_i)^2}{(\boldsymbol{\Sigma}_\theta)_i} \quad (45)$$

Substituting 1 / T as $\delta$

$$\lim_{\delta\to 0^+} \sum_{i=1}^d \frac{1}{\delta\boldsymbol{\sigma}_i^2(\mathbf{x}_\theta, t-\delta)\left(1 - \frac{\boldsymbol{\nu}_i(\mathbf{x}_\theta,t)}{\boldsymbol{\nu}_i(\mathbf{x}_\theta,t-\delta)}\right)} \times$$

$$\left[\frac{\boldsymbol{\alpha}_i(\mathbf{x}, t-\delta)}{\boldsymbol{\alpha}_i(\mathbf{x}, t)}\frac{\boldsymbol{\nu}_i(\mathbf{x}, t)}{\boldsymbol{\nu}_i(\mathbf{x}, t-\delta)}\mathbf{z}_t + \boldsymbol{\alpha}_i(\mathbf{x}, t-\delta)\left(1 - \frac{\boldsymbol{\nu}_i(\mathbf{x}, t)}{\boldsymbol{\nu}_i(\mathbf{x}, t-\delta)}\right)x_i \right.$$

$$\left. - \frac{\boldsymbol{\alpha}_i(\mathbf{x}_\theta, t-\delta)}{\boldsymbol{\alpha}_i(\mathbf{x}_\theta, t)}\frac{\boldsymbol{\nu}_i(\mathbf{x}_\theta, t)}{\boldsymbol{\nu}_i(\mathbf{x}_\theta, t-\delta)}\mathbf{z}_t + \boldsymbol{\alpha}_i(\mathbf{x}_\theta, t-\delta)\left(1 - \frac{\boldsymbol{\nu}_i(\mathbf{x}_\theta, t)}{\boldsymbol{\nu}_i(\mathbf{x}_\theta, t-\delta)}\right)(x_\theta)_i\right]^2 \quad (46)$$

Consider the scalar case: substituting $\delta = 1/T$,

$$\lim_{\delta\to 0} \frac{1}{\delta\sigma^2(\mathbf{x}_\theta, t-\delta)\left(1 - \frac{\nu(\mathbf{x}_\theta,t)}{\nu(\mathbf{x}_\theta,t-\delta)}\right)} \times$$

$$\left[\frac{\alpha(\mathbf{x}, t-\delta)}{\alpha(\mathbf{x}, t)}\frac{\nu(\mathbf{x}, t)}{\nu(\mathbf{x}, t-\delta)}\mathbf{z}_t + \alpha(\mathbf{x}, t-\delta)\left(1 - \frac{\nu(\mathbf{x}, t)}{\nu(\mathbf{x}, t-\delta)}\right)\mathbf{x} \right.$$

$$\left. - \frac{\alpha(\mathbf{x}_\theta, t-\delta)}{\alpha(\mathbf{x}_\theta, t)}\frac{\nu(\mathbf{x}_\theta, t)}{\nu(\mathbf{x}_\theta, t-\delta)}\mathbf{z}_t + \alpha(\mathbf{x}_\theta, t-\delta)\left(1 - \frac{\nu(\mathbf{x}_\theta, t)}{\nu(\mathbf{x}_\theta, t-\delta)}\right)\mathbf{x}_\theta\right]^2 \quad (47)$$

Notice that this equation is in indeterminate for when we substitute $\delta = 0$. One can apply L'Hospital rule twice or break it down into 3 terms below. For this reason let's write it as

expression 1: 
$$\lim_{\delta\to 0} \frac{1}{\delta} \times \left[\frac{\alpha(\mathbf{x}, t-\delta)}{\alpha(\mathbf{x}, t)}\frac{\nu(\mathbf{x}, t)}{\nu(\mathbf{x}, t-\delta)}\mathbf{z}_t + \alpha(\mathbf{x}, t-\delta)\left(1 - \frac{\nu(\mathbf{x}, t)}{\nu(\mathbf{x}, t-\delta)}\right)\mathbf{x} \right.$$

$$\left. - \frac{\alpha(\mathbf{x}_\theta, t-\delta)}{\alpha(\mathbf{x}_\theta, t)}\frac{\nu(\mathbf{x}_\theta, t)}{\nu(\mathbf{x}_\theta, t-\delta)}\mathbf{z}_t + \alpha(\mathbf{x}_\theta, t-\delta)\left(1 - \frac{\nu(\mathbf{x}_\theta, t)}{\nu(\mathbf{x}_\theta, t-\delta)}\right)\mathbf{x}_\theta\right] \quad (48)$$

expression 2: 
$$\lim_{\delta\to 0} \frac{1}{\left(1 - \frac{\nu(\mathbf{x}_\theta,t)}{\nu(\mathbf{x}_\theta,t-\delta)}\right)} \times \left[\frac{\alpha(\mathbf{x}, t-\delta)}{\alpha(\mathbf{x}, t)}\frac{\nu(\mathbf{x}, t)}{\nu(\mathbf{x}, t-\delta)}\mathbf{z}_t + \alpha(\mathbf{x}, t-\delta)\left(1 - \frac{\nu(\mathbf{x}, t)}{\nu(\mathbf{x}, t-\delta)}\right)\mathbf{x} \right.$$

$$\left. - \frac{\alpha(\mathbf{x}_\theta, t-\delta)}{\alpha(\mathbf{x}_\theta, t)}\frac{\nu(\mathbf{x}_\theta, t)}{\nu(\mathbf{x}_\theta, t-\delta)}\mathbf{z}_t + \alpha(\mathbf{x}_\theta, t-\delta)\left(1 - \frac{\nu(\mathbf{x}_\theta, t)}{\nu(\mathbf{x}_\theta, t-\delta)}\right)\mathbf{x}_\theta\right]^2 \quad (49)$$

Applying L'Hospital rule in expression 1 we get,

$$\frac{d}{d\delta}\left(\frac{\alpha(\mathbf{x}, t-\delta)}{\alpha(\mathbf{x}, t)}\frac{\nu(\mathbf{x}, t)}{\nu(\mathbf{x}, t-\delta)}\right)\bigg|_{\delta=0} = \frac{\nu(\mathbf{x}, t)}{\alpha(\mathbf{x}, t)}\frac{-\nu(\mathbf{x}, t)\alpha'(\mathbf{x}, t) + \alpha(\mathbf{x}, t)\nu'(\mathbf{x}, t)}{\nu^2(\mathbf{x}, t)}$$

$$= \frac{-\alpha'(\mathbf{x}, t)}{\alpha(\mathbf{x}, t)} + \frac{\nu'(\mathbf{x}, t)}{\nu(\mathbf{x}, t)} \tag{50}$$

$$\frac{d}{d\delta} \alpha(\mathbf{x}, t - \delta) \left(1 - \frac{\nu(\mathbf{x}, t)}{\nu(\mathbf{x}, t - \delta)}\right) \Bigg|_{\delta=0} = -\alpha(\mathbf{x}, t) \frac{\nu'(\mathbf{x}, t)}{\nu(\mathbf{x}, t)} \tag{51}$$

$$\left[ \left( \frac{-\alpha'(\mathbf{x}, t)}{\alpha(\mathbf{x}, t)} + \frac{\nu'(\mathbf{x}, t)}{\nu(\mathbf{x}, t)} + \frac{\alpha'(\mathbf{x}_\theta, t)}{\alpha(\mathbf{x}_\theta, t)} - \frac{\nu'(\mathbf{x}_\theta, t)}{\nu(\mathbf{x}_\theta, t)} \right) \mathbf{z}_t \right. \tag{52}$$

$$\left. -\alpha(\mathbf{x}, t) \frac{\nu'(\mathbf{x}, t)}{\nu(\mathbf{x}, t)} \mathbf{x} + \alpha(\mathbf{x}_\theta, t) \frac{\nu'(\mathbf{x}_\theta, t)}{\nu(\mathbf{x}_\theta, t)} \mathbf{x}_\theta \right]^2 \times \frac{\nu(\mathbf{x}, t)}{\nu'(\mathbf{x}, t)} \tag{53}$$

Thus the final result:

$$\sum_{i=1}^{d} \left[ \left( \frac{-\boldsymbol{\alpha}_i'(\mathbf{x}, t)}{\boldsymbol{\alpha}_i(\mathbf{x}, t)} + \frac{\boldsymbol{\nu}_i'(\mathbf{x}, t)}{\boldsymbol{\nu}_i(\mathbf{x}, t)} + \frac{\boldsymbol{\alpha}_i'(\mathbf{x}_\theta, t)}{\boldsymbol{\alpha}_i(\mathbf{x}_\theta, t)} - \frac{\boldsymbol{\nu}_i'(\mathbf{x}_\theta, t)}{\boldsymbol{\nu}_i(\mathbf{x}_\theta, t)} \right) \mathbf{z}_t \right.$$

$$\left. - \boldsymbol{\alpha}_i(\mathbf{x}, t) \frac{\boldsymbol{\nu}_i'(\mathbf{x}, t)}{\boldsymbol{\nu}_i(\mathbf{x}, t)} \mathbf{x} + \boldsymbol{\alpha}_i(\mathbf{x}_\theta, t) \frac{\boldsymbol{\nu}_i'(\mathbf{x}_\theta, t)}{\boldsymbol{\nu}_i(\mathbf{x}_\theta, t)} \mathbf{x}_\theta \right]^2 \times \frac{\boldsymbol{\nu}_i(\mathbf{x}, t)}{\boldsymbol{\nu}_i'(\mathbf{x}, t)}$$

$$= \Lambda^\top \text{diag}\left( \frac{\boldsymbol{\nu}(\mathbf{x}, t)}{\boldsymbol{\nu}'(\mathbf{x}, t)} \right) \Lambda$$

$$\text{where } \Lambda = \left[ \left( \frac{-\alpha'(\mathbf{x}, t)}{\alpha(\mathbf{x}, t)} + \frac{\nu'(\mathbf{x}, t)}{\nu(\mathbf{x}, t)} + \frac{\alpha'(\mathbf{x}_\theta, t)}{\alpha(\mathbf{x}_\theta, t)} - \frac{\nu'(\mathbf{x}_\theta, t)}{\nu(\mathbf{x}_\theta, t)} \right) \mathbf{z}_t - \alpha(\mathbf{x}, t) \frac{\nu'(\mathbf{x}, t)}{\nu(\mathbf{x}, t)} \mathbf{x} + \alpha(\mathbf{x}_\theta, t) \frac{\nu'(\mathbf{x}_\theta, t)}{\nu(\mathbf{x}_\theta, t)} \mathbf{x}_\theta \right] \tag{54}$$

For the second term we have the following:

$$\lim_{T \to \infty} \frac{T}{2} \left( \text{tr}\left( \boldsymbol{\Sigma}_q \boldsymbol{\Sigma}_p^{-1} - \mathbf{I}_n \right) - \log \frac{|\boldsymbol{\Sigma}_q|}{|\boldsymbol{\Sigma}_p|} \right)$$

$$= \lim_{T \to \infty} \frac{T}{2} \left[ \text{tr}\left( \text{diag}\left( \boldsymbol{\sigma}^2(\mathbf{c}, s) \left(1 - \frac{\boldsymbol{\nu}(\mathbf{c}, t)}{\boldsymbol{\nu}(\mathbf{c}, s)}\right) \right) \Big/ \text{diag}\left( \boldsymbol{\sigma}^2(\mathbf{c}_\theta, s) \left(1 - \frac{\boldsymbol{\nu}(\mathbf{c}_\theta, t)}{\boldsymbol{\nu}(\mathbf{c}_\theta, s)}\right) \right) - \mathbf{I}_n \right) \right.$$

$$\left. - \log \frac{\left| \text{diag}\left( \boldsymbol{\sigma}^2(\mathbf{c}, s)(1 - \frac{\boldsymbol{\nu}(\mathbf{c}, t)}{\boldsymbol{\nu}(\mathbf{c}, s)}) \right) \right|}{\left| \text{diag}\left( \boldsymbol{\sigma}^2(\mathbf{c}_\theta, s)(1 - \frac{\boldsymbol{\nu}(\mathbf{c}_\theta, t)}{\boldsymbol{\nu}(\mathbf{c}_\theta, s)}) \right) \right|} \right]$$

$$= \lim_{T \to \infty} \frac{T}{2} \sum_{i=1}^{d} \left( \frac{\boldsymbol{\sigma}_i^2(\mathbf{c}, s) \left(1 - \frac{\boldsymbol{\nu}_i(\mathbf{c}, t)}{\boldsymbol{\nu}_i(\mathbf{c}, s)}\right)}{\boldsymbol{\sigma}_i^2(\mathbf{c}_\theta, s) \left(1 - \frac{\boldsymbol{\nu}_i(\mathbf{c}_\theta, t)}{\boldsymbol{\nu}_i(\mathbf{c}_\theta, s)}\right)} - 1 - \log \frac{\boldsymbol{\sigma}_i^2(\mathbf{c}, s) \left(1 - \frac{\boldsymbol{\nu}_i(\mathbf{c}, t)}{\boldsymbol{\nu}_i(\mathbf{c}, s)}\right)}{\boldsymbol{\sigma}_i^2(\mathbf{c}_\theta, s) \left(1 - \frac{\boldsymbol{\nu}_i(\mathbf{c}_\theta, t)}{\boldsymbol{\nu}_i(\mathbf{c}_\theta, s)}\right)} \right) \tag{55}$$

$$\tag{56}$$

Let $p_i = \frac{\boldsymbol{\sigma}_i^2(\mathbf{c}, s)\left(1 - \frac{\boldsymbol{\nu}_i(\mathbf{c}, t)}{\boldsymbol{\nu}_i(\mathbf{c}, s)}\right)}{\boldsymbol{\sigma}_i^2(\mathbf{c}_\theta, s)\left(1 - \frac{\boldsymbol{\nu}_i(\mathbf{c}_\theta, t)}{\boldsymbol{\nu}_i(\mathbf{c}_\theta, s)}\right)}$

The sequence $\lim_{T \to \infty} \frac{T}{2} \sum_{i=1}^{d} (p_i - 1 - \log p_i)$ converges iff $\lim_{T \to \infty} \sum_{i=1}^{d} (p_i - 1 - \log p_i) = 0$. Notice that the function $f(x) = x - 1 - \log x \geq 0 \ \ \forall x \in \mathbb{R}$ and the equality holds for $x = 1$. Thus, the condition $\lim_{T \to \infty} \frac{T}{2} \sum_{i=1}^{d} (p_i - 1 - \log p_i)$ holds iff $\lim_{T \to \infty} p_i = 0 \ \ \forall i \in \{1, \ldots, d\}$. Thus,

$$\lim_{T \to \infty} p_i = 1$$

$$\implies \lim_{T \to \infty} \left( \frac{\boldsymbol{\sigma_i}^2(\mathbf{c},s) \left( 1 - \frac{\boldsymbol{\nu}_i(\mathbf{c},t)}{\boldsymbol{\nu}_i(\mathbf{c},s)} \right)}{\boldsymbol{\sigma_i}^2(\mathbf{c}_\theta,s) \left( 1 - \frac{\boldsymbol{\nu}_i(\mathbf{c}_\theta,t)}{\boldsymbol{\nu}_i(\mathbf{c}_\theta,s)} \right)} \right) = 1$$

Substituting $1/T$ as $\delta$,

$$\implies \lim_{\delta \to 0^+} \left( \frac{\boldsymbol{\sigma_i}^2(\mathbf{c},t-\delta) \left( 1 - \frac{\boldsymbol{\nu}_i(\mathbf{c},t)}{\boldsymbol{\nu}_i(\mathbf{c},t-\delta)} \right)}{\boldsymbol{\sigma_i}^2(\mathbf{c}_\theta,t-\delta) \left( 1 - \frac{\boldsymbol{\nu}_i(\mathbf{c}_\theta,t)}{\boldsymbol{\nu}_i(\mathbf{c}_\theta,t-\delta)} \right)} \right) = 1$$

$$\implies \frac{\boldsymbol{\sigma_i}^2(\mathbf{c},t)}{\boldsymbol{\sigma_i}^2(\mathbf{c}_\theta,t)} \lim_{\delta \to 0^+} \left( \frac{1 - \frac{\boldsymbol{\nu}_i(\mathbf{c},t)}{\boldsymbol{\nu}_i(\mathbf{c},t-\delta)})}{1 - \frac{\boldsymbol{\nu}_i(\mathbf{c}_\theta,t)}{\boldsymbol{\nu}_i(\mathbf{c}_\theta,t-\delta)}} \right) = 1$$

Applying L'Hospital rule,

$$\implies \frac{\boldsymbol{\sigma_i}^2(\mathbf{c},t)}{\boldsymbol{\sigma_i}^2(\mathbf{c}_\theta,t)} \left( \frac{\frac{-\boldsymbol{\nu}_i{}'(\mathbf{c},t))}{\boldsymbol{\nu}_i(\mathbf{c},t)}}{\frac{-\boldsymbol{\nu}_i{}'(\mathbf{c}_\theta,t)}{\boldsymbol{\nu}_i(\mathbf{c}_\theta,t)}} \right) = 1$$

$$\implies \frac{\boldsymbol{\sigma_i}^2(\mathbf{c},t)}{\boldsymbol{\sigma_i}^2(\mathbf{c}_\theta,t)} \left( \frac{\boldsymbol{\nu}_i{}'(\mathbf{c},t)\boldsymbol{\nu}_i(\mathbf{c}_\theta,t)}{\boldsymbol{\nu}_i(\mathbf{c},t)\boldsymbol{\nu}_i{}'(\mathbf{c}_\theta,t))} \right) = 1 \tag{57}$$

In the vector form the above equation can be written as,

$$\frac{\boldsymbol{\sigma}_t^2(\mathbf{c})\boldsymbol{\nu}_t(\mathbf{c}_\theta)\nabla_t\boldsymbol{\nu}(\mathbf{c},t)}{\boldsymbol{\sigma}_t^2(\mathbf{c}_\theta)\boldsymbol{\nu}_t(\mathbf{c})\nabla_t\boldsymbol{\nu}(\mathbf{c}_\theta,t)} = \mathbf{1_d} \tag{58}$$

Eq. 58 holds if:

- $x_\theta = x_0$ i.e. the unet can perfectly map $\mathbf{x}_t$ to $\mathbf{x}_0$ $\forall t \in [0,1]$ which is unrealistic.

- Clever parameterizations for $\boldsymbol{\sigma}, \boldsymbol{\alpha}, \boldsymbol{\nu}$ that ensure Eq. 58 holds.

Because of aforementioned challenges we evaluate this method with finite $T = 1000$. We demonstrate the performance of the model empirically in Fig. 4.

### D.2.2 Recovering VDM

If we substitute $\boldsymbol{\nu}_t(\mathbf{c}), \boldsymbol{\nu}_t(\mathbf{c}_\theta)$ with $\boldsymbol{\nu}(t)$ (since the SNR isn't conditioned on the context $\mathbf{c}$), $\boldsymbol{\sigma}_t(\mathbf{c}_\theta), \boldsymbol{\sigma}_t(\mathbf{c})$ with $\sigma_t$ and $\boldsymbol{\alpha}_t(\mathbf{c}_\theta), \boldsymbol{\alpha}_t(\mathbf{c})$ with $\boldsymbol{\alpha}_t$, Eq. 45 reduces to the intermediate loss in VDM i.e. $\frac{1}{2}(\mathbf{x}_\theta - \mathbf{x}_0)^\top \nabla_t \boldsymbol{\nu}(t) \, (\mathbf{x}_\theta - \mathbf{x}_0)$ and Eq. 55 reduces to 0.

### D.3 Experimental results

In Fig. 4 we demonstrate that the multivariate diffusion processes where $\mathbf{c} = $ "class labels" or $\mathbf{c} = \mathbf{x}_0$ perform worse than VDM. Since a continuous time formulation i.e. $T \to \infty$ for the case when $\mathbf{c} = \mathbf{x}_0$ isn't possible (unlike MULAN or VDM) we evaluate these models in the discrete time setting where we use $T = 1000$. Furthermore we also ablate $T = 10k, 100k$ for $\mathbf{c} = \mathbf{x}_0$ to show that the VLB degrades with increasing T whereas for VDM and MULAN it improves for increasing T; see Kingma et al. [20]. This empirical observation is consistent with our mathematical insights earlier. As these models consistently exhibit inferior performance w.r.t VDM, in line with our initial conjectures, we refrain from training them beyond 300k iterations due to the substantial computational cost involved.

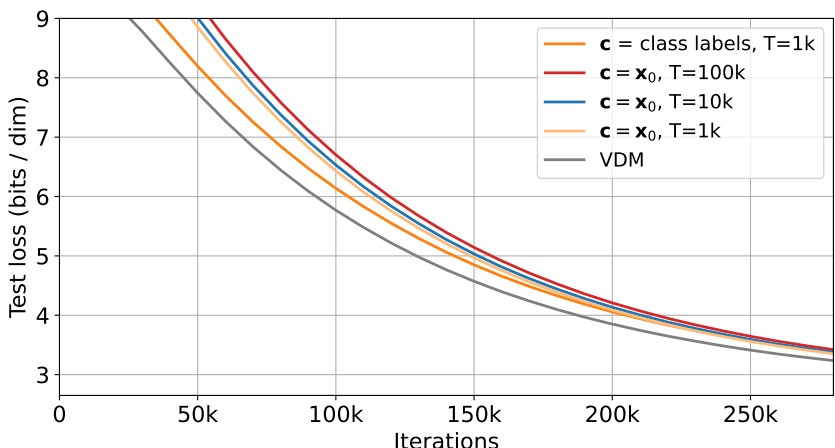

Figure 4: For **c** = "class labels" or **c** = $\mathbf{x}_0$ the likelihood estimates are worse than VDM. For **c** = $\mathbf{x}_0$, we see that the VLB degrades with increasing T, but for VDM and MᴜLAN, it improves with increasing T. This empirical observation is consistent with our mathematical insights earlier. As these models consistently exhibit inferior performance w.r.t VDM, in line with our initial conjectures, we refrain from training them beyond 300k iterations due to the substantial computational cost involved.

Table 7: Likelihood in bits per dimension (BPD) (mean and 95% confidence interval), on the test set of CIFAR-10 computed using VLB estimate.

| parameterization | Num training steps | CIFAR-10 ($\downarrow$) |
|---|---|---|
| Noise parameterization | 10M | $2.60 \pm 10^{-3}$ |
| v-parameterization | 8M | $2.59 \pm 10^{-3}$ |

# Appendix E    MᴜLAN: MUltivariate Latent Auxiliary variable Noise Schedule

## E.1    Parameterization in the reverse process

### E.1.1    Noise parameterization

Since the forward pass is given by $\mathbf{x}_t = \boldsymbol{\alpha}_t(\mathbf{z})\mathbf{x}_0 + \boldsymbol{\sigma}_t(\mathbf{z})\boldsymbol{\epsilon}_t$, we can write the noise $\boldsymbol{\epsilon}_t$ in terms of $\mathbf{x}_0, \mathbf{x}_t$ in the following manner:

$$\boldsymbol{\epsilon}_t = \frac{\mathbf{x}_t - \boldsymbol{\alpha}_t(\mathbf{z})\mathbf{x}_0}{\boldsymbol{\sigma}_t(\mathbf{z})} \tag{59}$$

Following Dhariwal & Nichol [10], Kingma et al. [20], instead of parameterizing $\mathbf{x}_\theta(\mathbf{x}_t, \mathbf{z}, t)$ using a neural network, we use Eq. 59 to parameterize the denoising model in terms of a noise prediction model $\boldsymbol{\epsilon}_\theta(\mathbf{x}_t, \mathbf{z}, t)$,

$$\boldsymbol{\epsilon}_\theta(\mathbf{x}_t, \mathbf{z}, t) = \frac{\mathbf{x}_t - \boldsymbol{\alpha}_t(\mathbf{z})\mathbf{x}_\theta(\mathbf{x}_t, \mathbf{z}, t)}{\boldsymbol{\sigma}_t(\mathbf{z})} \tag{60}$$

### E.1.2    Velocity parameterization

Following Salimans & Ho [45], Zheng et al. [65], we explore another parameterization of the denoising network which is given by

$$\mathbf{v}_\theta(\mathbf{x}_t, \mathbf{z}, t) = \frac{\boldsymbol{\alpha}_t(\mathbf{z})\mathbf{x}_t - \mathbf{x}_\theta(\mathbf{x}_t, \mathbf{z}, t)}{\boldsymbol{\sigma}_t(\mathbf{z})} \tag{61}$$

In practice, v-parameterization leads to a better performance than noise parameterization; as illustrated in Table 7.

## E.2 Polynomial Noise Schedule

Let $f(x; \psi)$ be a scalar-valued polynomial of degree $n$ with coefficients $\psi \in \mathbb{R}^{n+1}$ expressed as:

$$f(x; \psi) = \psi_n x^n + \psi_{n-1} x^{n-1} + \cdots + \psi_1 x + \psi_0,$$

and denote its derivative with respect to $x$ as $\frac{d}{dx} f(x; \psi)$, represented by $f'(x; \psi)$. Now we'd like to find least $n$ such that $f(x; \psi)$ satisfies the following properties:

1. $f(x; \psi)$ is monotonically increasing, i.e. $f'(x; \psi) \geq 0 \ \forall x \in \mathbb{R}, \psi \in \mathbb{R}^{n+1}$.
2. $f'(x_1; \psi) = 0, f'(x_2; \psi) = 0 \ \exists x_1, x_2 \in \mathbb{C}, x_1 \neq x_2, \forall \psi \in \mathbb{R}^{n+1}$.

For the **first** condition to hold, we can design $f'(x; \psi)$ such that it's a perfect square with real / imaginary roots. That way $f'(x; \psi) \geq 0 \ \forall x \in \mathbb{R}, \psi \in \mathbb{R}^{n+1}$. This means that $f'(x; \psi)$ is an even degree polynomial, i.e. the degree of $f'(x; \psi)$ can take the following values: $2, 4, \ldots$. Also, note that at least half of the roots of $f'(x; \psi)$ are repeated since $f'(x; \psi)$ can be expressed as a perfect square, i.e., if $f'(x; \psi)$ has a degree 2 then it has exactly 1 unique root (real / imaginary), if $f'(x; \psi)$ has a degree 4 then it has at most 2 unique roots (real / imaginary), and so on.

For the **second** condition to hold, $f'(x; \psi)$ needs to have at least 2 unique roots $\exists \psi \in \mathbb{R}^{n+1}$. For this reason $f'(x; \psi)$ is a polynomial of degree 4. Thus, $f'(x; \psi)$ can be written as $f'(x; \psi) = (\psi_3 x^2 + \psi_2 x + \psi_1)^2$. This ensures that $\exists \psi \in \mathbb{R}^5$ s.t. $f'(x; \psi) = 0$ twice in $x \in \mathbb{R}$, and $f'(x; \psi) \geq 0 \ \forall \psi \in \mathbb{R}^5$.

Thus, $f(x; \psi)$ takes the following functional form:

$$
\begin{aligned}
f(x; \psi) &= \int (\psi_3 x^2 + \psi_2 x + \psi_1)^2 dx \\
&= \frac{\psi_3^2}{5} x^5 + \frac{\psi_3 \psi_2}{2} x^4 + \frac{\psi_2^2 + 2\psi_3 \psi_1}{3} x^3 + \psi_2 \psi_1 x^2 + \psi_1^2 x + \text{constant}.
\end{aligned}
\tag{62}
$$

For the above-mentioned reasons we express $\gamma_\phi(\mathbf{c}, t) : \mathbb{R}^m \times [0, 1] \to \mathbb{R}^d$ as a degree 5 polynomial in $t$. We define neural networks $\mathbf{a}_\phi(\mathbf{c}) : \mathbb{R}^m \to \mathbb{R}^d$, $\mathbf{b}_\phi(\mathbf{c}) : \mathbb{R}^m \to \mathbb{R}^d$, and $\mathbf{d}_\phi(\mathbf{c}) : \mathbb{R}^m \to \mathbb{R}^d$ with parameters $\phi$. Let $f_\phi : \mathbb{R}^m \times [0, 1] \to \mathbb{R}^d$ be defined as:

$$f_\phi(\mathbf{c}, t) = \frac{\mathbf{a}_\phi^2(\mathbf{c})}{5} t^5 + \frac{\mathbf{a}_\phi(\mathbf{c})\mathbf{b}_\phi(\mathbf{c})}{2} t^4 + \frac{\mathbf{b}_\phi^2(\mathbf{c}) + 2\mathbf{a}_\phi(\mathbf{c})\mathbf{d}_\phi(\mathbf{c})}{3} t^3 + \mathbf{b}_\phi(\mathbf{c})\mathbf{d}_\phi(\mathbf{c})t^2 + \mathbf{d}_\phi^2(\mathbf{c})t$$

where the multiplication and division operations are elementwise. The the noise schedule, $\gamma(\mathbf{c}, t)$, is given as follows:

$$\gamma_\phi(\mathbf{c}, t) = \gamma_{\min} + (\gamma_{\max} - \gamma_{\min}) \frac{f_\phi(\mathbf{c}, t)}{f_\phi(\mathbf{c}, t = 1)} \tag{63}$$

Notice that $\gamma_\phi(\mathbf{c}, t)$ has these interesting properties:

- Is increasing in $t \in [0, 1]$ which is crucial as mentioned in Sec. 3.5.
- $\gamma_\phi(\mathbf{c}, t)$ has end points at $t = 0$ and $t = 1$ which the user can specify via $\gamma_{\min}$ and $\gamma_{\max}$. Specificaly, $\gamma_\phi(\mathbf{c}, t = 0) = \gamma_{\min} \mathbf{1_d}$ and $\gamma_\phi(\mathbf{c}, t = 1) = \gamma_{\max} \mathbf{1_d}$.
- Its time-derivative i.e. $\nabla_t \gamma_\phi(\mathbf{c}, t)$ **can** be zero twice in $t \in [0, 1]$. This isn't a necessary condition but it's nice to have a flexible noise schedule whose time-derivative can be 0 at the beginning and the end of the diffusion process.

## E.3 Variational Lower Bound

In this section we derive the VLB. For ease of reading we use the notation $\mathbf{x}_t$ to denote $\mathbf{x}_{t(i)}$ and $\mathbf{x}_{t-1}$ to denote $\mathbf{x}_{t(i-1)} \equiv \mathbf{x}_{s(i)}$ in the following derivation.

$$
\begin{aligned}
&- \log p_\theta(\mathbf{x}_0) \\
&\leq \mathbb{E}_{q_\phi} \left[ -\log \frac{p_\theta(\mathbf{z}, \mathbf{x}_{0:T})}{q_\phi(\mathbf{z}, \mathbf{x}_{1:T}|\mathbf{x}_0)} \right] \\
&= \mathbb{E}_{q_\phi} \left[ -\log \frac{p_\theta(\mathbf{x}_{0:T-1}|\mathbf{z}, \mathbf{x}_T)}{q_\phi(\mathbf{z}, \mathbf{x}_{1:T}|\mathbf{x}_0)} - \log p_\theta(\mathbf{x}_T) - \log p_\theta(\mathbf{z}) \right]
\end{aligned}
$$

$$= \mathbb{E}_{q_\phi}\left[-\log \frac{p_\theta(\mathbf{x}_{0:T-1}|\mathbf{z}, \mathbf{x}_T)}{q_\phi(\mathbf{x}_{1:T}|\mathbf{z}, \mathbf{x}_0)} - \log \frac{1}{q_\phi(\mathbf{z}|\mathbf{x}_0)} - \log p_\theta(\mathbf{x}_T) - \log p_\theta(\mathbf{z})\right]$$

$$= \mathbb{E}_{q_\phi}\left[-\log \frac{p_\theta(\mathbf{x}_{0:T-1}|\mathbf{z}, \mathbf{x}_T)}{q_\phi(\mathbf{x}_{1:T}|\mathbf{z}, \mathbf{x}_0)} - \log p_\theta(\mathbf{x}_T) - \log \frac{p_\theta(\mathbf{z})}{q_\phi(\mathbf{z}|\mathbf{x}_0)}\right]$$

$$= \mathbb{E}_{q_\phi}\left[-\sum_{t=1}^{T} \log \frac{p_\theta(\mathbf{x}_{t-1}|\mathbf{z}, \mathbf{x}_t)}{q_\phi(\mathbf{x}_t|\mathbf{x}_{t-1}, \mathbf{z}, \mathbf{x}_0)} - \log p_\theta(\mathbf{x}_T) - \log \frac{p_\theta(z)}{q_\phi(z|\mathbf{x}_0)}\right]$$

$$= \mathbb{E}_{q_\phi}\left[-\log \frac{p_\theta(\mathbf{x}_0|\mathbf{z}, \mathbf{x}_1)}{q_\phi(\mathbf{x}_1|\mathbf{x}_0, \mathbf{z})} - \sum_{t=2}^{T} \log \frac{p_\theta(\mathbf{x}_{t-1}|\mathbf{z}, \mathbf{x}_t)}{q_\phi(\mathbf{x}_t|\mathbf{x}_{t-1}, \mathbf{z}, \mathbf{x}_0)} - \log p_\theta(\mathbf{x}_T) - \log \frac{p_\theta(z)}{q_\phi(\mathbf{z}|\mathbf{x}_0)}\right]$$

$$= \mathbb{E}_{q_\phi}\left[-\log \frac{p_\theta(\mathbf{x}_0|\mathbf{z}, \mathbf{x}_1)}{q_\phi(\mathbf{x}_1|\mathbf{x}_0, \mathbf{z})} - \sum_{t=2}^{T} \log \frac{p_\theta(\mathbf{x}_{t-1}|\mathbf{z}, \mathbf{x}_t)q_\phi(\mathbf{x}_{t-1}|\mathbf{z}, \mathbf{x}_0)}{q_\phi(\mathbf{x}_{t-1}|\mathbf{x}_t, \mathbf{z}, \mathbf{x}_0)q_\phi(\mathbf{x}_t|\mathbf{z}, \mathbf{x}_0)} - \log p_\theta(\mathbf{x}_T) - \log \frac{p_\theta(\mathbf{z})}{q_\phi(\mathbf{z}|\mathbf{x}_0)}\right]$$

$$= \mathbb{E}_{q_\phi}\left[-\log \frac{p_\theta(\mathbf{x}_0|\mathbf{z}, \mathbf{x}_1)}{q_\phi(\mathbf{x}_1|\mathbf{x}_0, \mathbf{z})} - \sum_{t=2}^{T} \log \frac{p_\theta(\mathbf{x}_{t-1}|\mathbf{z}, \mathbf{x}_t)}{q_\phi(\mathbf{x}_{t-1}|\mathbf{x}_t, \mathbf{z}, \mathbf{x}_0)} - \sum_{t=2}^{T} \log \frac{q_\phi(\mathbf{x}_{t-1}|\mathbf{z}, \mathbf{x}_0)}{q_\phi(\mathbf{x}_t|\mathbf{z}, \mathbf{x}_0)} - \log p_\theta(\mathbf{x}_T) - \log \frac{p_\theta(z)}{q_\phi(\mathbf{z}|\mathbf{x}_0)}\right]$$

$$= \mathbb{E}_{q_\phi}\left[-\log \frac{p_\theta(\mathbf{x}_0|\mathbf{z}, \mathbf{x}_1)}{q_\phi(\mathbf{x}_1|\mathbf{x}_0, \mathbf{z})} - \sum_{t=2}^{T} \log \frac{p_\theta(\mathbf{x}_{t-1}|\mathbf{z}, \mathbf{x}_t)}{q_\phi(\mathbf{x}_{t-1}|\mathbf{x}_t, \mathbf{z}, \mathbf{x}_0)} - \log \frac{q(\mathbf{x}_1|\mathbf{z}, \mathbf{x}_0)}{q_\phi(\mathbf{x}_T|\mathbf{z}, \mathbf{x}_0)} - \log p_\theta(\mathbf{x}_T) - \log \frac{p_\theta(\mathbf{z})}{q_\phi(\mathbf{z}|\mathbf{x}_0)}\right]$$

$$= \mathbb{E}_{q_\phi}\left[-\log p_\theta(\mathbf{x}_0|\mathbf{z}, \mathbf{x}_1) - \sum_{t=2}^{T} \log \frac{p_\theta(\mathbf{x}_{t-1}|\mathbf{z}, \mathbf{x}_t)}{q_\phi(\mathbf{x}_{t-1}|\mathbf{x}_t, \mathbf{z}, \mathbf{x}_0)} - \log \frac{1}{q_\phi(\mathbf{x}_T|\mathbf{z}, \mathbf{x}_0)} - \log p_\theta(\mathbf{x}_T) - \log \frac{p_\theta(\mathbf{z})}{q_\phi(\mathbf{z}|\mathbf{x}_0)}\right]$$

$$= \mathbb{E}_{q_\phi}\left[-\log p_\theta(\mathbf{x}_0|\mathbf{z}, \mathbf{x}_1) - \sum_{t=2}^{T} \log \frac{p_\theta(\mathbf{x}_{t-1}|\mathbf{z}, \mathbf{x}_t)}{q_\phi(\mathbf{x}_{t-1}|\mathbf{x}_t, \mathbf{z}, \mathbf{x}_0)} - \log \frac{p_\theta(\mathbf{x}_T)}{q_\phi(\mathbf{x}_T|\mathbf{z}, \mathbf{x}_0)} - \log \frac{p_\theta(\mathbf{z})}{q_\phi(\mathbf{z}|\mathbf{x}_0)}\right]$$

$$= \mathbb{E}_{q_\phi}\left[\underbrace{-\log p_\theta(\mathbf{x}_0|\mathbf{z}, \mathbf{x}_1)}_{\mathcal{L}_{\text{recons}}} + \underbrace{\sum_{t=2}^{T} \mathrm{D}_{\mathrm{KL}}[p_\theta(\mathbf{x}_{t-1}|\mathbf{z}, \mathbf{x}_t)\|q_\phi(\mathbf{x}_{t-1}|\mathbf{x}_t, \mathbf{z}, \mathbf{x}_0)]}_{\mathcal{L}_{\text{diffusion}}}\right]$$

$$+ \mathbb{E}_{q_\phi}\left[\underbrace{\mathrm{D}_{\mathrm{KL}}[p_\theta(\mathbf{x}_T)\|q_\phi(\mathbf{x}_T|\mathbf{z}, \mathbf{x}_0)]}_{\mathcal{L}_{\text{prior}}} + \underbrace{\mathrm{D}_{\mathrm{KL}}[p_\theta(\mathbf{z})\|q(\mathbf{z}|\mathbf{x}_0)]}_{\mathcal{L}_{\text{latent}}}\right] \tag{64}$$

Switching back to the notation used throughout the paper, the VLB is given as:

$$-\log p_\theta(\mathbf{x}_0)$$

$$\leq \mathbb{E}_{q_\phi}\left[\underbrace{-\log p_\theta(\mathbf{x}_0|\mathbf{z}, \mathbf{x}_1)}_{\mathcal{L}_{\text{recons}}} + \underbrace{\sum_{i=2}^{T} \mathrm{D}_{\mathrm{KL}}[p_\theta(\mathbf{x}_{s(i)}|\mathbf{z}, \mathbf{x}_{t(i)})\|q_\phi(\mathbf{x}_{s(i)}|\mathbf{x}_{t(i)}, \mathbf{z}, \mathbf{x}_0)]}_{\mathcal{L}_{\text{diffusion}}}\right]$$

$$+ \mathbb{E}_{q_\phi}\left[\underbrace{\mathrm{D}_{\mathrm{KL}}[p_\theta(\mathbf{x}_1)\|q_\phi(\mathbf{x}_1|\mathbf{z}, \mathbf{x}_0)]}_{\mathcal{L}_{\text{prior}}} + \underbrace{\mathrm{D}_{\mathrm{KL}}[p_\theta(\mathbf{z})\|q_\phi(\mathbf{z}|\mathbf{x}_0)]}_{\mathcal{L}_{\text{latent}}}\right] \tag{65}$$

### E.4   Diffusion Loss

To derive the diffusion loss, $\mathcal{L}_{\text{diffusion}}$ in Eq. 9, we first derive an expression for $\mathrm{D}_{\mathrm{KL}}(q_\phi(\mathbf{x}_s|\mathbf{z}, \mathbf{x}_t, \mathbf{x}_0)\|p_\theta(\mathbf{x}_s|\mathbf{z}, \mathbf{x}_t))$ using Eq. 4 and Eq. 6 in the following manner (details in Suppl. E):

$$\mathrm{D}_{\mathrm{KL}}(q_\phi(\mathbf{x}_s|\mathbf{z}, \mathbf{x}_t, \mathbf{x}_0)\|p_\theta(\mathbf{x}_s|\mathbf{z}, \mathbf{x}_t))$$

$$= \frac{1}{2}\left((\boldsymbol{\mu}_{q_\phi} - \boldsymbol{\mu}_p)^\top \boldsymbol{\Sigma}_\theta^{-1}(\boldsymbol{\mu}_{q_\phi} - \boldsymbol{\mu}_p) + \mathrm{tr}\left(\boldsymbol{\Sigma}_{q_\phi}\boldsymbol{\Sigma}_p^{-1} - \mathbf{I}_n\right) - \log \frac{|\boldsymbol{\Sigma}_{q_\phi}|}{|\boldsymbol{\Sigma}_p|}\right)$$

$$= \frac{1}{2}\left((\mathbf{x}_0 - \mathbf{x}_\theta)^\top \mathrm{diag}(\boldsymbol{\nu}(\mathbf{z}, s) - \boldsymbol{\nu}(\mathbf{z}, t))(\mathbf{x}_0 - \mathbf{x}_\theta)\right) \tag{66}$$

Let $\lim_{T\to\infty} T(\boldsymbol{\nu}_s(z) - \boldsymbol{\nu}_t(z)) = -\nabla_t\boldsymbol{\nu}(\mathbf{z},t)$ be the partial derivative of the vector $\boldsymbol{\nu}(\mathbf{z},t)$ w.r.t scalar $t$. Then we derive the diffusion loss, $\mathcal{L}_{\text{diffusion}}$, for the continuous case in the following manner (for brevity we use the notation $s$ for $s(i) = (i-1)/T$ and $t$ for $t(i) = i/T$):

$\mathcal{L}_{\text{diffusion}}$

$$= \lim_{T\to\infty} \frac{1}{2} \sum_{i=2}^{T} \mathbb{E}_{\epsilon\sim\mathcal{N}(0,\mathbf{I}_n)} D_{\text{KL}}(q(\mathbf{x}_s|\mathbf{x}_t,\mathbf{x}_0,\mathbf{z}) \| p_\theta(\mathbf{x}_s|\mathbf{x}_t,\mathbf{z}))$$

Using Eq. 66 we get,

$$= \lim_{T\to\infty} \frac{1}{2} \sum_{i=2}^{T} \mathbb{E}_{\epsilon\sim\mathcal{N}(0,\mathbf{I}_n)} (\mathbf{x}_0 - \mathbf{x}_\theta(\mathbf{x}_t,t(i)))^\top \text{diag}\left(\boldsymbol{\nu}(s(i),\mathbf{z}) - \boldsymbol{\nu}(t(i),\mathbf{z})\right)(\mathbf{x}_0 - \mathbf{x}_\theta(\mathbf{x}_t,t(i)))$$

$$= \frac{1}{2} \mathbb{E}_{\epsilon\sim\mathcal{N}(0,\mathbf{I}_n)}\left[ \lim_{T\to\infty} \sum_{i=2}^{T} T(\mathbf{x}_0 - \mathbf{x}_\theta(\mathbf{x}_t,t(i)))^\top \text{diag}\left(\boldsymbol{\nu}(s(i),\mathbf{z}) - \boldsymbol{\nu}(t(i),\mathbf{z})\right)(\mathbf{x}_0 - \mathbf{x}_\theta(\mathbf{x}_t,t(i)))\frac{1}{T} \right]$$

Using the fact that $\lim_{T\to\infty} T\left(\boldsymbol{\nu}(s,\mathbf{z}) - \boldsymbol{\nu}(\mathbf{z},t)\right) = -\nabla_t\boldsymbol{\nu}(t,\mathbf{z})$ we get,

$$= -\frac{1}{2} \mathbb{E}_{t\sim\{0,...,1\}}\left[ (\mathbf{x}_0 - \mathbf{x}_\theta(\mathbf{x}_t,t))^\top (\nabla_t\boldsymbol{\nu}_t(z))(\mathbf{x}_0 - \mathbf{x}_\theta(\mathbf{x}_t,t)) \right]$$

Substituting $\mathbf{x}_0 = \boldsymbol{\alpha}_t^{-1}(\mathbf{z})(\mathbf{x}_t - \boldsymbol{\sigma}_t(\mathbf{z})\boldsymbol{\epsilon}_t)$ from Eq. 59 and

Substituting $\mathbf{x}_\theta(\mathbf{x}_t,\mathbf{z},t) = \boldsymbol{\alpha}_t^{-1}(\mathbf{z})(\mathbf{x}_t - \boldsymbol{\sigma}_t(\mathbf{z})\boldsymbol{\epsilon}_\theta(\mathbf{x}_t,t))$ from Eq. 60 we get,

$$= -\frac{1}{2} \mathbb{E}_{t\sim[0,1]}\left[ (\boldsymbol{\epsilon}_t - \boldsymbol{\epsilon}_\theta(\mathbf{x}_t,t))^\top \left( \frac{\boldsymbol{\sigma}_t^2(\mathbf{z})}{\boldsymbol{\alpha}_t^2(\mathbf{z})} \times \nabla_t\boldsymbol{\nu}_t(\mathbf{z}) \right)(\boldsymbol{\epsilon}_t - \boldsymbol{\epsilon}_\theta(\mathbf{x}_t,t)) \right]$$

Let $\boldsymbol{\nu}^{-1}(\mathbf{z},t)$ denote the reciprocal of the values in the vector $\boldsymbol{\nu}(\mathbf{z},t)$.

$$= -\frac{1}{2} \mathbb{E}_{t\sim[0,1]}\left[ (\boldsymbol{\epsilon}_t - \boldsymbol{\epsilon}_\theta(\mathbf{x}_t,t))^\top \text{diag}\left(\boldsymbol{\nu}^{-1}(t)(\mathbf{z})\nabla_t\boldsymbol{\nu}_t(\mathbf{z})\right)(\boldsymbol{\epsilon}_t - \boldsymbol{\epsilon}_\theta(\mathbf{x}_t,t)) \right]$$

Substituting $\boldsymbol{\nu}(\mathbf{z},t) = \exp(-\boldsymbol{\gamma}(\mathbf{z},t))$ from Sec. E.1.1

$$= -\frac{1}{2} \mathbb{E}_{t\sim[0,1]}\left[ (\boldsymbol{\epsilon}_t - \boldsymbol{\epsilon}_\theta(\mathbf{x}_t,t))^\top \text{diag}\left(\exp\left(\boldsymbol{\gamma}(\mathbf{z},t)\right)\nabla_t\exp\left(-\boldsymbol{\gamma}(\mathbf{z},t)\right)\right)(\boldsymbol{\epsilon}_t - \boldsymbol{\epsilon}_\theta(\mathbf{x}_t,t)) \right]$$

$$= \frac{1}{2} \mathbb{E}_{t\sim[0,1]}\left[ (\boldsymbol{\epsilon}_t - \boldsymbol{\epsilon}_\theta(\mathbf{x}_t,t))^\top \text{diag}\left(\exp\left(\boldsymbol{\gamma}(\mathbf{z},t)\right)\exp\left(-\boldsymbol{\gamma}(\mathbf{z},t)\right)\nabla_t\boldsymbol{\gamma}(\mathbf{z},t)\right)(\boldsymbol{\epsilon}_t - \boldsymbol{\epsilon}_\theta(\mathbf{x}_t,t)) \right]$$

$$= \frac{1}{2} \mathbb{E}_{t\sim[0,1]}\left[ (\boldsymbol{\epsilon}_t - \boldsymbol{\epsilon}_\theta(\mathbf{x}_t,t))^\top \text{diag}\left(\nabla_t\boldsymbol{\gamma}(\mathbf{z},t)\right)(\boldsymbol{\epsilon}_t - \boldsymbol{\epsilon}_\theta(\mathbf{x}_t,t)) \right] \tag{67}$$

### E.5 Recovering VDM from the Vectorized Representation of the diffusion loss

Notice that we recover the loss function in VDM when $\boldsymbol{\nu}(\mathbf{z},t) = \nu(t)\mathbf{1_d}$ where $\nu_t \in \mathbb{R}^+$ and $\mathbf{1_d}$ represents a vector of 1s of size $d$ and the noising schedule isn't conditioned on $\mathbf{z}$.

$$\int_0^1 \langle \mathbf{f}_\theta(\mathbf{x}_0, \boldsymbol{\nu}(\mathbf{z},t)), \frac{\mathrm{d}}{\mathrm{d}t}\boldsymbol{\nu}(t)\rangle \mathrm{d}t = \int_0^1 \langle \mathbf{f}_\theta(\mathbf{x}_0, \boldsymbol{\nu}(t)), \frac{\mathrm{d}}{\mathrm{d}t}(\nu(t)\mathbf{1_n})\rangle \mathrm{d}t$$

$$= \int_0^1 \langle \mathbf{f}_\theta(\mathbf{x}_0, \boldsymbol{\nu}(t)), \mathbf{1_d}\rangle \nu'(t) \mathrm{d}t$$

$$= \int_0^1 \nu'(t)\|\mathbf{f}_\theta(\mathbf{x}_0, \boldsymbol{\nu}(t))\|_1^1 \mathrm{d}t$$

$$= \int_0^1 \nu'(t)\|(\mathbf{x}_0 - \tilde{\mathbf{x}}_\theta(\mathbf{x}_{\boldsymbol{\nu}(t)}, \boldsymbol{\nu}(t)))\|_2^2 \mathrm{d}t \tag{68}$$

$\int_0^1 \frac{d}{dt}\nu(t)\|(\mathbf{x}_0 - \tilde{\mathbf{x}}_\theta(\mathbf{x}_{\boldsymbol{\nu}(t)}, \boldsymbol{\nu}(t)))\|_2^2 dt$ denotes the diffusion loss, $\mathcal{L}_{\text{diffusion}}$, as used in VDM; see Kingma et al. [20].

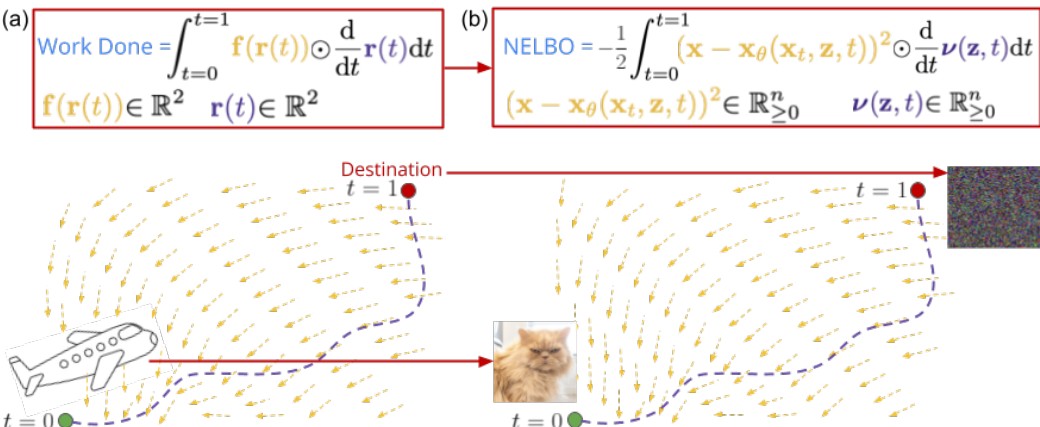

Figure 5: **(a)** Imagine piloting a plane across a region with cyclones and strong winds, as shown in Fig. 5. Plotting a direct, straight-line course through these adverse weather conditions requires more fuel and effort due to increased resistance. By navigating around the cyclones and winds, however, the plane reaches its destination with less energy, even if the route is longer. This intuition translates into mathematical and physical terms. The plane's trajectory is denoted by $\mathbf{r}(t) \in \mathbb{R}^n_+$, while the forces acting on it are represented by $\mathbf{f}(\mathbf{r}(t)) \in \mathbb{R}^n$. The work required to navigate is given by $\int_0^1 \mathbf{f}(\mathbf{r}(t)) \cdot \frac{d}{dt}\mathbf{r}(t), dt$. Here, the work depends on the trajectory because $\mathbf{f}(\mathbf{r}(t))$ is not a conservative field.
**(b)** This concept also applies to the diffusion NELBO. From Eq. 12, it's clear that the trajectory $\mathbf{r}(t)$ is parameterized by the noise schedule $\boldsymbol{\nu}(\mathbf{z}, t)$, which is influenced by complex forces, $\mathbf{f}$ (analogous to weather patterns), represented by the dimension-wise reconstruction error of the denoising model, $(\mathbf{x}_0 - \mathbf{x}_\theta(\mathbf{x}_t, \mathbf{z}, t))^2$. Thus, the diffusion loss, $\mathcal{L}_{\text{diffusion}}$, can be interpreted as the work done along the trajectory $\boldsymbol{\nu}(\mathbf{z}, t)$ in the presence of these vector field forces $\mathbf{f}$. By learning the noise schedule, we can avoid "high-resistance" paths (those where the loss accumulates rapidly), thereby minimizing the overall "energy" expended, as measured by the NELBO.

## Appendix F    Subset Sampling

Sampling a subset of $k$ items from a collection of collection of $n$ items, $x_1, x_2, \ldots, x_3$ belongs to a category of algorithms called reservoir algorithms. In weighted reservoir sampling, every $x_i$ is associated with a weight $w_i \geq 0$. The probability associated with choosing the sequence $S_{\text{wrs}} = [i_1, i_2, \ldots, i_k]$ be a tuple of indices. Then the probability associated with sampling this sequence is

$$p(S_{\text{wrs}}|\mathbf{w}) = \frac{w_{i_1}}{Z} \frac{w_{i_2}}{Z - w_{i_1}} \cdots \frac{w_{i_k}}{Z - \sum_{j=1}^{k-1} w_{i_j}} \tag{69}$$

Efraimidis & Spirakis [13] give an algorithm for weighted reservoir sampling where each item is assigned a random key $r_i = u_i^{\frac{1}{w_i}}$ where $u_i$ is drawn from a uniform distribution [0, 1] and $w_i$ is the weight of item $x_i$. Let TopK$(\mathbf{r}, k)$ which takes keys $\mathbf{r} = [r_1, r_2, \ldots, r_n]$ and returns a sequence $[i_1, i_2, \ldots, i_k]$. Efraimidis & Spirakis [13] proved that TopK$(\mathbf{r}, k)$ is distributed according to $p(S_{\text{wrs}}|\mathbf{w})$.

Let's represent a subset $S \in \{0, 1\}^n$ with exactly $k$ non-zero elements that are equal to 1. Then the probability associated with sampling $S$ is given as,

$$p(S|\mathbf{w}) = \sum_{S_{\text{wrs}} \in \Pi(S)} p(S_{\text{wrs}}|\mathbf{w}) \tag{70}$$

where $\Pi(S)$ denotes all possible permutations of the sequence $S$. By ignoring the ordering of the elements in $S_{\text{wrs}}$ we can sample using the same algorithm. Xie & Ermon [61] show that this sampling algorithm is equivalent to TopK$(\hat{\mathbf{r}}, k)$ where $\hat{\mathbf{r}} = [\hat{r}_1, \hat{r}_2, \ldots, \hat{r}_n]$ where $\hat{r}_i = -\log(-\log(r_i)) = \log w_i + \text{Gumbel}(0, 1)$. This holds true because the monotonic transformation $-\log(-\log(x))$ preserves the ordering of the keys and thus TopK$(\mathbf{r}, k) \equiv$ TopK$(\hat{\mathbf{r}}, k)$.

**Sum of Gamma Distribution.** Niepert et al. [34] show that adding SOG noise instead of Gumbel noise leads to better performance.

Niepert et al. [34] show that $\mathbf{z} \sim p_\theta(\mathbf{z}; \theta)$ is equivalent to $\mathbf{z} = \arg\max_{y \in Y} \langle \theta + \epsilon_g, y \rangle$ where $\epsilon_g$ is a sample from Sum-of-Gamma distribution given by

$$\text{SoG}(k, \tau, s) = \frac{\tau}{k} \left( \sum_{i=1}^{s} \text{Gamma}\left(\frac{1}{k}, \frac{k}{i}\right) - \log s \right), \tag{71}$$

where $s$ is a positive integer and $\text{Gamma}(\alpha, \beta)$ is the Gamma distribution with $(\alpha, \beta)$ as the shape and scale parameters.

And hence, given logits $\log \mathbf{w}$, we sample a $k$-hot vector using $\text{TopK}(\log \mathbf{w} + \epsilon)$. We choose a categorical prior with uniform distribution across $n$ classes. Thus the KL loss term is given by:

$$- \sum_{i=1}^{n} \frac{w_i}{Z} \log \left( n \frac{w_i}{Z} \right) \tag{72}$$

## Appendix G  Experiment Details

### G.1  Model Architecture

**Denoising network.** Our model architecture is extremely similar to VDM. The UNet of our pixel-space diffusion has an unchanged architecture from Kingma et al. [20].This structure is specifically designed for optimal performance in maximum likelihood training. We employ features from VDM such as the elimination of internal downsampling/upsampling processes and the integration of Fourier features to enhance fine-scale prediction accuracy. In alignment with the configurations suggested by Kingma et al. (2021), our approach varies depending on the dataset: For CIFAR-10, we employ a U-Net with a depth of 32 and 128 channels; for ImageNet-32, the U-Net also has a depth of 32, but the channel count is increased to 256. Additionally, all these models incorporate a dropout rate of 0.1 in their intermediate layers.

**Encoder network.** $q_\phi(\mathbf{z}|\mathbf{x})$ is modeled using a sequence of 4 Resnet blocks with a channel count of 128 for CIFAR-10 and 256 for ImageNet-32 with a drop out of 0.1 in their intermediate layers.

**Noise schedule.** For polynomial noise schedule, we use an MLP that maps the latent vector $\mathbf{z}$ to $\mathbf{a}_\phi(\mathbf{z}), \mathbf{b}_\phi(\mathbf{z}), \mathbf{c}(\mathbf{z})$; see Eq. E.2 for details. The MLP has 2 hidden layers of size 3072 with `swish` activation function. The final layer is a linear mapping to $3 \times 3072$ values corresponding to $\mathbf{a}_\phi(\mathbf{z}), \mathbf{b}_\phi(\mathbf{z}), \mathbf{c}(\mathbf{z})$. Note that $\mathbf{a}_\phi(\mathbf{z}), \mathbf{b}_\phi(\mathbf{z}), \mathbf{c}(\mathbf{z})$ have the same dimensionality of 3072.

### G.2  Hardware.

For the ImageNet experiments, we used a single GPU node with 8-A100s. For the cifar-10 experiments, the models were trained on 4 GPUs spanning several GPUs types like V100, A5000s, A6000s, and 3090s with `float32` precision.

### G.3  Hyperparameters

We follow the same default training settings as Kingma et al. [20]. For all our experiments, we use the Adam [21] optimizer with learning rate $2 \times 10^{-4}$, exponential decay rates of $\beta_1 = 0.9$, $\beta_2 = 0.99$ and decoupled weight decay [29] coefficient of 0.01. We also maintain an exponential moving average (EMA) of model parameters with an EMA rate of 0.9999 for evaluation. For other hyperparameters, we use fixed start and end times which satisfy $\gamma_{\min} = -13.3$, $\gamma_{\max} = 5.0$, which is used in Kingma et al. [20], Zheng et al. [65].

## Appendix H  Datasets and Visualizations

In this section we provide a brief description of the datasets used in the paper and visualize the generated samples and the noise schedules.

## H.1 CIFAR-10

The CIFAR-10 dataset [25] is a collection of images consisting of 60,000 $32 \times 32$ color images in 10 different classes, with each class representing a distinct object or scene. The dataset is divided into 50,000 training images and 10,000 test images, with each class having an equal representation in both sets. The classes in CIFAR-10 include: Airplane, Automobile, Bird, Cat, Deer, Dog, Frog, Horse, Ship, Truck.

Randomly generated samples for the CIFAR-10 datasaet are provided in Fig. 6a for MULAN and Fig. 6b for VDM. We visualize the noise schedule in Fig. 13.

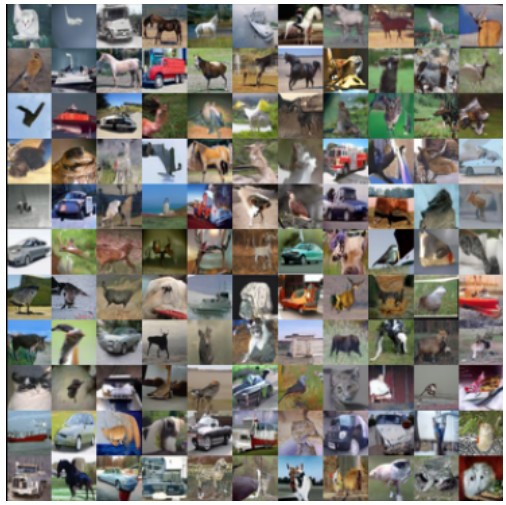 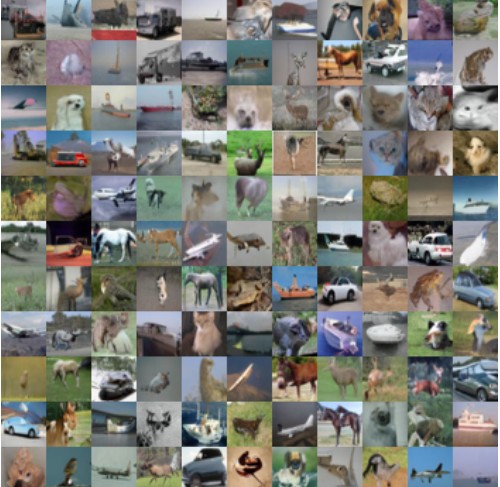

(a) MULAN with velocity reparameterization after 8M training iterations.

(b) VDM after 10M training iterations.

Figure 6: CIFAR-10 samples generated by different methods.

## H.2 ImageNet-32

ImageNet-32 is a dataset derived from ImageNet [7], where the original images have been resized to a resolution of $32 \times 32$. This dataset comprises 1,281,167 training samples and 50,000 test samples, distributed across 1,000 labels.

Randomly generated samples for the ImageNet datasaet are provided in Fig. 7 for MULAN and Fig. 8 for VDM. We visualize the noise schedule in Fig. 13.

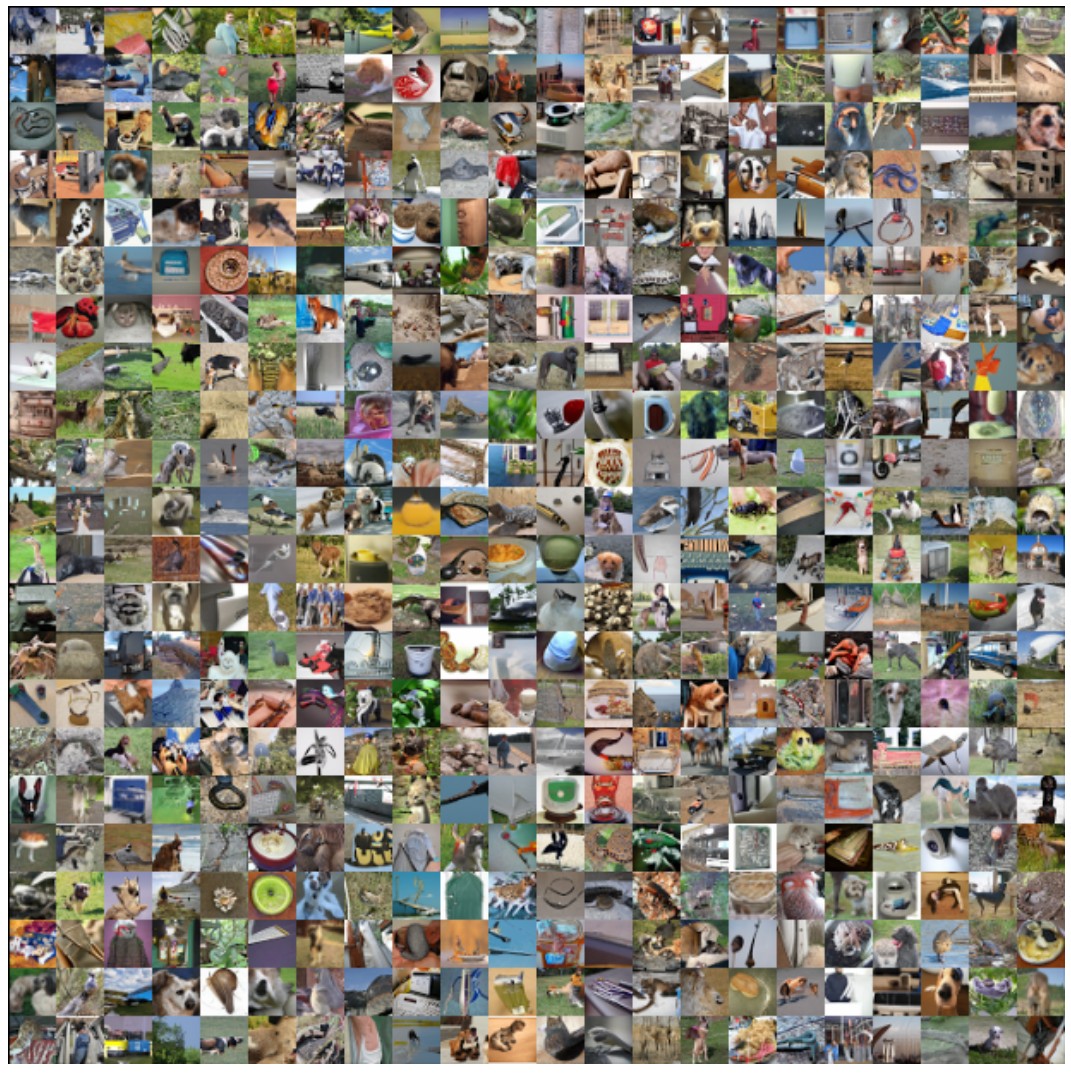

Figure 7: MᴜLAN with noise parameterization after 2M training iterations.

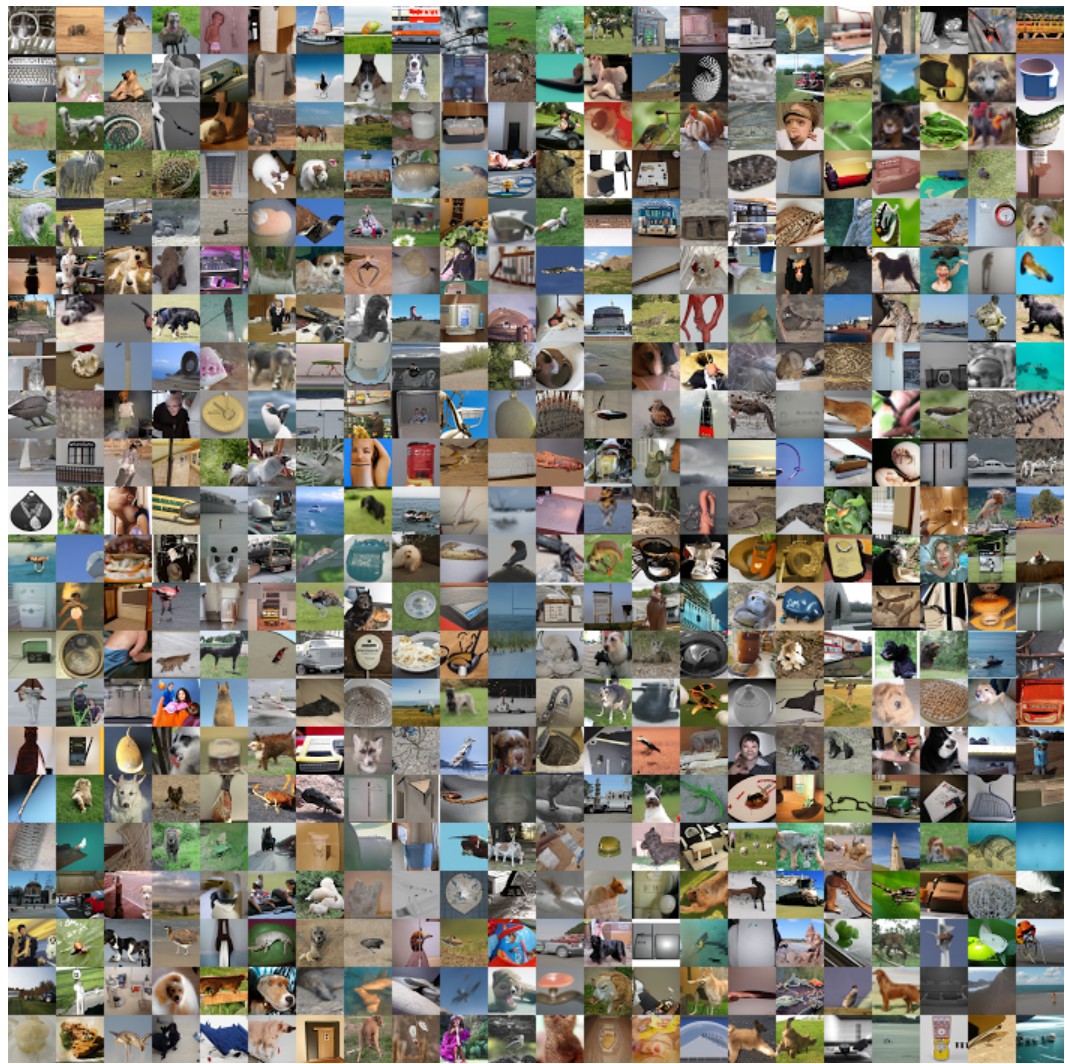

Figure 8: VDM after 2M training iterations.

## H.3 Frequency

To see if MuLAN learns different noise schedules for parts of the images with different frequencies, we modify the images in the CIFAR-10 dataset where we modify an image where we randomly remove the low frequency component an image or remove the high frequency with equal probability. Fig. 9a shows the training samples. MuLAN was trained for 500K steps. The samples generated by MuLAN is shown in Fig. 9b. The corresponding noise schedules is shown in Fig. 13. As compared to CIFAR-10, we notice that the spatial variation in the noise schedule increases (SNRs for all the pixels form a wider band) while the variance of the noise schedule across instances decreases slightly.

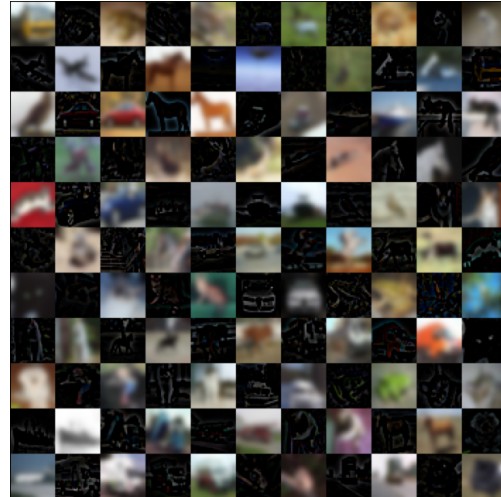
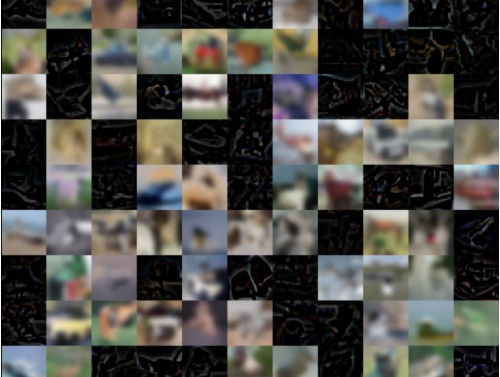

(a) Training samples.

(b) Samples generated by MᴜLAN with noise parameterization after 500K training iterations.

Figure 9: Frequency Split CIFAR-10 dataset.

## H.4 Frequency-2

To see if MᴜLAN learns different noise schedules for images with different frequencies, we modify the images in the CIFAR-10 dataset where we modify an image where we randomly remove the low frequency component an image or remove the high frequency with equal probability. Fig. 9a shows the training samples. MᴜLAN was trained for 500K steps. The samples generated by MᴜLAN is shown in Fig. 9b. The corresponding noise schedules is shown in Fig. 13. As compared to CIFAR-10, we notice that the spatial variation in the noise schedule increases (SNRs for all the pixels form a wider band) and the variance of the noise schedule across instances increases as well.

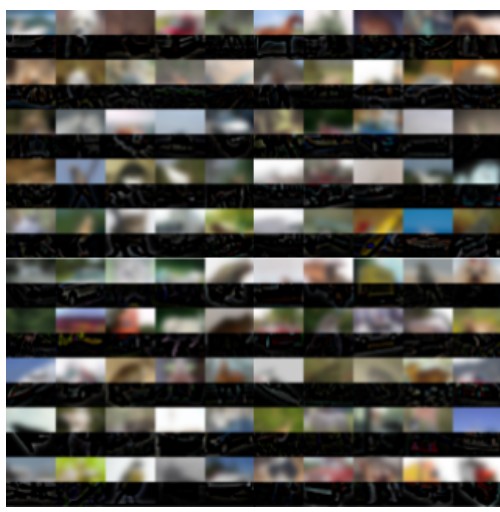
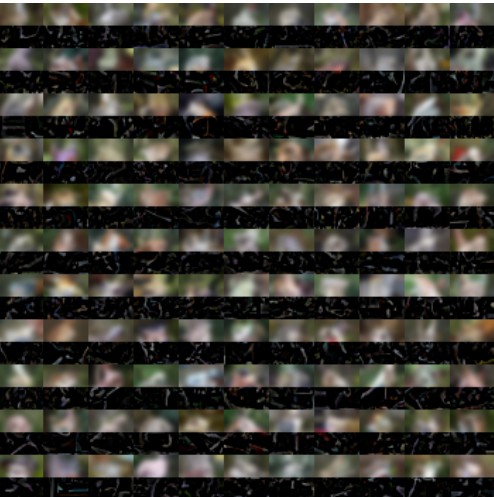

(a) Training samples.

(b) Samples generated by MᴜLAN with noise parameterization after 500K training iterations.

Figure 10: Frequency Split-2 CIFAR-10 dataset.

## H.5 CIFAR-10: Intensity

To see if MᴜLAN learns different noise schedules for images with different intensities, we modify the images in the CIFAR-10 dataset where we randomly convert an image into a low intensity or

a high intensity image with equal probability. Originally, the CIFAR10 images are in the range [0, 255]. To convert an image into a low intensity image we multiply all pixel values by 0.5. To convert an image into a high intensity image we multiply all the pixel values by 0.5 and add 127.5 to them. Fig. 11a shows the training samples. MᴜLAN was trained for 500K steps. The samples generated by MᴜLAN is shown in Fig. 11b. The corresponding noise schedules is shown in Fig. 13. As compared to CIFAR-10, we notice that the spatial variation in the noise schedule slightly increases (SNRs for all the pixels form a wider band) while the variance of the noise schedule across instances slightly decreases.

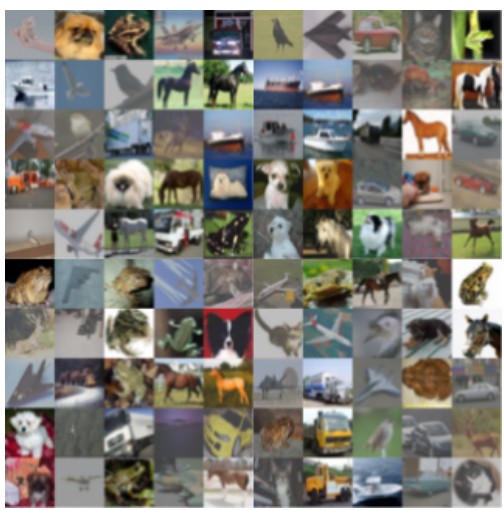

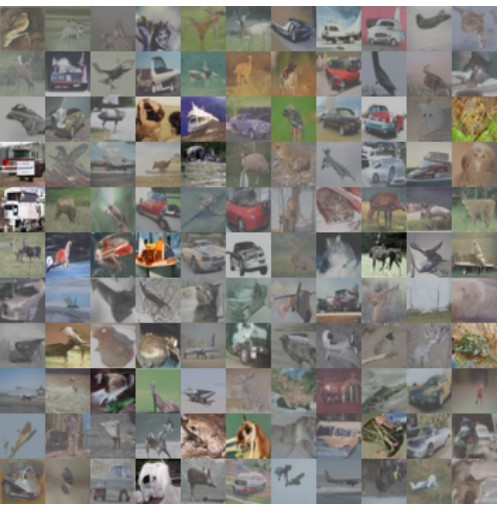

(a) Training samples.

(b) Samples generated by MᴜLAN with noise parameterization after 500K training iterations.

Figure 11: Intensity CIFAR-10 dataset.

## H.6 Mask

We modify the CIFAR-10 dataset where we randomly mask (i.e. replace with **0**s) the top of an image or the bottom half of an image with equal probability. Fig. 12a shows the training samples. MᴜLAN was trained for 500K steps. The samples generated by MᴜLAN is shown in Fig. 12b. The corresponding noise schedules is shown in Fig. 13. As compared to CIFAR-10, we notice that the spatial variation in the noise schedule slightly increases (SNRs for all the pixels form a wider band) while the variance of the noise schedule across instances decreases.

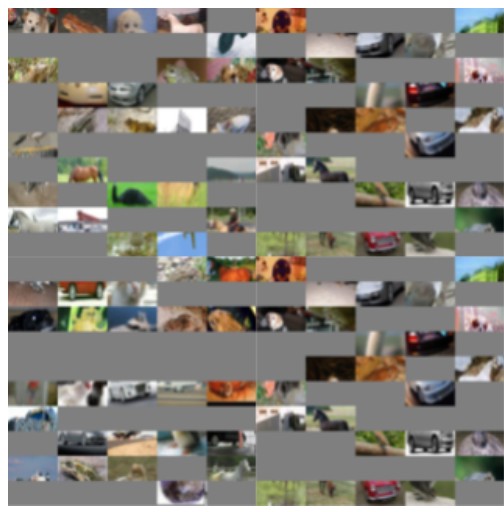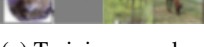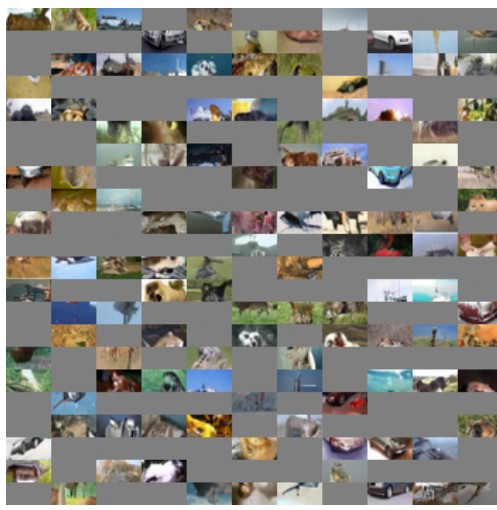

(a) Training samples.

(b) Samples generated by MULAN with noise parameterization after 500K training iterations.

Figure 12: Intensity CIFAR-10 dataset.

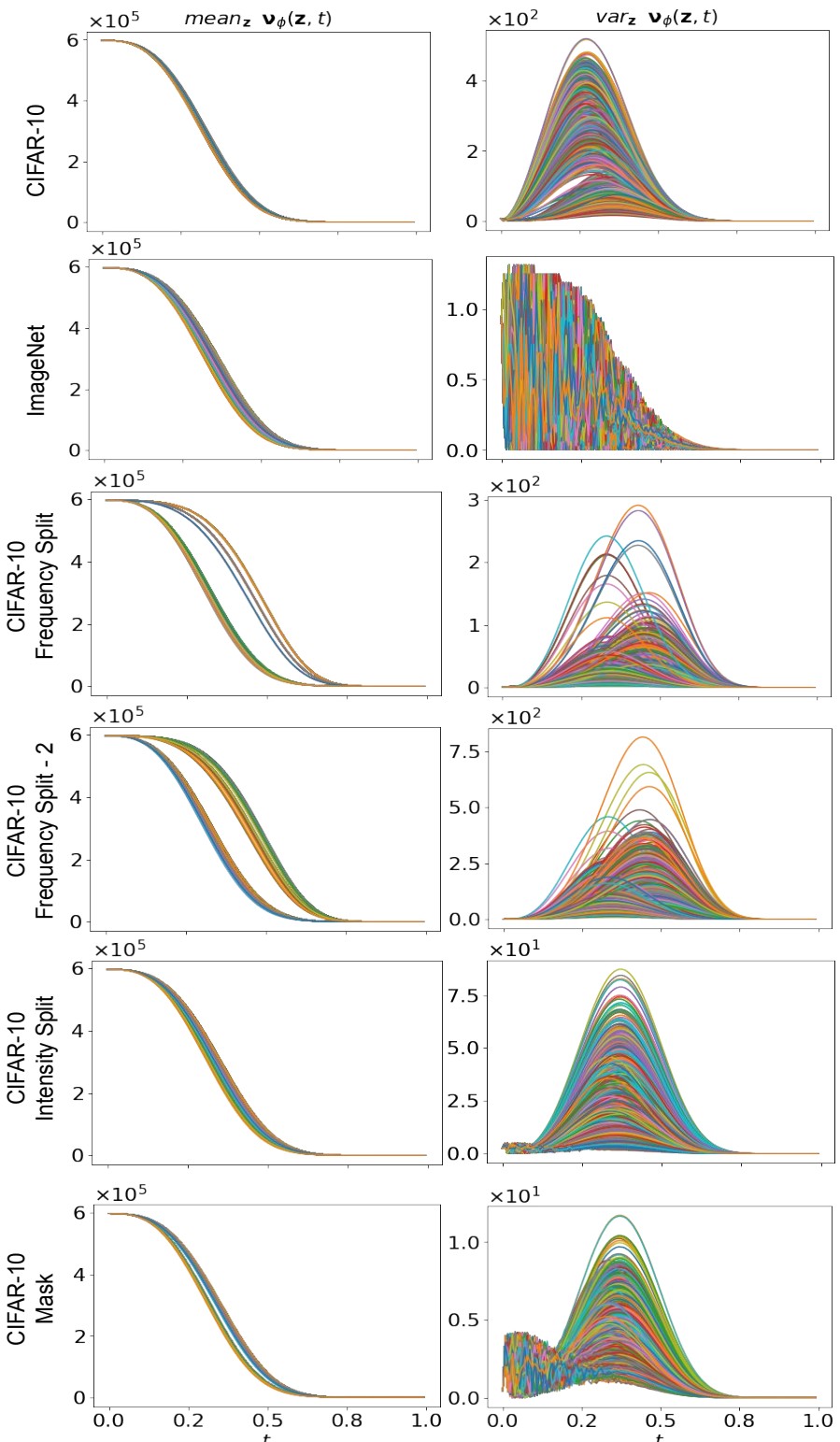

Figure 13: signal-to-noise ratio for different datasets.

# Appendix I  Likelihood Estimation

We used both Variance Lower Bound (VLB) and ODE-based methods to compute BPD.

## I.1  VLB Estimate

In the VLB-based approach, we employ Eq. 9. To compute $\mathcal{L}_{\text{diffusion}}$, we use $T = 128$ in Eq. 10, discretizing the timesteps, $t \in [0, 1]$ into 128 bins.

## I.2  Exact likelihood computation using Probability Flow ODE

A diffusion process whose marginal is given by (the same as in Eq. 2),

$$q(\mathbf{x}_t|\mathbf{x}_0) = \mathcal{N}(\mathbf{x}_t; \boldsymbol{\alpha}_t\mathbf{x}_0, \text{diag}(\boldsymbol{\sigma}_t^2)), \ \mathbf{x}_0 \sim q_0(\mathbf{x}_0), \tag{73}$$

can be modeled as the solution to an Itô Stochastic Differential Equation (SDE):

$$\mathrm{d}\mathbf{x}_t = \mathbf{f}(t)\mathbf{x}_t\mathrm{d}t + \mathbf{g}(t)\mathrm{d}\mathbf{w}_t, \ \mathbf{x}_0 \sim q_0(\mathbf{x}_0), \tag{74}$$

where $\mathbf{f}(t) \in \mathbb{R}^d, \mathbf{g}(t) \in \mathbb{R}^d$ take the following expressions [53]:

$$\mathbf{f}(t) = \frac{\mathrm{d}}{\mathrm{d}t} \log \boldsymbol{\alpha}_t,$$

$$\mathbf{g}^2(t) = \frac{\mathrm{d}}{\mathrm{d}t}\boldsymbol{\sigma}_t^2 - 2\boldsymbol{\sigma}_t^2\frac{\mathrm{d}}{\mathrm{d}t} \log \boldsymbol{\alpha}_t$$

The corresponding reverse process, Eq. 4, can also be modelled by an equivalent reverse-time SDE:

$$\mathrm{d}\mathbf{x}_t = [\mathbf{f}(t) - \mathbf{g}(t)^2\nabla_{\mathbf{x}_t} \log q(\mathbf{x}_t|\mathbf{x}_0)]\mathrm{d}t + \mathbf{g}(t)\mathrm{d}\bar{\mathbf{w}}_t, \ \mathbf{x}_1 \sim p_\theta(\mathbf{x}_1), \tag{75}$$

where $\bar{\mathbf{w}}$ is a standard Wiener process when time flows backwards from $1 \rightarrow 0$, and $\mathrm{d}t$ is an infinitesimal negative timestep. Song et al. [53] show that the marginals of Eq. 75 can be described by the following Ordinary Differential Equation (ODE) in the reverse process:

$$\mathrm{d}\mathbf{x}_t = \left[\mathbf{f}(t)\mathbf{x}_t - \frac{1}{2}\mathbf{g}^2(t)\nabla_{\mathbf{x}_t} \log q(\mathbf{x}_t)\right] \mathrm{d}t. \tag{76}$$

This ODE, also called the probablity flow ODE, allows us to compute the exact likelihood on any input data via the instantaneous change of variables formula as proposed in Chen et al. [2]. Note that during the reverse process, the term $q(\mathbf{x}_t)$ is unknown and is approximated by parameterized by $p_\theta(\mathbf{x}_t)$. For the probability flow defined in Eq. 76, Chen et al. [2] show that the log- likelihood of $p_\theta(\mathbf{x}_0)$ can be computed using the following equation:

$$\log p_\theta(\mathbf{x}_0) = \log p_\theta(\mathbf{x}_1) - \int_{t=0}^{t=1} \text{tr}\left(\nabla_{\mathbf{x}_t}\mathbf{h}_\theta(\mathbf{x}_t, t)\right)\mathrm{d}t, \tag{77}$$

$$\text{where } \mathbf{h}_\theta(\mathbf{x}_t, t) \equiv \mathbf{f}(t)\mathbf{x}_t - \frac{1}{2}\mathbf{g}^2(t)\nabla_{\mathbf{x}_t} \log p_\theta(\mathbf{x}_t)$$

### I.2.1  Probability Flow ODE for MULAN.

Similarly for the forward process conditioned on the auxiliary latent variable, $\mathbf{z}$,

$$q_\phi(\mathbf{x}_t|\mathbf{x}_0, \mathbf{z}) = \mathcal{N}(\mathbf{x}_t; \boldsymbol{\alpha}_t(\mathbf{z})\mathbf{x}_0, \text{diag}(\boldsymbol{\sigma}_t^2(\mathbf{z}))), \ \mathbf{x}_0 \sim q_0(\mathbf{x}_0), \ \mathbf{z} \sim q_\phi(\mathbf{z}|\mathbf{x}_0), \tag{78}$$

we can extend Eq. 74 in the following manner,

$$\mathrm{d}\mathbf{x}_t = \mathbf{f}(\mathbf{z}, t)\mathbf{x}_t\mathrm{d}t + \mathbf{g}(\mathbf{z}, t)\mathrm{d}\mathbf{w}_t, \ \mathbf{x}_0 \sim q_0(\mathbf{x}_0), \ \mathbf{z} \sim q_\phi(\mathbf{z}|\mathbf{x}_0), \tag{79}$$

to obtain the corresponding SDE formulation. Notice that the random variable $\mathbf{z}$ in the above equation doesn't have a subscript $t$, and hence, $\mathbf{z}$ is drawn from $q_\phi(\mathbf{z}|\mathbf{x}_0)$ once and the same $\mathbf{z}$ is used as $\mathbf{x}_0$ diffuses to $\mathbf{x}_1$. The expressions for $\mathbf{f}(\mathbf{z}, t) : \mathbb{R}^m \times [0, 1] \rightarrow \mathbb{R}^d, \mathbf{g}(\mathbf{z}, t) : \mathbb{R}^m \times [0, 1] \rightarrow \mathbb{R}^d$ is given as follows:

$$\mathbf{f}(\mathbf{z}, t) = \frac{\mathrm{d}}{\mathrm{d}t} \log \boldsymbol{\alpha}_t(\mathbf{z}),$$

$$\mathbf{g}^2(\mathbf{z}, t) = \frac{\mathrm{d}}{\mathrm{d}t}\boldsymbol{\sigma}_t^2(\mathbf{z}) - 2\boldsymbol{\sigma}_t^2(\mathbf{z})\frac{\mathrm{d}}{\mathrm{d}t}\log\boldsymbol{\alpha}_t(\mathbf{z})$$

Recall that $\boldsymbol{\alpha}_t^2(\mathbf{z}) = \mathrm{sigmoid}(-\boldsymbol{\gamma}_\phi(\mathbf{z}, t))$, $\boldsymbol{\sigma}_t^2(\mathbf{z}) = \mathrm{sigmoid}(\boldsymbol{\gamma}_\phi(\mathbf{z}, t))$. Substituting these in the above equations, the expressions for $\mathbf{f}(\mathbf{z}, t)$ and $\mathbf{g}^2(\mathbf{z}, t)$ simplify to the following:

$$\mathbf{f}(\mathbf{z}, t) = -\frac{1}{2}\mathrm{sigmoid}(\boldsymbol{\gamma}_\phi(\mathbf{z}, t))\frac{\mathrm{d}}{\mathrm{d}t}\boldsymbol{\gamma}_\phi(\mathbf{z}, t),$$

$$\mathbf{g}^2(\mathbf{z}, t) = \mathrm{sigmoid}(\boldsymbol{\gamma}_\phi(\mathbf{z}, t))\frac{\mathrm{d}}{\mathrm{d}t}\boldsymbol{\gamma}_\phi(\mathbf{z}, t)$$

The corresponding reverse-time SDE is given as:

$$\mathrm{d}\mathbf{x}_t = [\mathbf{f}(t) - \mathbf{g}(t)^2 \nabla_{\mathbf{x}_t}\log q_\phi(\mathbf{x}_t|\mathbf{x}_0, \mathbf{z})]\mathrm{d}t + \mathbf{g}(t)\mathrm{d}\bar{\mathbf{w}}_t, \ \mathbf{x}_1 \sim p_\theta(\mathbf{x}_1), \ \mathbf{z} \sim p_\theta(\mathbf{z}), \quad (80)$$

where $\bar{\mathbf{w}}$ is a standard Wiener process when time flows backwards from $1 \to 0$, and $\mathrm{d}t$ is an infinitesimal negative timestep. Given, $\mathbf{s}_\theta(\mathbf{x}_t, \mathbf{z})$, an approximation to the true score function, $\nabla_{\mathbf{x}_t}\log q_\phi(\mathbf{x}_t|\mathbf{x}_0, \mathbf{z})$, Song et al. [53] show that the marginals of Eq. 80 can be described by the following Ordinary Differential Equation (ODE):

$$\mathrm{d}\mathbf{x}_t = \left[\mathbf{f}(\mathbf{z}, t) - \frac{1}{2}\mathbf{g}^2(\mathbf{z}, t)\mathbf{s}_\theta(\mathbf{x}_t, \mathbf{z})\right]\mathrm{d}t, \quad (81)$$

Zheng et al. [65] show that the score function, $\mathbf{s}_\theta(\mathbf{x}_t, \mathbf{z})$, for the noise and the v-parameterization is given as follows:

$$\mathbf{s}_\theta(\mathbf{x}_t, \mathbf{z}) = \begin{cases} -\dfrac{\boldsymbol{\epsilon}_\theta(\mathbf{x}_t, t)}{\boldsymbol{\sigma}_t(\mathbf{z})} & \text{Noise parameterization; see Sec. E.1.1 (82a)} \\[2ex] -\mathbf{x}_t - \exp\left(-\dfrac{1}{2}\boldsymbol{\gamma}_\phi(\mathbf{z}, t)\right)\mathbf{v}_\theta(\mathbf{x}_t, \mathbf{z}, t) & \text{v-parameterization; see Sec. E.1.2 (82b)} \end{cases}$$

Applying the change of variables formula [2] on Eq. 81, $\log p_\theta(\mathbf{x}_0|\mathbf{z})$ can be computed in the following manner:

$$\log p_\theta(\mathbf{x}_0|\mathbf{z}) = \log p_\theta(\mathbf{x}_1) - \int_{t=0}^{t=1} \mathrm{tr}\left(\nabla_{\mathbf{x}_t}\mathbf{h}_\theta(\mathbf{x}_t, \mathbf{z}, t)\right)\mathrm{d}t, \quad (83)$$

$$\text{where } \mathbf{h}_\theta(\mathbf{x}_t, \mathbf{z}, t) \equiv \mathbf{f}(\mathbf{z}, t) - \frac{1}{2}\mathbf{g}^2(\mathbf{z}, t)\mathbf{s}_\theta(\mathbf{x}_t, \mathbf{z})$$

The expression for log-likelihood (Eq. 8) is as follows,

$$\log p_\theta(\mathbf{x}_0) \geq \mathbb{E}_{q_\phi(\mathbf{z}|\mathbf{x}_0)}[\log p_\theta(\mathbf{x}_0|\mathbf{z})] - \mathrm{D_{KL}}(q_\phi(\mathbf{z}|\mathbf{x}_0)\|p_\theta(\mathbf{z}))$$

$$\text{Using Eq. 83,}$$

$$= \mathbb{E}_{q_\phi(\mathbf{z}|\mathbf{x}_0)}\left[\log p_\theta(\mathbf{x}_1) - \int_{t=0}^{t=1} \mathrm{tr}\left(\nabla_{\mathbf{x}_t}\mathbf{h}_\theta(\mathbf{x}_t, t, \mathbf{z})\right)\mathrm{d}t\right] - \mathrm{D_{KL}}(q_\phi(\mathbf{z}|\mathbf{x}_0)\|p_\theta(\mathbf{z})) \quad (84)$$

Computing $\mathrm{tr}\left(\nabla_{\mathbf{x}_t}\mathbf{h}_\theta(\mathbf{x}_t, t, \mathbf{z})\right)$ is expensive and we follow Chen et al. [2], Zheng et al. [65], Grathwohl et al. [15] to estimate it with Skilling-Hutchinson trace estimator [50, 19]. In particular, we have

$$\mathrm{tr}\left(\nabla_{\mathbf{x}_t}\mathbf{h}_\theta(\mathbf{x}_t, t, \mathbf{z})\right) = \mathbb{E}_{p(\epsilon)}\left[\epsilon^\top \nabla_{\mathbf{x}_t}\mathbf{h}_\theta(\mathbf{x}_t, t, \mathbf{z})\epsilon\right], \quad (85)$$

where the random variable $\epsilon$ satisfies $\mathbb{E}_{p(\epsilon)}[\epsilon] = \mathbf{0}$ and $\mathrm{Cov}_{p(\epsilon)}[\epsilon] = \mathbf{I}$. Common choices for $p(\epsilon)$ include Rademacher or Gaussian distributions. Notably, the term $\nabla_{\mathbf{x}_t}\mathbf{h}_\theta(\mathbf{x}_t, t, \mathbf{z})\epsilon$ can be computed efficiently using "Jacobian-vector-product" computation in JAX. In our experiments, we follow the exact evaluation procedure for computing likelihood as outlined in Song et al. [53], Grathwohl et al. [15]. Specifically, for the computation of Eq. 85, we employ a Rademacher distribution for $p(\epsilon)$. To calculate the integral in Eq. 84, we utilize the RK45 ODE solver [11] provided by `scipy.integrate.solve_ivp` with `atol=1e-5` and `rtol=1e-5`.

### I.2.2 Dequantization.

Real-world datasets for images or texts often consist of discrete data. Attempting to learn a continuous density model directly on these discrete data points can lead to degenerate outcomes [56] and fail to provide meaningful density estimations. Dequantization [46, 16, 65] is a common solution in such cases. To elaborate, let $x_0$ represent 8-bit discrete data scaled to [-1, 1]. Dequantization methods assume that we have trained a continuous model distribution $p_\theta$ for $x_0$, and define the discrete model distribution by

$$P_\theta(\mathbf{x}_0) = \int_{[-\frac{1}{256}, \frac{1}{256})^d} p_\theta(\mathbf{x}_0 + u)\mathrm{d}u.$$

To train $P_\theta(\mathbf{x}_0)$ by maximum likelihood estimation, variational dequantization [16, 65] introduces a dequantization distribution $q(u|\mathbf{x}_0)$ and jointly train $p_{\mathrm{model}}$ and $q(u|\mathbf{x}_0)$ by a variational lower bound:

$$\log P_\theta(\mathbf{x}_0) \geq \mathbb{E}_{q(u|\mathbf{x}_0)}[p_\theta(\mathbf{x}_0 + u) - \log q(u|\mathbf{x}_0)]. \tag{86}$$

**Truncated Normal Dequantization.** Zheng et al. [65] show that truncated Normal distribution,

$$q(u|\mathbf{x}_0) = \mathcal{TN}\left(\mathbf{0}, \mathbf{I}, -\frac{1}{256}, \frac{1}{256}\right)$$

with mean $\mathbf{0}$, covariance $\mathbf{I}$, and bounds $\left[-\frac{1}{256}, \frac{1}{256}\right]$ along each dimension, leads to a better likelihood estimate. Thus, Eq. 86 simplifies to the following (for details please refer to section A. in Zheng et al. [65]):

$$\log P_\theta(\mathbf{x}_0) \geq \mathbb{E}_{\hat{\epsilon}\sim\mathcal{TN}(\mathbf{0},\mathbf{I},-\tau,\tau)}\left[\log p_\theta\left(\mathbf{x}_0 + \frac{\sigma_\epsilon}{\alpha_\epsilon}\hat{\epsilon}\right)\right] + \frac{d}{2}(1 + \log(2\pi\sigma_\epsilon^2)) - 0.01522 \times d \tag{87}$$

$$\text{with } \frac{\sigma_\epsilon}{\alpha_\epsilon} = \exp(-\frac{1}{2} \times 13.3),$$

$$\sigma_\epsilon = \mathtt{sqrt}(\mathrm{sigmoid}(-13.3)), \text{ and } \tau = 3.$$

$\log p_\theta\left(\mathbf{x}_0 + \frac{\sigma_\epsilon}{\alpha_\epsilon}\hat{\epsilon}\right)$ is evaluated using Eq. 84.

**Importance Weighted Estimator.** Eq. 87 can also be extended to obtain an importance weighted likelihood estimator to get a tighter bound on the likelihood. The variational bound is given by (for details please refer to section A. in Zheng et al. [65]):

$$\log P_\theta(\mathbf{x}_0) \geq \mathbb{E}_{\hat{\epsilon}^{(1)},\ldots,\hat{\epsilon}^{(K)}\sim\mathcal{TN}(\mathbf{0},\mathbf{I},-\tau,\tau)}\left[\log\left(\frac{1}{K}\sum_{i=1}^{K}\frac{p_\theta\left(\mathbf{x}_0 + \frac{\sigma_\epsilon}{\alpha_\epsilon}\hat{\epsilon}^{(k)}\right)}{q(\hat{\epsilon}^{(i)})}\right)\right] + d\log\sigma_\epsilon \tag{88}$$

$$\text{with } \frac{\sigma_\epsilon}{\alpha_\epsilon} = \exp(-\frac{1}{2} \times 13.3), \log\sigma_\epsilon = \frac{1}{2}(-13.3 + \mathrm{softplus}(-13.3)),$$

$$q(\hat{\epsilon}) = \frac{1}{(2\pi Z)^2}\exp\left(-\frac{1}{2}\|\hat{\epsilon}\|_2^2\right), Z = 0.9974613, \text{ and } \tau = 3.$$

Note that for $K = 1$, Eq. 88 is equivalent to Eq. 87; see Zheng et al. [65]. $\log p_\theta\left(\mathbf{x}_0 + \frac{\sigma_\epsilon}{\alpha_\epsilon}\hat{\epsilon}\right)$ is evaluated using Eq. 84. In Table 8, we report BPD values for MULAN on CIFAR10 (8M training steps, v-parameterization) and ImageNet (2M training steps, noise parameterization) using both the VLB-based approach, and the ODE-based approach with $K = 1$ and $K = 20$ importance samples.

Table 8: NLL (mean and 95% Confidence Interval for MULAN) on CIFAR10 (8M training steps, v-parameterization) and ImageNet (2M training steps, noise parameterization) using both the VLB-based approach, and the ODE-based approach. $K = 1$ means that we do not use importance weighted estimator since Eq. 88 is equivalent to Eq. 87 for this case; see Zheng et al. [65].

| Likelihood Estimation type | CIFAR-10 ($\downarrow$) | Imagenet ($\downarrow$) |
|---|---|---|
| VLB-based | $2.59 \pm 10^{-3}$ | $3.71 \pm 10^{-3}$ |
| ODE-based ($K = 1$; Eq. 87) | $2.59 \pm 3 \times 10^{-4}$ | $3.71 \pm 10^{-3}$ |
| ODE-based ($K = 20$; Eq. 88) | $2.55 \pm 3 \times 10^{-4}$ | $3.67 \pm 10^{-3}$ |

