# OpenReview forum: "Diffusion Models With Learned Adaptive Noise"
_NeurIPS.cc/2024/Conference — NeurIPS 2024 spotlight_

### Official Review · Reviewer_uSJa · 2024-07-05

**Soundness:** 2
**Presentation:** 3
**Contribution:** 2
**Rating:** 6
**Confidence:** 4

**Summary:**

This paper introduces the Multivariate Learned Adaptive Noise (MULAN) model, a novel diffusion process that adapts the noise schedule to different regions of an image. The authors claim significant improvements in log-likelihood estimation and training efficiency, challenging the conventional assumption that the ELBO is invariant to the noise process, and demonstrate state-of-the-art performance in density estimation on CIFAR-10 and ImageNet, reducing training steps by 50%.

I appreciate the paper because it is based on the presumption, which I share, that a significant improvement in generative diffusion approaches can be achieved by adjusting the forward process to the data. Specifically, the authors suggest adapting both drift and noise to the input data. My understanding is that the adaptivity of the MULAN model is achieved by utilizing more degrees of freedom associated with the forward process within the ELBO-style learning/training. However, the claim that the “optimal” schedule is spatially inhomogeneous is based purely on empirical evidence. I would like to see the authors turn this empirical finding into a more systematic exploration. For instance, can we draw conclusions about the spatio-temporal details of the optimal schedule, such as whether the earlier stages of the forward process are more spatially inhomogeneous than the later ones? Do we see more inhomogeneity on the periphery of the image? How does the inhomogeneity correlate with a spatially coarse-grained (filtered) version of the image? Is it more beneficial to have inhomogeneity in the drift (deterministic) or diffusion (stochastic) part of the forward process?

The paper also discusses using labels and other high-level features of the image, but it is unclear how these affect or interplay with the spatial inhomogeneity.

Overall, while the paper presents interesting ideas, I feel it is not yet ready for prime time. The empirical claims need to be backed by a more systematic and thorough analysis.

**Strengths:**

see summary

**Weaknesses:**

see summary

**Questions:**

see summary

**Limitations:**

see summary

---

> ### Author Rebuttal · Authors · 2024-08-07
>
> # Response to uSJa
>
> We thank the reviewer for their detailed and thorough feedback. We address their concerns below. In the appendix, we have provided numerous experiments trying to explore how the noise schedule relates to different aspects of an image namely:
>
> 1. Frequency distribution.
> 2. Intensity.
>
> ## Concern 1: Finding Interpretable Spatio-Temporal Variation in the Noise Schedule
>
> We have provided a thorough analysis trying to relate the learnt noise schedule to different properties of the input image such as intensity and frequency in sec. 14 in the Appendix. We summarize three relevant experiments hereby:
>
> 1. CIFAR-10 Frequency Split
>
>     To see if MULAN learns different noise schedules for pixels in an image with different frequencies, we modify the images in the CIFAR-10 dataset where we split an image into high frequency and low frequency components using gaussian blur.
>
>     **Observation** We do notice two separate bands in the noise schedule (see fig. 11) however these bands don’t necessarily correspond to the low / high freq regions in the image. Upon comparing the schedules to that of the original CIFAR10-dataset, we do notice an increase in the inter-spatial variation of the noise schedules.
>
> 2. CIFAR-10 Masking
>
>     We modify the CIFAR-10 dataset where we randomly mask (i.e. replace with 0s) the top of an image or the bottom half of an image with equal probability. Fig. 10a shows the training samples. MULAN was trained for 500K steps. The samples generated by MULAN is shown in Fig. 10b.
>
>     **Observation**  We don’t observe any interpretable patterns in the noise schedule for the masked pixels vs non-masked pixels. Upon comparing the schedules to that of the original CIFAR10-dataset, we do notice an increase in the inter-spatial variation of the noise schedules.
>
> 3. CIFAR-10 Intensity Split
>
>     To see if MULAN learns different noise schedules for images with different intensities, we modify the images in the CIFAR-10 dataset where we randomly convert an image into a low intensity or a high intensity image with equal probability. Fig. 9a shows the training samples. MULAN was trained for 500K steps. The samples generated by MULAN is shown in Fig. 9b.
>
>     **Observation**  We don’t observe any interpretable patterns in the noise schedules for the high intensity images vs low intensity images. Upon comparing the schedules to that of the original CIFAR10-dataset, we do notice an increase in the inter-spatial variation of the noise schedules.
>
>
> **Temporal Variation** In fig. 11 (left) in the appendix, we visualize the noise schedules for all the pixels and we observe that the schedules are spatially identical in the beginning and the end of the diffusion processes i.e. at $t=0$ and $t=1$.
>
> Furthermore, the periphery of the image did not exhibit more inhomogeneity than the other parts of the image.
>
> ## Concern 2: Optimal schedule is spatially inhomogeneous is empirical
>
> We use principles of physics to motivate spatial inhomogeneity of an optimal noise schedule. In Section 3.5, we show that the diffusion loss is a line integral and can be interpreted as the amount of work done, $\int_{0}^{1} \mathbf{f}(\mathbf{r}(t)) \cdot \frac{d}{dt}\mathbf{r}(t) \, dt$,
> along the diffusion trajectory $\mathbf{r}(t) \in \mathbb{R}^n_{+}$  in the presence of a force field $\mathbf{f}(\mathbf{r}(t)) \in \mathbb{R}^n_+$. A line integrand is almost always path-dependent unless its integral corresponds to a conservative force field, which is rarely the case for a diffusion process [1].  The path $\mathbf{r}(t)$ is parameterized by the noise schedule, and thus by learning the noising process, we are effectively learning a path that incurs the least work done. Since this force field can be “arbitrary” as it is parameterized by the denoising neural network, it is highly unlikely that the path of least work done is a straight line joining $\mathbf{r(0)}$ and $\mathbf{r(1)}$. In other words, it is highly “unlikely” that the optimal noise schedule is scalar. Note that a straight line trajectory represents a scalar noise schedule as described in the appendix sec 11.5.
>
> ----
> Reference:
>
> [1] Richard E. Spinney and Ian J. Ford. Fluctuation relations: a pedagogical overview, 2012.
>
>
> ## Concern 3: Inhomogeneity in the drift / diffusion term
>
>
> In this work we consider variance preserving diffusion process which means that we can’t possibly disentangle the inhomogeneity in the drift and the diffusion term. The forward process takes the form
> $\mathbf{x}_t = \mathbf{\alpha}\_{t|s}(\mathbf{z}) \mathbf{x}_s + \sqrt{1 -  \mathbf{\alpha}\_{t|s}^2(\mathbf{z})} \epsilon$
> with $s < t$. Removing the adaptivity in the drift or the diffusion term would mean that the marginals, $\mathbf{x}_t \sim q(\mathbf{x}_t | \mathbf{x}_0)$ are intractable to compute and would require simulating the diffusion process to obtain intermediate latents $\mathbf{x}_t$ such as Diffusion Normalizing Flows [1].
>
> ----
> Reference:
>
> [1] Zhang, Q. and Chen, Y., 2021. Diffusion normalizing flow. *Advances in neural information processing systems*, *34*, pp.16280-16291.

---

> > ### Comment · Reviewer_uSJa · 2024-08-13
> >
> > Good response.  My new score is 6.

---

> > > ### Author Response · Authors · 2024-08-13
> > > **Current rating doesn't reflect the updated score of 6**
> > >
> > > Dear Reviewer,
> > >
> > > Thank you very much for evaluating our manuscript and providing invaluable feedback.
> > >
> > > We would like to bring to your attention that the current rating still reflects the previous score of 4 instead of the **updated score of 6**. We wanted to mention this in case there was any confusion.

---

### Official Review · Reviewer_22Fq · 2024-07-09

**Soundness:** 3
**Presentation:** 3
**Contribution:** 3
**Rating:** 7
**Confidence:** 4

**Summary:**

This paper proposes to extend diffusion models by learning a per-pixel noise schedule for the forward noising process, that can be conditional on a context or on an auxiliary variable. This leads to faster convergence and SOTA results on density estimation on the simple benchmark datasets CIFAR-10 and ImageNet-32.

**Strengths:**

I believe this is overall a good paper. The research question is relevant, the method is theoretically sound, well motivated, and explained clearly and in detail. The experiments are sufficiently extensive and the results are good, although only on 2 benchmark datasets that are relatively simple by today's standards.

**Weaknesses:**

- On tables 2, 7, 8 there are confidence intervals on the means, but it's unclear how many random seeds were used, and what's the impact of randomness (I guess weight initialization). Also, on the plots (I'm looking at those with ablations) there seem to be no CIs.
- Sections 3.2.1 and 3.2.2 are a bit repetitive, since 3.2.2 extends 3.2.1 only in a minor way. Especially Eqs. (3) and (4) are exactly the same (except for the context dependency).
- It would be helpful to have a clear self-contained definition for the forward and backward processes, after introducing everything, to summarize the overall setup.
- In the forward process, the dependency on the context $\mathbf{c}$ or the latent variables $\mathbf{z}$ seems to be only through the element-wise noise schedule. In particular, the mentioned NDM and DiffEnc methods appear to be at least partially orthogonal and might lead to further improvements. As far as I can tell, this is not explicitly discussed.
- Only up to 32x32 images are considered here. I think ImageNet-64 should really be included at this point. Since by now density estimation on these benchmarks works very well, it might be interesting and relevant to scale it up to even higher resolutions, but as a proof of concept I think something like the standard ImageNet-64 (used for example in the VDM paper, too) should be sufficient.
- Typo: at line 54, $\sigma_t$ should be $\sigma_t^2$.

**Questions:**

Why are the results in Table 3 reported separately? I think it would be clearer to include them in Table 2 (maybe separated by a horizontal line from older methods).

**Limitations:**

Yes.

Not necessarily a limitation per se, but I would potentially mention that the forward diffusion process depends on the context or latent variables only through the pixel-wise noise schedule. There are orthogonal methods (which are already mentioned in the paper) that learn the mean of the forward process too.

---

> ### Author Rebuttal · Authors · 2024-08-07
>
> ## Response to 22Fq
>
> We thank the reviewer for their detailed and thorough feedback. We address their concerns below.
>
> ## Concern 1: Evaluating on larger scale experiments (ImageNet-64)
>
> Unfortunately, it is not feasible for us to train an ImageNet-64 model within our academic research group. Using the official VDM Jax codebase, it takes 8 days on 8xA100 to train a VDM model on ImageNet-32. We would need to train multiple models on the larger 64x64 dataset, and this is outside the scope of what out group has access to. However, the 32x32 CIFAR10 and ImageNet datasets on which we report our results are standard benchmarks in the field and we report state-of-the-art performance on these benchmarks.
>
> We anticipate our method to scale to larger datasets with similar computational complexity as existing diffusion models. Mulan requires an encoder that is much smaller than the denoising model, and requires 5-10% of additional memory.
>
> ## Concern 2: Comparisons with other learned diffusion algorithms
>
> Our method can be seen as a special case of a diffusion normalizing flow (DNF), which uses the following forward process: $\text{d}\mathbf{x} = \mathbf{f}_\theta(\mathbf{x}, t)\text{d}t +  g(t)\text{d}\mathbf{w}$ . However, for such general processes, training requires backpropagating through sampling from the model, which is computationally expensive. In practice, this reduces scalability and produces worse results. However, there may be a class of processes between full normalizing flows and our simpler noise processes that will admit scalable training and improved performance.
>
> Our work shares similarity with the NDM and DiffEnc methods, which add an additive learned term to the noise process instead of an multiplicative one like in our paper. They still admit efficient sampling. These methods can potentially be combined with our work: in theory they are orthogonal, but we anticipate that getting them to work will require non-trivial engineering work around architecture tuning and optimization strategies.
>
> We do provide a brief discussion in lines 583-589. We’ll add discussions to the updated manuscript.
>
> ## Concern 3: Other Concerns
>
> We thank the reviewer for pointing out the typo. The reviewers comments on the presentation style such as consolidating sec 3.2.1 and 3.2.2 with self-contained definition for the forward and backward processes is also very useful. We’ll factor these comments into the next version of the manuscript.
>
> ## Question 1: Clarification regarding confidence Intervals
>
> To compute the confidence interval, we calculate the variance, $\sigma$, of the likelihood across all the data points in the validation/test set and report the `95% confidence interval` using the formula: $\pm 1.96 \frac{\sigma}{\sqrt{n}}$, where $n$ denotes the number of data points. Given the very small confidence interval of `0.001` on the likelihood, we did not further randomize the computation of likelihood using additional seeds.

---

> > ### Comment · Reviewer_22Fq · 2024-08-10
> >
> > Thank you for addressing my questions and providing a thorough and insightful rebuttal to all reviewers. I am satisfied with your responses and recommend acceptance, though I will maintain my current score for now.

---

> ### Author Response · Authors · 2024-08-13
> **Thank you for you invaluable feedback**
>
> Dear Reviewer,
>
> We truly appreciate you providing invaluable feedback on our manuscript.
>
> Thanks,
> authors

---

### Official Review · Reviewer_QQVN · 2024-07-09

**Soundness:** 3
**Presentation:** 2
**Contribution:** 3
**Rating:** 7
**Confidence:** 4

**Summary:**

This paper introduces an enhanced framework for variational diffusion models (VDM). Rather than using a uniform noise scheduler for all pixels, the new approach assigns different schedulers to individual pixels, adapting to the data distributions. The authors highlight the novelty of this extension, noting that the Evidence Lower Bound (ELBO) depends on the entire trajectory induced by the scheduler, not just the endpoints, as is the case in traditional VDM. Empirical studies demonstrate that their framework achieves state-of-the-art density estimation on CIFAR10 and ImageNet, reducing the number of training steps by 50%.

**Strengths:**

The ELBO perspective of diffusion models has drawn much attention since the diffusion models' introduction. This paper made a non-trivial extension to the current framework and showed that the well-known assumption is no longer held that noise schedule does not alter EBLO. This observation gives additional flexibility to increase ELBO, resulting in a better performance in density estimation.

**Weaknesses:**

1. In section 3.5, the authors discussed why the generalization makes the ELBO rely on the entire trajectory; however, they state the fact without giving any intuitive explanations. The authors should discuss this issue more. In addition, providing some toy examples to clearly show how the extended framework is differentiated from the existing one could significantly improve the paper's presentation.

2. The FIDs of the proposed model are significantly worse than the existing diffusion models.

3. The FIDs reported for VDM seem inconsistent with the one reported in the original paper. Can authors clarify how the FIDs are obtained.

**Questions:**

Please refer to the Weakness section.

**Limitations:**

I am not aware of any potential negative societal impact of this work.

---

> ### Author Response · Authors · 2024-08-09
> **Response to reviewer QQVN**
>
> ## **Concern 1:** Intuition behind ELBO being dependent on the diffusion trajectory
>
> Imagine you’re piloting a plane across a region where cyclones and strong winds are present. If you plot a straight line course directly through these adverse weather conditions, the journey requires more fuel and effort due to the resistance encountered. Instead, if you navigate around the cyclones and adverse winds, it takes less energy to reach your destination, even though the path might be longer.
>
> This intuition maps to mathematical and physical terms. The trajectory of the plane is denoted by $\mathbf{r}(t) \in \mathbb{R}^n_{+}$ and the forces acting on the plane are given by  $\mathbf{f}(\mathbf{r}(t)) \in \mathbb{R}^n_+$. The work used to navigate the plane is $\int_{0}^{1} \mathbf{f}(\mathbf{r}(t)) \cdot \frac{d}{dt}\mathbf{r}(t) \, dt$. The work here is dependent of the trajectory because  $\mathbf{f}(\mathbf{r}(t)) \in \mathbb{R}^n_+$ is not a conservative field.
>
> In Section 3.5, we argue that the same holds for the diffusion ELBO. The trajectory, $\mathbf{r}(t) \in \mathbb{R}^n_{+}$, is parameterized by the noise schedule which is influenced by complex forces, $\mathbf{f}(\mathbf{r}(t)) \in \mathbb{R}^n_+$ (akin to weather patterns). The diffusion loss can be interpreted as the amount of work done, $\int_{0}^{1} \mathbf{f}(\mathbf{r}(t)) \cdot \frac{d}{dt}\mathbf{r}(t) \, dt$, along the diffusion trajectory in the presence of these forces (the forces are modeled by dimension-wise reconstruction error of the denoising model). By carefully selecting (learning the noise schedule) our path—avoiding “high-resistance” areas where the loss accumulates quickly—we can minimize the overall “energy” expended, measured in terms of the NELBO.
>
> ## **Concern 2:** Worse FIDs than the existing diffusion models.
>
> Note that MuLAN does not incorporate many tricks that improve FID such as exponential moving averages, truncations, specialized learning schedules, etc.; our FID numbers can be improved in future work using these techniques. However, our method yields state-of-the-art log-likelihoods.  We leave the extension to tuning schedules for FID to future work. Optimizing for the likelihood is directly motivated by applied problems such as data compression [2]. In that domain, arithmetic coding techniques can take a generative model and produce a compression algorithm that provably achieves a compression rate (in bits per dimension) that equals the model’s log-likelihood [3]. Other applications of log-likelihood estimation include adversarial example detection [4], semi-supervised learning [5], and others.
>
> ## **Concern 3:** Inconsistent FID numbers for VDM  as reported in the original paper.
>
> Note that in their paper, VDM reported FIDs using the DDPM sampler with `T=1000`, whereas in our paper, we generate samples using an ODE solver that solves the reverse-time Probability Flow ODE, as shown in Eqn (76) for VDM and Eqn (81) for MuLAN. We employ the RK45 ODE solver [6] provided by `scipy.integrate.solve_ivp` with `atol=1e-5` and `rtol=1e-5`. These settings align with prior work by Song et al., 2021 [7].
>
> One method to measure the complexity of the learned noise schedule is by comparing the Number of Function Evaluations (NFEs) required to generate samples using an ODE solver to solve the associated Probability Flow ODE. In Table 1, we aimed to study the difficulty of drawing samples from the diffusion model (measured by NFEs) and the quality of the samples (measured by FID). We observe that MuLAN does not degrade FIDs while improving log-likelihood estimation.
>
> ---
> References:
>
> [1] Theis, L., Oord, A. V. D., & Bethge, M. (2015). A note on the evaluation of generative models. *arXiv preprint arXiv:1511.01844*.
>
> [2] David JC MacKay. *Information theory, inference and learning algorithms*. Cambridge university press, 2003.
>
> [3] Thomas M Cover and Joy A Thomas. Data compression. *Elements of Information Theory*, pp. 103–158, 2005.
>
> [4] Yang Song, Taesup Kim, Sebastian Nowozin, Stefano Ermon, and Nate Kushman. Pixeldefend: Leveraging generative models to understand and defend against adversarial examples. *arXiv preprint arXiv:1710.10766*, 2017.
>
> [5] Zihang Dai, Zhilin Yang, Fan Yang, William W Cohen, and Russ R Salakhutdinov. Good semi-supervised learning that requires a bad gan. *Advances in neural information processing systems*, 30, 2017.
>
> [6] J.R. Dormand and P.J. Prince. A family of embedded runge-kutta formulae. *Journal of Computational and Applied Mathematics*, 6(1):19–26, 1980. ISSN 0377-0427. doi: https://doi. org/10.1016/0771-050X(80)90013-3. URL https://www.sciencedirect.com/science/ article/pii/0771050X80900133.
>
> [7] Yang Song, Conor Durkan, Iain Murray, and Stefano Ermon. Maximum likelihood training of score-based diffusion models. *Advances in neural information processing systems*, 34: 1415–1428, 2021.

---

> > ### Comment · Reviewer_QQVN · 2024-08-10
> > **Thank you for the detailed responses.**
> >
> > I appreciate the authors' thorough responses. All my concerns have been addressed. However, I would appreciate it if the authors could incorporate discussions on the intuition behind ELBO into the final version. I have adjusted the score accordingly. Nice work!

---

> > > ### Author Response · Authors · 2024-08-13
> > > **Thank you for your feedback**
> > >
> > > Dear Reviewer,
> > >
> > > We are immensely grateful to you for providing detailed and thorough feedback on our manuscript. We'll incorporate the discussion on the intuition on ELBO into the final version of the manuscript.
> > >
> > > Thanks,
> > > authors.

---

### Official Review · Reviewer_1B1k · 2024-07-12

**Soundness:** 3
**Presentation:** 3
**Contribution:** 3
**Rating:** 7
**Confidence:** 5

**Summary:**

The authors propose MuLAN, a method to learn a multivariate (pixel-wise for images) noise injection schedule for diffusion models, leading to improved likelihood estimates compared to prior work. They provide extensive experimental results and an ablation study to demonstrate the efficiency of their method.

**Strengths:**

- The paper is well-written and easy to follow.
- The authors present a theoretical motivation for making the noise schedule multivariate and learnable from a variational inference perspective, showing that it can impact the ELBO of the model unlike a univariate schedule.
- The proposed method demonstrates state-of-the-art likelihood estimation results.
- The authors conduct an extensive ablation study to thoroughly investigate the proposed method.

Overall, I believe MuLAN is a well-motivated and logical extension of diffusion models that could be useful in practical applications.

**Weaknesses:**

- According to the experimental results provided, the only benefit of MuLAN is improved likelihood estimation. The experiments are conducted exclusively in the image domain, but MuLAN does not yield better FID scores compared to prior works. As discussed in Section 3.1, a model with better likelihood estimation may be useful for tasks like compression or adversarial example detection. However, the authors only provide likelihood estimation results and do not demonstrate improved practical outcomes in downstream tasks. It's not completely clear where MuLAN should be preferred over other models. Including results from downstream tasks, especially in non-image domains where uniform Gaussian noise may not be as effective, would enhance the paper.
- The derivation of the multivariate noise schedule closely follows the derivations from VDM, substituting the original scalar $\gamma$ function with a multivariate (and conditional) one. This approach limits the novelty of the proposed method.
- The significant part of the contribution seems to come from the introduction of the auxiliary latent variable $z$, as indicated by the ablation study in Section 4.4. I would appreciate a more detailed discussion of this technique and its efficacy. This step appears to add an extra hierarchical layer to the variational autoencoder. Since the vanilla diffusion model can be seen as a hierarchical VAE itself, it's not entirely clear what this addition offers. A more detailed discussion of this idea and its connection to other works, such as diffusion in the latent space of VAEs [1] or diffusion where the noising process does not necessarily match the prior at t=1 [2], would be beneficial.
- The authors note that the learned noise injection schedule is challenging to interpret. However, including some visualizations of the learned schedule at different time steps would provide at least some intuition about its characteristics.

[1] Vahdat, A., Kreis, K., & Kautz, J. (2021). Score-based generative modeling in latent space.

[2] Lee, S., Kim, B., & Ye, J. C. (2023, July). Minimizing trajectory curvature of ode-based generative models.

**Questions:**

- You claim (e.g., on lines 190-193) that MuLAN leads to a tighter NLL-ELBO gap. Do you report the gap for MuLAN and compare it with prior works? In Table 2, to which you refer, you only provide likelihood estimation results, without an ELBO comparison.
- At the end of the day, the polynomial parameterization of the $\gamma$ function involves just three parameters, which seems significantly less flexible than the parameterization from VDM. Am I correct in understanding that you claim that the main reason the polynomial parameterization performs better is because it is easier to optimize?
- As you discuss, the polynomial parameterization may have two points with zero derivative. Do you restrict the polynomial parameterization to have zero derivative at t=0 and t=1?
- Could you clarify how you calculate the likelihood? In regular diffusion, we may integrate the density starting from the unit Gaussian distribution and following the ODE dynamics. However, in MuLAN, you introduce auxiliary variables z, and it’s unclear how to incorporate them into the likelihood estimation pipeline. Do you simply sample z from its prior and then integrate the ODE starting from the data point x, conditionally on z?
- In Section 4.3, where you discuss connections with other approaches that also learn the forward process, you mention that these papers do not provide experimental results on the same version of the ImageNet dataset. However, from what I see, NDM references the same version of ImageNet. I think it’s worth including this comparison in Table 3. While NDM reports a lower NLL score, as you discussed, your approach is significantly different, so I don’t believe it characterizes your work negatively.

**Limitations:**

The authors have adequately addressed the limitations.

---

> ### Author Response · Authors · 2024-08-09
> **Response to reviewer 1B1k (1/3)**
>
> We want to thank the reviewer for their constructive feedback. We address each concern below.
>
> ## Concern 1: Demonstrating the utility of density estimation on downstream tasks
>
> Optimizing for the likelihood is directly motivated by applied problems such as data compression [2]. In that domain, arithmetic coding techniques can take a generative model and produce a compression algorithm that provably achieves a compression rate (in bits per dimension) that equals the model’s log-likelihood [3]. Other applications of log-likelihood estimation include adversarial example detection [4], semi-supervised learning [5], and others. In this work, we take the first step of improving the likelihood, and we aim to explore major downstream applications in future work. We note that one immediate simple application of our work is faster training: we attain strong likelihoods and FIDs on ImageNet using half the training steps (Table 1).
>
> Note that MuLAN does not incorporate many tricks that improve FID such as exponential moving averages, truncations, specialized learning schedules, etc.; our FID numbers can be improved in future work using these techniques. However, our method yields state-of-the-art log-likelihoods.  We leave the extension to tuning schedules for FID to future work.
>
> ---
>
> References:
>
> [1] Theis, L., Oord, A. V. D., & Bethge, M. (2015). A note on the evaluation of generative models. *arXiv preprint arXiv:1511.01844*.
>
> [2] David JC MacKay. *Information theory, inference and learning algorithms*. Cambridge university press, 2003.
>
> [3] Thomas M Cover and Joy A Thomas. Data compression. *Elements of Information Theory*, pp. 103–158, 2005.
>
> [4] Yang Song, Taesup Kim, Sebastian Nowozin, Stefano Ermon, and Nate Kushman. Pixeldefend: Leveraging generative models to understand and defend against adversarial examples. *arXiv preprint arXiv:1710.10766*, 2017.
>
> [5] Zihang Dai, Zhilin Yang, Fan Yang, William W Cohen, and Russ R Salakhutdinov. Good semi-supervised learning that requires a bad gan. *Advances in neural information processing systems*, 30, 2017.
>
> ## Concern 2: Discussion on auxiliary latent variables
>
> Diffusion models augmented with an auxiliary latent space [1, 2] have been used for representation learning. These auxiliary latents, $\mathbf{z} \in \mathbb{R}^m$, have a smaller dimensionality than the input $\mathbf{x}_0 \in \mathbb{R}^d$ and are used to encode a high-level semantic representation of $\mathbf{x}_0$.
>
> In this work we make an observation that an input conditioned multivariate noise schedule can indeed improve likelihoods unlike previously thought so. However, simply conditioning on the noise schedule is challenging because of the reasons mentioned in Sec 10.2 in the appendix. In this work we resolve these challenges by conditioning the noise schedule on the input via an auxiliary latent space.
>
> The key differences between these methods:
>
> |  | Learned noise | Multivariate noise | Input Conditioned noise | Auxiliary latents | Noise parameterization |
> | --- | --- | --- | --- | --- | --- |
> | InfoDiffusion [1] | No | No | No | In denoising process | Cosine schedule |
> | MuLAN (ours) | **Yes** | **Yes** | **Yes** | In **noising** & denoising processes | Polynomial (**novel**) |
>
> These auxiliary latents, $\mathbf{z} \sim p_\theta(\mathbf{z})$ , differ from the noisy inputs, $\mathbf{x}_t$ in the sense that these latents are kept fixed throughout the reverse generation process; however, the $\mathbf{x}_t$ is progressively denoised to get clean $\mathbf{x}_0$.
>
> References:
>
> [1] Yingheng Wang, Yair Schiff, Aaron Gokaslan, Weishen Pan, Fei Wang, Christopher De Sa, and Volodymyr Kuleshov. Infodiffusion: Representation learning using information maximizing diffusion models. In *International Conference on Machine Learning*, pp. xxxx–xxxx. PMLR, 2023.
>
> [2] Ruihan Yang and Stephan Mandt. Lossy image compression with conditional diffusion models, 2023.

---

> ### Author Response · Authors · 2024-08-09
> **Response to reviewer 1B1K (2/3)**
>
> ## Concern 3: Visualizations of the learned schedule
>
> We have provided visualizations for the noise schedule for different pixels in Fig. 11 (left column) in the appendix for various datasets such as CIFAR-10 and ImageNet. Furthermore, we have numerous experiments trying to relate the learned noise schedule to different properties of the input image such as intensity and frequency in sec. 14 in the Appendix. We summarize three relevant experiments hereby:
>
> 1. CIFAR-10 Frequency Split
>
>     To see if MULAN learns different noise schedules for pixels in an image with different frequencies, we modify the images in the CIFAR-10 dataset where we split an image into high frequency and low frequency components using gaussian blur.
>
>     **Observation** We do notice two separate bands in the noise schedule (see fig. 11) however these bands don’t necessarily correspond to the low / high freq regions in the image. Upon comparing the schedules to that of the original CIFAR10-dataset, we do notice an increase in the inter-spatial variation of the noise schedules.
>
> 2. CIFAR-10 Masking
>
>     We modify the CIFAR-10 dataset where we randomly mask (i.e. replace with 0s) the top of an image or the bottom half of an image with equal probability. Fig. 10a shows the training samples. MULAN was trained for 500K steps. The samples generated by MULAN is shown in Fig. 10b.
>
>     **Observation**  We don’t observe any interpretable patterns in the noise schedule for the masked pixels vs non-masked pixels. Upon comparing the schedules to that of the original CIFAR10-dataset, we do notice an increase in the inter-spatial variation of the noise schedules.
>
> 3. CIFAR-10 Intensity Split
>
>     To see if MULAN learns different noise schedules for images with different intensities, we modify the images in the CIFAR-10 dataset where we randomly convert an image into a low intensity or a high intensity image with equal probability. Fig. 9a shows the training samples. MULAN was trained for 500K steps. The samples generated by MULAN is shown in Fig. 9b.
>
>     **Observation**  We don’t observe any interpretable patterns in the noise schedules for the high intensity images vs low intensity images. Upon comparing the schedules to that of the original CIFAR10-dataset, we do notice an increase in the inter-spatial variation of the noise schedules.
>
>
> **Temporal Variation** In fig. 11 (left) in the appendix, we visualize the noise schedules for all the pixels and we observe that the schedules are spatially identical in the beginning and the end of the diffusion processes i.e. at $t=0$ and $t=1$.
>
> ## Concern 4:  Comparisons to other works
>
> ### **Variational Diffusion Models**
>
> We'd like to emphasize the fact that this work **changes the understanding of noise schedules** and how they affect the likelihood of a diffusion model. It discards the existing notion that the likelihood of a diffusion model is invariant to the choice of noise schedules. To achieve this we introduce the concept of adaptive noise schedules where the noise schedule is conditioned on the input via auxiliary latent variables.
>
> We include a comparison to VDM hereby that illustrates the key differences between these methods.
>
> |  | Learned noise | Multivariate noise | Input Conditioned noise | Auxiliary latents | Noise parameterization |
> | --- | --- | --- | --- | --- | --- |
> | VDM (Kingma et al., 2021) | Yes | No | No | No | Monotonic neural network |
> | MuLAN (ours) | Yes | Yes | Yes | In noising & denoising processes | Polynomial, sigmoid |
>
> ### **Score-based generative modeling in latent space.**
>
> Latent Score-based Generative Model (LSGM) [1] defines a diffusion process in the latent space, rather than directly on the input data. In contrast, MuLAN defines the diffusion process on the input data but conditions the forward and reverse processes on the auxiliary latent space. Furthermore, we’d like to highlight that MuLAN achieves a Negative Log Likelihood (NLL) of 2.65 on CIFAR-10, which is significantly better than LSGM’s 2.87.
>
> ### **Minimizing trajectory curvature of ode-based generative models.**
>
> In Minimizing Trajectory Curvature (MTC) of ODE-based generative models, the primary goal is to design the forward diffusion process that is optimal for fast sampling; however, MuLAN strives to learn a more expressive forward process that optimizes for log-likelihood. In the appendix, we have provided a detailed explanation in sec 7.4  and Tab. 6 highlights the key differences between the methods.
>
> References:
>
> [1] Vahdat, A., Kreis, K., & Kautz, J. (2021). Score-based generative modeling in latent space.
>
> [2] Lee, S., Kim, B., & Ye, J. C. (2023, July). Minimizing trajectory curvature of ode-based generative models.

---

> ### Author Response · Authors · 2024-08-09
> **Response to reviewer 1B1k (3/3)**
>
> ## Concern 5: Other concerns
>
> > ### Polynomial Parameterization
>
> Although the Monotonic Neural Network parameterization proposed in VDM is intended to be more expressive, the design of the neural network restricts its expressivity. Specifically, the weights must always be positive, and it utilizes sigmoid activations to maintain the monotonicity of the noise schedule. We observed that in deeper layers, the activations could easily saturate, thus limiting the expressivity of the function. In contrast, the polynomial parameterization features only 3 free variables that can be set arbitrarily while still ensuring the noise schedule’s monotonicity.
>
> Furthermore, as the reviewer noted, the polynomial parameterization may have two zero derivatives between $t=0$ and $t=1$. However, we do not explicitly set the derivatives at the endpoints to zero.
>
> > ### Imagenet likelihood for NDM
>
> The reason we did not include the likelihood numbers for NDM on ImageNet is that they employed numerous tricks to achieve improved likelihood, which the baselines reported in Table 2 did not use, namely:
>
> 1. Data augmentation. NDM pre-processes their data using flipping which is a commonly used trick to improve the likelihood. For ex, on CIFAR-10, VDM w/o data augmentation achieves an NLL of 2.65 but with data augmentation, the NLL improves to 2.49; see [1].
> 2. Parameter Count.  NDM [2] uses an encoder model that is of the same size as that of the denoising model (see sec 4.1 in [2]) and hence they use almost 2x more parameters than ours. In contrast, MuLAN’s encoder is only ~10% size of the denoising model.
>
> References:
>
> [1] Diederik Kingma, Tim Salimans, Ben Poole, and Jonathan Ho. Variational diffusion models. *Advances in neural information processing systems*, 34:21696–21707, 2021.
>
> [2]  Grigory Bartosh, Dmitry Vetrov, and Christian A Naesseth. Neural diffusion models. *arXiv preprint arXiv:2310.08337*, 2023.
>
> > ### Clarification on Likelihood computation
>
> To compute the likelihood of a datapoint $\mathbf{x}_0$ we need to evaluate the following equation (eqn. (84) in the paper):
>
> $- \mathbb{E}\_{q\_\phi(\mathbf{z}|\mathbf{x}\_0)} \log p\_\theta(\mathbf{x}\_0 | \mathbf{z}) +  \mathbb{D}\_{\text{KL}} \left( q\_\phi(\mathbf{z}|\mathbf{x}\_0) \parallel p_\theta(\mathbf{z}) \right)$
>
> where $\mathbf{x}\_0$ denotes the input data,  $\mathbf{z}$ denotes the auxiliary latent variable, $q_\phi(\mathbf{z}|\mathbf{x}\_0)$ and $p_\theta(\mathbf{z})$ denote the approximate posterior and the prior distributions for the auxiliary latent variables respectively, $p_\theta(\mathbf{x}_0 | \mathbf{z})$ is the likelihood of the diffusion model conditioned on the auxiliary latent.
>
> 1. To compute the term $\mathbb{E}\_{q\_\phi(\mathbf{z}|\mathbf{x}\_0)} \log p_\theta(\mathbf{x}\_0 | \mathbf{z})$, we draw a sample $\mathbf{z} \sim q_\phi(\mathbf{z} | \mathbf{x}_0)$ and then use an ODE solver to compute the ODE dynamics in Eqn (83) in the paper.
> 2. The term $\mathbb{D}\_{\text{KL}} \left( q_\phi(\mathbf{z}|\mathbf{x}\_0) \parallel p_\theta(\mathbf{z}) \right)$ is computed in closed form; see line 223 in the paper.

---

> > ### Comment · Reviewer_1B1k · 2024-08-09
> >
> > I thank the authors for their comprehensive rebuttal. I will maintain my score.

---

> > > ### Author Response · Authors · 2024-08-13
> > > **Thank you for the detailed review**
> > >
> > > Dear reviewer,
> > >
> > > We are immensely grateful to you for providing such detailed and thorough feedback on our paper. We will incorporate these discussions into the next version of the manuscript.
> > >
> > > Thanks,
> > > authors

---

### Author Response · Authors · 2024-08-09
**General Response to reviewers**

We thank the reviewers for their helpful feedback. Here is a summary of our clarifications in response to their concerns:

### 1. **Visualization of the learned noise schedule.**
We’d like to point out that the visualizations for the noise schedule has been provided in Fig. 11 (left column) in the appendix for various datasets such as CIFAR-10 and ImageNet. Furthermore, we have numerous experiments trying to relate the learned noise schedule to different properties of the input image such as intensity and frequency in sec. 14 in the Appendix.

### 2. **The intuition behind ELBO is dependent on the diffusion trajectory.**
Imagine you’re piloting a plane across a region where cyclones and strong winds are present. If you plot a straight line course directly through these adverse weather conditions, the journey requires more fuel and effort due to the resistance encountered. Instead, if you navigate around the cyclones and adverse winds, it takes less energy to reach your destination, even though the path might be longer.

This intuition maps to mathematical and physical terms. The trajectory of the plane is denoted by $\mathbf{r}(t) \in \mathbb{R}^n_{+}$ and the forces acting on the plane are given by  $\mathbf{f}(\mathbf{r}(t)) \in \mathbb{R}^n_+$. The work used to navigate the plane is $\int_{0}^{1} \mathbf{f}(\mathbf{r}(t)) \cdot \frac{d}{dt}\mathbf{r}(t) \, dt$. The work here is dependent of the trajectory because  $\mathbf{f}(\mathbf{r}(t)) \in \mathbb{R}^n_+$ is not a conservative field.

In Section 3.5, we argue that the same holds for the diffusion ELBO. The trajectory, $\mathbf{r}(t) \in \mathbb{R}^n_{+}$, is parameterized by the noise schedule which is influenced by complex forces, $\mathbf{f}(\mathbf{r}(t)) \in \mathbb{R}^n_+$ (akin to weather patterns). The diffusion loss can be interpreted as the amount of work done, $\int_{0}^{1} \mathbf{f}(\mathbf{r}(t)) \cdot \frac{d}{dt}\mathbf{r}(t) \, dt$, along the diffusion trajectory in the presence of these forces (the forces are modeled by dimension-wise reconstruction error of the denoising model). By carefully selecting (learning the noise schedule) our path—avoiding “high-resistance” areas where the loss accumulates quickly—we can minimize the overall “energy” expended, measured in terms of the NELBO.

### 3. **FIDs do not yet match those of existing diffusion models.**
Note that MuLAN does not incorporate many tricks that improve FID such as exponential moving averages, truncations, specialized learning schedules, etc.; our FID numbers can be improved in future work using these techniques. However, our method yields state-of-the-art log-likelihoods.  We leave the extension to tuning schedules for FID to future work. Optimizing for the likelihood is directly motivated by applied problems such as data compression [2]. In that domain, arithmetic coding techniques can take a generative model and produce a compression algorithm that provably achieves a compression rate (in bits per dimension) that equals the model’s log-likelihood [3]. Other applications of log-likelihood estimation include adversarial example detection [4], semi-supervised learning [5], and others.

Finally, we'd like to emphasize the fact that this work **changes the understanding of noise schedules** and how they affect the likelihood of a diffusion model. It discards the existing notion that the likelihood of a diffusion model is invariant to the choice of noise schedules.

---
**References:**

[1] Theis, L., Oord, A. V. D., & Bethge, M. (2015). A note on the evaluation of generative models. *arXiv preprint arXiv:1511.01844*.

[2] David JC MacKay. *Information theory, inference and learning algorithms*. Cambridge university press, 2003.

[3] Thomas M Cover and Joy A Thomas. Data compression. *Elements of Information Theory*, pp. 103–158, 2005.

[4] Yang Song, Taesup Kim, Sebastian Nowozin, Stefano Ermon, and Nate Kushman. Pixeldefend: Leveraging generative models to understand and defend against adversarial examples. *arXiv preprint arXiv:1710.10766*, 2017.

[5] Zihang Dai, Zhilin Yang, Fan Yang, William W Cohen, and Russ R Salakhutdinov. Good semi-supervised learning that requires a bad gan. *Advances in neural information processing systems*, 30, 2017.

---

### Decision · Program_Chairs · 2024-09-25

**Decision:**

Accept (spotlight)

**Comment:**

The paper presents a variant of diffusion models that excels at likelihood estimation. The method is based on generalizing the scaling and noise variance to a multi-variate (per-pixel) version, conditioning them on the data or a representation of data, and learn the noise schedule along with training. The authors remarked that this introduces additional flexibility in diffusion models for optimizing ELBO, as now the noise schedule also affects the loss in general cases. The method achieves a new state-of-the-art of likelihood estimation on a few image datasets. The reviewers acknowledged the innovation and the empirical achievement, while also raised a few imperfections, e.g., still not resolving the long-standing conflict between generation quality and density estimation, results on higher-resolution images, and more realistic downstream utilities. Nevertheless, the contribution and quality of finish are compelling enough for an accept.